# Communication-Efficient Federated AUC Maximization with Cyclic Client Participation

**Umesh Vangapally**[*]                                                                      *Z2008841@students.niu.edu*
**Wenhan Wu**[†]                                                                              *wwu25@uncc.edu*
**Chen Chen**[‡]                                                                              *chen.chen@crcv.ucf.edu*
**Zhishuai Guo**[*]                                                                           *zguo@niu.edu*

[*]*Department of Computer Science, Northern Illinois University*
[†]*Department of Computer Science, University of North Carolina at Charlotte*
[‡]*Center for Research in Computer Vision, University of Central Florida*

**Reviewed on OpenReview:** *https://openreview.net/forum?id=18yPFLbVRy*

## Abstract

Federated AUC maximization is a powerful approach for learning from imbalanced data in federated learning (FL). However, existing methods typically assume full client availability, which is rarely practical. In real-world FL systems, clients often participate in a cyclic manner: joining training according to a fixed, repeating schedule. This setting poses unique optimization challenges for the non-decomposable AUC objective. This paper addresses these challenges by developing and analyzing communication-efficient algorithms for federated AUC maximization under cyclic client participation. We investigate two key settings: First, we study AUC maximization with a squared surrogate loss, which reformulates the problem as a nonconvex-strongly-concave minimax optimization. By leveraging the Polyak-Łojasiewicz (PL) condition, we establish a state-of-the-art communication complexity of $\widetilde{O}(1/\epsilon^{1/2})$ and iteration complexity of $\widetilde{O}(1/\epsilon)$. Second, we consider general pairwise AUC losses. We establish a communication complexity of $O(1/\epsilon^3)$ and an iteration complexity of $O(1/\epsilon^4)$. Further, under the PL condition, these bounds improve to communication complexity of $\widetilde{O}(1/\epsilon^{1/2})$ and iteration complexity of $\widetilde{O}(1/\epsilon)$. Extensive experiments on benchmark tasks in image classification, medical imaging, and fraud detection demonstrate the superior efficiency and effectiveness of our proposed methods.

## 1 Introduction

FL enables collaborative model training without centralizing raw data, making it invaluable for privacy-sensitive domains like healthcare, finance and mobile computing (Pati et al., 2022; McMahan et al., 2017; Konečnỳ et al., 2016; Hard et al., 2018; Kairouz et al., 2021c). However, most FL research focuses on Empirical Risk Minimization (ERM) (Yu et al., 2019b; Stich, 2018; Mohri et al., 2019), which is ill-suited for imbalanced data, as it minimizes aggregate surrogate losses that do not directly address model bias toward majority classes (Elkan, 2001). In contrast, directly optimizing metrics such as the Area Under the ROC Curve (AUC) has been shown to better capture performance under class imbalance (Ying et al., 2016; Cortes & Mohri, 2003; Rakotomamonjy, 2004; Yuan et al., 2021b).

Building on prior work (Gao et al., 2013; Ying et al., 2016; Zhao et al., 2011b; Kotlowski et al., 2011; Gao & Zhou, 2015; Calders & Jaroszewicz, 2007; Charoenphakdee et al., 2019a), deep AUC maximization problem can be formulated as:

$$\min_{\mathbf{w}\in\mathbb{R}^d} \mathbb{E}_{\mathbf{z}\in\mathcal{S}_+} \mathbb{E}_{\mathbf{z}'\in\mathcal{S}_-} \psi(h(\mathbf{w};\mathbf{z}), h(\mathbf{w};\mathbf{z}')). \tag{1}$$

where $\mathcal{S}_+$ and $\mathcal{S}_-$ denote the sets of positive and negative samples, respectively. The function $\psi(\cdot,\cdot)$ is a surrogate loss, and $h(\mathbf{w};\mathbf{z})$ represents the prediction score of input data $\mathbf{z}$ by a deep neural network

parameterized by $\mathbf{w}$. Recent work has extended AUC maximization to the federated setting (Guo et al., 2020; Yuan et al., 2021a; Guo et al., 2023a). Guo et al. (2020); Yuan et al. (2021a) studied AUC maximization with a squared surrogate loss, reformulating the problem as a minimax optimization without explicit pairwise coupling. Guo et al. (2023a) considered a more general pairwise AUC formulation by allowing clients to share prediction scores. However, a major limitation of these methods is the assumption of full client availability in every communication round. They either require all clients to participate or randomly sample a subset each round—conditions rarely met in practice. Real-world FL systems often exhibit cyclic client participation, where clients join training in a fixed, repeating schedule (Cho et al., 2023). Cyclic Client Participation (CyCP) is a structured approach designed to transition FL from idealized models to practical, real-world deployments where client availability is inherently intermittent due to factors such as battery limitations and unstable network connectivity (Huba et al., 2022; Paulik et al., 2021). By enforcing a guarantee that all clients or client groups participate within a defined meta epoch, CyCP offers significant benefits over purely random sampling: it ensures comprehensive data coverage across the entire population—an important property for mitigating non-IID data bias (Cho et al., 2023; Zhu et al., 2023)—and, critically, this controlled, predictable participation frequency enhances privacy preservation by strictly limiting how often each client contributes with the global model (Kairouz et al., 2021a). While convergence guarantees have been established for ERM under such settings (Cho et al., 2023; Crawshaw & Liu, 2024; Wang & Ji, 2022), the non-decomposable nature of the AUC objective introduces unique challenges that remain unexplored.

To make federated AUC maximization practical under realistic deployment conditions, we study this problem under cyclic client participation for two key classes of surrogate losses. First, for the *squared surrogate loss* $\psi(a, b) = (1 - a + b)^2$, Problem (1) can be reformulated as a minimax optimization problem (Ying et al., 2016). *The minimax formulation does not require explicit construction of positive–negative pairs, making it naturally well-suited for online learning, where data arrive sequentially, and for federated settings, where data reside across many devices—since it simplifies implementation and avoids cross-client pair management.* Previous studies (Guo et al., 2020; Yuan et al., 2021a; Sharma et al., 2022; Deng & Mahdavi, 2021) developed communication-efficient algorithms for such minimax problems, but all assume full client participation.

Second, we consider *general pairwise surrogate losses* (e.g., sigmoid) for Problem (1). *The minimax reformulation above does not cover all AUC surrogate losses (Zhao et al., 2011a; Kotlowski et al., 2011; Gao & Zhou, 2015; Calders & Jaroszewicz, 2007; Charoenphakdee et al., 2019a). In particular, symmetric pairwise losses have been shown to be more robust to label noise than the squared surrogate loss (Charoenphakdee et al., 2019b; Zhu et al., 2022).* It generalizes to tasks such as bipartite ranking and metric learning (Cohen et al., 1997; Clémençon et al., 2008; Kotlowski et al., 2011; Dembczynski et al., 2012; Radenović et al., 2016; Wu et al., 2017). While Guo et al. (2023a) provided a communication-efficient federated algorithm for this objective, their analysis relies on random client sampling. In contrast, cyclic participation introduces deterministic delays, as only specific clients are active at each step. The specific formulations of these two types of problems are presented in Section 4 and 5, respectively. We discuss the compatibility of our proposed algorithms with multiple specific AUC-consistent losses in Appendix G.

Our contributions are as follows.

**Federated Nonconvex-Strongly-Concave Minimax Problems.** Under cyclic participation, both primal ($\mathbf{v}$) and dual ($\alpha$) updates are complicated by deterministic client scheduling, which breaks the common independent sampling assumption. Consequently, global gradient estimates become biased, invalidating existing convergence analyses in (Guo et al., 2020; Yuan et al., 2021a). To this end, we use a stagewise algorithm, where each stage comprises multiple full cycles of communication with all client groups in a predefined sequence. Our analysis constructs auxiliary sequences that track model states at cycle boundaries. These anchors enable us to bound the drift caused by biased updates within each cycle and disentangle the complex error propagation across client groups. By exploiting the strong concavity in $\alpha$, we further control the dual dynamics. Under the Polyak–Łojasiewicz (PL) condition, our method achieves a communication complexity of $\widetilde{O}(1/\epsilon^{1/2})$, matching the best-known rate in the full-participation setting (Guo et al., 2020)—demonstrating that cyclic participation does not inherently degrade efficiency.

**Federated Pairwise Optimization.** For general pairwise objectives, the key challenge is to handle the coupling between different clients to construct the loss function. We design an active–passive algorithm that decomposes gradients into components computed locally (active) and via shared information from others

(passive). Our analysis traces model updates back to the start of the previous cycle, where all components are virtually evaluated at states independent of current updates, thus mitigating bias. Unlike (Guo et al., 2023a), our approach must handle a delay of two full cycles, making the error dynamics more complex. Nevertheless, our algorithm achieves the same $\tilde{O}(1/\epsilon^3)$ communication complexity (and improved rates under PL), confirming that efficient pairwise optimization remains feasible under cyclic participation.

**Empirical Evaluation** We validate our methods on benchmark datasets in diverse domains, including CIFAR-10, CIFAR-100 (image classification), ChestMNIST (medical imaging), and a large-scale insurance fraud detection dataset. Results consistently demonstrate the effectiveness and efficiency of our proposed algorithms under cyclic client participation.

## 2 Related Work

Our work lies at the intersection of federated optimization for non-ERM objectives and the study of client participation schemes. We briefly review both areas.

### 2.1 Federated Non-ERM Optimization

While the FL literature is dominated by Empirical Risk Minimization (ERM), there is growing interest in non-ERM objectives such as minimax and compositional optimization, motivated by applications like AUC maximization (Ying et al., 2016) and distributionally robust optimization (Namkoong & Duchi, 2016).

**Federated Minimax Optimization.** Several works have studied federated minimax problems (Guo et al., 2020; Yuan et al., 2021a; Deng & Mahdavi, 2021; Sharma et al., 2022). Closest to ours, Guo et al. (2020) and Yuan et al. (2021a) address AUC maximization by reformulating it as a nonconvex–strongly-concave minimax problem. They leverage the PL condition and employ stagewise algorithms that solve a sequence of subproblems. Guo et al. (2020) achieved an iteration complexity of $O(1/(\mu^2\epsilon))$ and a communication complexity of $O(1/(\mu^{3/2}\epsilon^{1/2}))$, where $\mu$ is the PL modulus. Yuan et al. (2021a) later incorporated variance reduction techniques to further reduce the communication complexity. A key limitation of these approaches is their assumption of full or independently random client participation, which our work relaxes.

**Federated Compositional Optimization.** Compositional objectives introduce a nested structure that is especially challenging in FL. Gao et al. (2022) analyze a simpler setting where each client $k$ has entirely local inner ($g_k$) and outer ($f_k$) functions, avoiding inter-client coupling. In contrast, Guo et al. (2023a) consider a more general pairwise objective, where data across clients are inherently coupled. Their algorithm employs an active–passive gradient decomposition strategy and achieves linear speedup under random client sampling. Other heuristic approaches exist (Wu et al., 2022; Li & Huang, 2022), though they generally lack rigorous theoretical guarantees.

### 2.2 Client Participation in Federated Learning

Various client participation schemes have been explored in FL to balance efficiency, scalability, and robustness. The simplest strategy is full participation, where all clients perform local updates in every round (Stich, 2018; Yu et al., 2019a;b). Unbiased client sampling selects a random subset of clients each round (Karimireddy et al., 2020; Jhunjhunwala et al., 2022; Yang et al., 2021a), while biased sampling prioritizes clients based on predefined rules (Ruan et al., 2021) or data characteristics such as local loss values (Cho et al., 2020; Goetz et al., 2019). Asynchronous participation has also been proposed to accommodate heterogeneous client speeds and availability (Yu et al., 2019b; Wang & Ji, 2022). More recently, cyclic client participation, where clients join in a fixed, repeating order, has gained attention for its practical and privacy advantages (Kairouz et al., 2021b). This structured scheduling addresses diverse availability challenges. In cross-device FL, clients may operate in different time zones or charge their devices at preferred times (Cho et al., 2023; Yang et al., 2018). In cross-silo FL, clients such as hospitals face planned constraints, including managing large local datasets, internal network security, or scheduled IT maintenance windows. Convergence guarantees for cyclic participation in ERM have been established by Cho et al. (2023) and later strengthened with variance reduction by Crawshaw & Liu (2024). However, all of these methods focus exclusively on ERM. Our

work bridges this gap by developing communication-efficient algorithms and providing rigorous convergence analyses for both federated AUC maximization under the realistic setting of cyclic client participation.

## 3    Preliminaries and Notations

We begin by establishing standard definitions and notations used throughout the paper. A function $f$ is said to be $C$-Lipschitz continuous if for all $\mathbf{x}, \mathbf{x}'$ in its domain, $|f(\mathbf{x}) - f(\mathbf{x}')| \leq C|\mathbf{x} - \mathbf{x}'|$. A differentiable function $f$ is $L$-smooth if its gradient is Lipschitz continuous, meaning $|\nabla f(\mathbf{x}) - \nabla f(\mathbf{x}')| \leq L|\mathbf{x} - \mathbf{x}'|$.

We consider a federated learning environment with $N$ clients, which are further divided into $K$ groups. The global datasets $\mathcal{D}^1$ (positive set) and $\mathcal{D}^2$ (negative set) are partitioned into $N$ non-overlapping subsets distributed across these clients, denoted as $\mathcal{D}_1 = \mathcal{D}_1^1 \cup \mathcal{D}_1^2 \cup \ldots \cup \mathcal{D}_1^N$ and $\mathcal{D}_2 = \mathcal{D}_2^1 \cup \mathcal{D}_2^2 \cup \ldots \cup \mathcal{D}_2^N$. Here, $\mathbb{E}_{\mathbf{z} \sim \mathcal{D}}$ denotes the empirical expectation over dataset $\mathcal{D}$.

We divide the $N$ clients into $K$ disjoint groups, each containing $N/K$ clients, denoted by $\mathcal{G}^k$ for $k \in [K]$. During training, the server cycles through these groups in a fixed, pre-determined order $(\mathcal{G}^1, \ldots, \mathcal{G}^K)$, implementing a cyclic participation pattern. In each communication round, $M$ clients are selected uniformly at random without replacement from the active group.

This framework is especially relevant for real-world federated learning applications. For instance, in cross-device FL, mobile devices can be grouped by regions or time zones, while random sampling within each group provides flexibility especially when the number of clients in a group is large. For simplicity, the framework can also be interpreted as a fully deterministic schedule in the special case of $K = N$ or $M = N/K$, where all clients in the active group participate in every round; in this case, the theoretical guarantees still hold.

The AUC for a scoring function $h : \mathcal{X} \to \mathbb{R}$ is defined as:

$$AUC(h) = \Pr(h(\mathbf{x}) \geq h(\mathbf{x}')|y = 1, y' = -1), \tag{2}$$

where $\mathbf{z} = (\mathbf{x}, y)$ and $\mathbf{z}' = (\mathbf{x}', y')$ are drawn independently from the data distribution $\mathbb{P}$.

**Notations.** Let $\mathbf{w} \in \mathbf{R}^d$ denote the model parameters. In the minimax formulation, we define $\mathbf{v} = (\mathbf{w}, a, b)$ as the primal variable, where $a, b \in \mathbf{R}$. For the stagewise algorithms, $\mathbf{v}_0^s$ and $\alpha_0^s$ denote the initialization at the beginning of stage $s$.

Within each stage, $\mathbf{v}_{m,t}^{e,k}$ denotes the variable at epoch $e$, client group $k$, client $m$, and iteration $t$. The notation $\mathbf{v}_m^{e,k}$ represents the final output of client $m$ in group $k$ at epoch $e$. The group-level aggregation is given by

$$\mathbf{v}^{e,k} = \frac{1}{|\mathcal{G}^{e,k}|} \sum_{m \in \mathcal{G}^{e,k}} \mathbf{v}_m^{e,k},$$

which is the average output of all participating clients in group $k$ during epoch $e$.

The same notation convention applies to $\mathbf{w}_{m,t}^{e,k}$ and $\alpha_{m,t}^{e,k}$. A complete list of symbols used in this paper is provided in Appendix A.

## 4    Minimax Optimization

To maximize AUC, we adopt the surrogate loss formulation in (1). Following Ying et al. (2016), we use the squared surrogate loss $\psi(a, b) = (1 - a + b)^2$, which reformulates the AUC maximization problem as the following minimax optimization:

$$\min_{\mathbf{w},a,b} \max_{\alpha} f(\mathbf{w}, a, b, \alpha) = \mathbb{E}_{\mathbf{z}}[F(\mathbf{w}, a, b, \alpha; \mathbf{z})], \tag{3}$$

where

$$\begin{aligned} F(\mathbf{w}, a, b, \alpha; \mathbf{z}) =& (1 - p)(h(\mathbf{w}; \mathbf{z}) - a)^2 \mathbb{I}_{[y=1]} + p(h(\mathbf{w}; \mathbf{z}) - b)^2 \mathbb{I}_{[y=-1]} \\ &+ 2(1 + \alpha)(ph(\mathbf{w}; \mathbf{z})\mathbb{I}_{[y=-1]} - (1 - p)h(\mathbf{w}; \mathbf{z})\mathbb{I}_{[y=1]}) - p(1 - p)\alpha^2. \end{aligned}$$

Let $\mathbf{v} := (\mathbf{w}, a, b)$ denote the set of primal variables. This reformulation enables a natural decomposition across clients, leading to the following federated optimization problem:

$$\min_{\substack{\mathbf{w} \in \mathbb{R}^d \\ (a,b) \in \mathbb{R}^2}} \max_{\alpha \in \mathbb{R}} f(\mathbf{w}, a, b, \alpha) = \frac{1}{K} \sum_{k=1}^{K} \frac{1}{|\mathcal{G}^k|} \sum_{m \in \mathcal{G}^k} f_{k,m}(\mathbf{w}, a, b, \alpha), \tag{4}$$

where $f_{k,m}(\mathbf{w}, a, b, \alpha) = \mathbb{E}_{\mathbf{z} \sim \mathbb{P}_{k,m}}[F(\mathbf{w}, a, b, \alpha; \mathbf{z}^k)]$ and $\mathbb{P}_{k,m}$ denotes the local data distribution on client $m$ of group $k$.

We now present our algorithm for federated AUC maximization under the practical constraint of cyclic client participation. At the $s$-th stage, we construct a strongly convex–strongly concave subproblem centered at the previous stage's output $\mathbf{v}_0^s$. The local objective for client $m$ in group $k$ is defined as:

$$\min_{\mathbf{v}} \max_{\alpha} f_{k,m}(\mathbf{v}, \alpha) + \frac{\gamma}{2} |\mathbf{v} - \mathbf{v}_0^s|^2. \tag{5}$$

While our approach builds on the stagewise framework of Guo et al. (2020); Yuan et al. (2021a), the key novelty lies in the intra-stage optimization: we introduce multiple cycle epochs and provide a theoretical analysis that handles deterministic client ordering—an aspect that poses significant technical challenges. The detailed procedure for a single stage is outlined in Algorithm 1, and the overarching multi-stage framework is presented in Algorithm 2. Each stage initializes the primal and dual variables using the outputs from the previous stage. Within a stage, the algorithm runs for multiple epochs, during which all client groups are visited multiple times. Each client group samples a subset of clients to participate, performs $I$ local update steps, and then passes the updated primal and dual variables to the next client group. The primal variables are updated via stochastic gradient descent, while the dual variable is updated via stochastic gradient ascent. After completing a stage, the step sizes and number of epochs are adjusted before proceeding to the next stage.

We define the following key quantities for our analysis:

$$
\begin{aligned}
&\phi(\mathbf{v}) = \max_{\alpha} f(\mathbf{v}, \alpha), \quad \phi_s(\mathbf{v}) = \phi(\mathbf{v}) + \frac{\gamma}{2}\|\mathbf{v} - \mathbf{v}_{s-1}\|^2, f^s(\mathbf{v}, \alpha) = f(\mathbf{v}, \alpha) + \frac{\gamma}{2}\|\mathbf{v} - \mathbf{v}_{s-1}\|^2, \\
&F_{k,m}^s(\mathbf{v}, \alpha; \mathbf{z}_{k,m}) = F(\mathbf{v}, \alpha; \mathbf{z}_{k,m}) + \frac{\gamma}{2}\|\mathbf{v} - \mathbf{v}_{s-1}\|^2, \mathbf{v}_\phi^* = \arg\min_{\mathbf{v}} \phi(\mathbf{v}), \quad \mathbf{v}_{\phi_s}^* = \arg\min_{\mathbf{v}} \phi_s(\mathbf{v}).
\end{aligned}
\tag{6}
$$

Our convergence analysis is based on the following standard assumptions:

**Assumption 4.1.** *(i) Initialization: There exist $\mathbf{v}_0, \Delta_0 > 0$ such that $\phi(\mathbf{v}_0) - \phi(\mathbf{v}_\phi^*) \le \Delta_0$.*
*(ii) PL condition: $\phi(\mathbf{v})$ satisfies the $\mu$-PL condition, i.e., $\mu(\phi(\mathbf{v}) - \phi(\mathbf{v}_*)) \le \frac{1}{2}\|\nabla\phi(\mathbf{v})\|^2$; (iii) Smoothness: For any $\mathbf{z}$, $f(\mathbf{v}, \alpha; \mathbf{z})$ is $\ell$-smooth in $\mathbf{v}$ and $\alpha$. $\phi(\mathbf{v})$ is $L$-smooth, i.e., $\|\nabla\phi(\mathbf{v}_1) - \nabla\phi(\mathbf{v}_2)\| \le L\|\mathbf{v}_1 - \mathbf{v}_2\|$;(iv) Bound gradients: $\|\nabla_{\mathbf{v}} f_{k,m}^s(\mathbf{v}, \alpha; \mathbf{z})\|^2 \le G^2$; (v) Bounded variance:*

$$
\begin{aligned}
\mathbb{E}[\|\nabla_{\mathbf{v}} f_{k,m}(\mathbf{v}, \alpha) - \nabla_{\mathbf{v}} F_{k,m}(\mathbf{v}, \alpha; \mathbf{z})\|^2] &\le \sigma^2, \\
\mathbb{E}[|\nabla_\alpha f_{k,m}(\mathbf{v}, \alpha) - \nabla_\alpha F_{k,m}(\mathbf{v}, \alpha; \mathbf{z})|^2] &\le \sigma^2.
\end{aligned}
\tag{7}
$$

**Remark.** These assumptions are consistent with prior literature (Guo et al., 2020; Yuan et al., 2021a). The PL condition for minimax AUC maximization has been theoretically and empirically verified (Guo et al., 2023b). Since $f(\mathbf{v}, \alpha; \mathbf{z})$ is $\ell$-smooth in $\mathbf{v}$, it is also $\ell$-weakly convex in $\mathbf{v}$. Accordingly, in the subsequent analysis we choose $\gamma = 2\ell$, which guarantees that the subproblem (5) in each stage becomes $\ell$-strongly convex in $\mathbf{v}$.

Our first step is to establish a bound on the suboptimality gap within a stage (Lemma 4.2).

---

**Algorithm 1** One Stage Federated Minimax (OSFM)

---

**On Server:**
Initialization: $\mathbf{v}^{1,0}, \alpha^{1,0}$
**for** $e \in [E]$ cycle-epochs **do**
    **for** $k \in [K]$ **do**
        Sample $M$ clients from $k$-th client set uniformly at random w/o replacement to get client set $\mathcal{G}^{e,k}$
        Send global model $\mathbf{v}^{e,k-1}$ to clients in $\mathcal{G}^{e,k}$
        Each client $m \in \mathcal{G}^{e,k}$ in parallel do:
            $\mathbf{v}_m^{e,k}, \alpha_m^{e,k} \leftarrow LocalUpdate(\mathbf{v}^{e,k-1}, \alpha^{e,k-1})$
    $\mathbf{v}^{e,k} = \frac{1}{|\mathcal{G}^{e,k}|} \sum\limits_{m \in \mathcal{G}^{e,k}} \mathbf{v}_m^{e,k}$
    $\alpha^{e,k} = \frac{1}{|\mathcal{G}^{e,k}|} \sum\limits_{m \in \mathcal{G}^{e,k}} \alpha_m^{e,k}$
    **end for**
    $\mathbf{v}^{e+1,0} = \mathbf{v}^{e,K}$, $\alpha^{e+1,0} = \alpha^{(e,K)}$
**end for**
Output: $\frac{1}{E} \sum_e \frac{1}{K} \sum_k \mathbf{v}^{e,k}$, $\frac{1}{E} \sum_e \frac{1}{K} \sum_k \alpha^{e,k}$

---

$LocalUpdate(\mathbf{v}_0, \alpha_0)$:
**for** $t \in [I]$ **do**
    Sample mini-batch $\mathbf{z}_{m,t}^{e,k}$ from local dataset
    Update $\mathbf{v}_{m,t}^{e,k} = \mathbf{v}_{m,t-1}^{e,k} - \eta \nabla_{\mathbf{v}} f(\mathbf{v}_{m,t}^{e,k}, \alpha_{m,t}^{e,k}; \mathbf{z}_{m,t}^{e,k})$
    Update $\alpha_{m,t}^{e,k} = \alpha_{m,t-1}^{e,k} + \eta \nabla_{\alpha} f(\mathbf{v}_{m,t}^{e,k}, \alpha_{m,t}^{e,k}; \mathbf{z}_{m,t}^{e,k})$
**end for**
Return $\mathbf{v}_I, \alpha_I$

---

**Algorithm 2** Federated Minimax with Cyclic Client Participation (CyCp-Minimax)

---

Initialization: Primal variable $\mathbf{x}^{1,0}$, Client Groups $\delta(k)$, $k \in [K]$
**for** $s \in [S]$ **do**
    $\mathbf{v}_s, \alpha_s = \text{OSFM}(\mathbf{v}_{s-1}, \alpha_{s-1}, \eta_s, E_s)$
**end for**

---

**Lemma 4.2.** *Suppose Assumption 4.1 holds and by running Algorithm 1 for one stage with output denoted by $(\bar{\mathbf{v}}, \bar{\alpha})$, we have*

$$f^s(\bar{\mathbf{v}}, \alpha) - f^s(\mathbf{v}, \bar{\alpha}) \leq \frac{1}{E} \sum_{e=0}^{E-1} [f^s(\mathbf{v}^{e+1,0}, \alpha) - f^s(\mathbf{v}, \alpha^{e+1,0})]$$

$$\leq \frac{1}{E} \sum_{e=0}^{E-1} \bigg[ \underbrace{\langle \partial_{\mathbf{v}} f^s(\mathbf{v}^{e,0}, \alpha^{e,0}), \mathbf{v}^{e+1,0} - \mathbf{v} \rangle}_{A_1} + \underbrace{\langle \partial_{\alpha} f^s(\mathbf{v}^{e,0}, \alpha^{e,0}), \alpha - \alpha^{e+1,0} \rangle}_{A_2}$$

$$+ \underbrace{\frac{3\ell + 3\ell^2/\mu_2}{2} \|\mathbf{v}^{e+1,0} - \mathbf{v}^{e,0}\|^2 + 2\ell(\alpha^{e+1,0} - \alpha^{e,0})^2}_{A_3} - \frac{\ell}{3} \|\mathbf{v} - \mathbf{v}^{e,0}\|^2 - \frac{\mu_2}{3}(\alpha^{e,0} - \alpha)^2 \bigg].$$

The resulting bound contains terms like $A_1$ and $A_2$ that couple the randomness from different client groups. To decouple these dependencies, we introduce novel *virtual sequences* $\hat{\mathbf{v}}^{e,0}$, $\tilde{\mathbf{v}}^{e,0}$, $\hat{\alpha}^{e,0}$, and $\tilde{\alpha}^{e,0}$ (Lemmas 4.3 and B.4). These sequences are purely conceptual and do not require any actual computation. Different from (Guo et al., 2020; Yuan et al., 2021a), these virtual sequences further depend on virtual estimates of gradient e.g, evaluating gradient using current data and old model. These sequences are carefully designed to isolate the "signal" (the true gradient) from the "noise" (the stochastic gradient error) and to manage the bias introduced by the cyclic schedule.

**Lemma 4.3.** *Define $\hat{\mathbf{v}}^{e+1,0} = \mathbf{v}^{e,0} - \eta \sum_{k=1}^{K} \frac{1}{M} \sum_{m \in \mathcal{G}^{e,k}} \sum_{t=1}^{I} \nabla_{\mathbf{v}} f_{k,m}^s(\mathbf{v}^{e,0}, \alpha^{e,0})$ and*

$$\tilde{\mathbf{v}}^{e+1,0} = \tilde{\mathbf{v}}^{e,0} - \frac{\eta}{M} \sum_{k=1}^{K} \sum_{m \in \mathcal{G}^{e,k}} \sum_{t=1}^{I} \left( \nabla_{\mathbf{v}} F_k^s(\mathbf{v}^{e,0}, \alpha^{e,0}; z_{m,t}^{e,k}) - \nabla_{\mathbf{v}} f_k^s(\mathbf{v}^{e,0}, \alpha^{e,0}) \right), \; \text{for } t > 0; \; \tilde{\mathbf{v}}_0 = \mathbf{v}_0. \quad (8)$$

*then we have*

$$\mathbb{E}_{e,0}[A_1] \leq \frac{6\tilde{\eta}K\sigma^2}{MKI} + \frac{6\ell}{KMI} \sum_k \sum_{m \in \mathcal{G}^{e,k}} \sum_t \mathbb{E}(\|\mathbf{v}^{e,0} - \mathbf{v}_{m,t}^{e,k}\|^2 + \|\alpha^{e,0} - \alpha_{m,t}^{e,k}\|^2) + \frac{\ell}{3} \|\mathbf{v}^{e+1,0} - \mathbf{v}\|^2$$

$$+ \frac{1}{2\eta KI} \mathbb{E}(\|\mathbf{v}^{e,0} - \mathbf{v}\|^2 - \|\mathbf{v}^{e,0} - \mathbf{v}^{e+1,0}\|^2 - \|\mathbf{v}^{e+1,0} - \mathbf{v}\|^2) + \frac{1}{2\eta KI}(\|\mathbf{v} - \tilde{\mathbf{v}}^{e,0}\|^2 - \|\mathbf{v} - \tilde{\mathbf{v}}^{e+1,0}\|^2),$$

*where $\mathbf{E}_{e,0}$ denotes expectation with respect to all randomness realized prior to epoch $e$.*

With the help of the two virtual sequences $\hat{\mathbf{v}}^{e,0}$ and $\tilde{\mathbf{v}}^{e,0}$, we have successfully disentangled the interdependency between different clients and covert the bounds into standard variance bound plus model drift, i.e., the second term on the RHS.

Putting things together, we have the convergence analysis for one stage of the Algorithm as

**Lemma 4.4.** *Suppose Assumption 4.1 holds. Running one stage of Algorithm 1 ensures that*

$$\mathbb{E}[f^s(\bar{\mathbf{v}}, \alpha) - f^s(\mathbf{v}, \bar{\alpha})] \leq \frac{1}{4\eta EKI} \|\mathbf{v}_0 - \mathbf{v}\|^2 + \frac{1}{4\eta EKI} \|\alpha_0 - \alpha\|^2 + \left( \frac{3\ell^2}{2\mu_2} + \frac{3\ell}{2} \right) 36\eta^2 I^2 K^2 G^2 + \frac{3\eta\sigma^2}{M},$$

**Remark.** Lemma above shows that the bound for the output of a stage (Algorithm 1) depends on the quality of its inputs, $\mathbf{v}_0$ and $\alpha_0$, which are the outputs of th e previous stage. By employing a stagewise algorithm (Algorithm 2) and setting parameters appropriately, we can ensure that the duality gap decreases exponentially across stages. Extending the one-stage result to the full double-loop (Algorithm 2) procedure yields the following theorem.

**Theorem 4.5.** *Define $\hat{L} = L + 2\ell, c = \frac{\mu/\hat{L}}{5+\mu/\hat{L}}$. Set $\gamma = 2\ell$, $\eta_s = \eta_0 \exp(-(s-1)c)$, $T_s = \frac{212}{\eta_0 \min(\ell, \mu_2)} \exp((s-1)c)$. To return $\mathbf{v}_S$ such that $\mathbb{E}[\phi(\mathbf{v}_S) - \phi(\mathbf{v}_\phi^*)] \leq \epsilon$, it suffices to choose $S \geq O\left( \frac{5\hat{L}+\mu}{\mu} \max \left\{ \log\left(\frac{2\Delta_0}{\epsilon}\right), \log S + \log\left[\frac{2\eta_0}{\epsilon} \frac{12(\sigma^2)}{5K}\right] \right\} \right)$. The iteration complexity is $\tilde{O}\left(\frac{\hat{L}}{\mu^2 M \epsilon}\right)$ and the communication complexity is $\tilde{O}\left(\frac{K}{\mu^{3/2}\epsilon^{1/2}}\right)$ by setting $I_s = \Theta(\frac{1}{K\sqrt{M\eta_s}})$, where $\tilde{O}$ suppresses logarithmic factors.*

**Remark. Linear Speedup and Efficiency:** The iteration complexity exhibits linear speedup with respect to the number of simultaneous participating clients $M$, i.e., $\tilde{O}(1/(M\epsilon))$. This indicates that the proposed algorithm efficiently leverages parallelism even under cyclic participation. The need for a smaller communication interval (scaled by $K$) as the number of groups increases arises naturally from the cyclic participation protocol. A larger $K$ introduces longer delays between consecutive client updates, heightening staleness and drift. Adapting the local update count $I_s$ based on $K$ effectively balances communication efficiency against the resulting error.

**Comparison to Prior Work:** Our communication complexity matches the dependency on $\mu$ and $\epsilon$ achieved by Guo et al. (2020); Yuan et al. (2021a) under the more idealistic assumption of random client sampling. Thus, our analysis demonstrates that cyclic participation does not degrade the asymptotic convergence rate. To our knowledge, this is the first such guarantee for federated minimax optimization.

# 5 Pairwise Objective

In this section, we study a broader class of federated pairwise optimization problems, which generalizes AUC maximization to arbitrary pairwise loss functions. We propose a novel algorithm and provide convergence

analysis for this setting under cyclic client participation, both with and without assuming the PL condition. We consider the following federated pairwise objective:

$$\min_{\mathbf{w}\in\mathbb{R}^d} F(\mathbf{w}) = \frac{1}{N}\sum_{i=1}^N \mathbb{E}_{\mathbf{z}\in\mathcal{D}_1^i} \frac{1}{N}\sum_{j=1}^N \mathbb{E}_{\mathbf{z}'\in\mathcal{D}_2^j} \psi(h(\mathbf{w};\mathbf{z}), h(\mathbf{w};\mathbf{z}')). \tag{9}$$

A major challenge in optimizing equation 9 is that its gradient inherently couples data across all clients. Let $\nabla_1\psi(\cdot,\cdot)$ and $\nabla_2\psi(\cdot,\cdot)$ denote the partial derivatives of $\psi$ with respect to its first and second arguments, respectively. We decompose the global gradient as:

$$\nabla F(\mathbf{w}) = \frac{1}{N}\sum_{i=1}^N \mathbb{E}_{\mathbf{z}\in\mathcal{D}_1^i} \underbrace{\frac{1}{N}\sum_{j=1}^N \mathbb{E}_{\mathbf{z}'\in\mathcal{D}_2^j} \nabla_1\psi(h(\mathbf{w};\mathbf{z}), h(\mathbf{w};\mathbf{z}'))\nabla h(\mathbf{w};\mathbf{z})}_{\Delta_{i1}}$$

$$+ \frac{1}{N}\sum_{i=1}^N \mathbb{E}_{\mathbf{z}'\in\mathcal{D}_2^i} \underbrace{\frac{1}{N}\sum_{j=1}^N \mathbb{E}_{\mathbf{z}\in\mathcal{D}_1^j} \nabla_2\psi(h(\mathbf{w};\mathbf{z}), h(\mathbf{w};\mathbf{z}'))\nabla h(\mathbf{w};\mathbf{z}')}_{\Delta_{i2}}.$$

Defining the local gradient component as $\nabla F_i(\mathbf{w}) := \Delta_{i1} + \Delta_{i2}$, the global gradient can be expressed compactly as $\nabla F(\mathbf{w}) = \frac{1}{N}\sum_{i=1}^N \nabla F_i(\mathbf{w})$.

The difficulty in computing $\Delta_{i1}$ and $\Delta_{i2}$ lies in its dependence on global information—specifically, the expectation over all clients' datasets $\mathcal{D}_2^j$ and $\mathcal{D}_1^j$:

$$\Delta_{i1} = \mathbb{E}_{\mathbf{z}\in\mathcal{D}_1^i} \frac{1}{N}\sum_{j=1}^N \mathbb{E}_{\mathbf{z}'\in\mathcal{D}_2^j} \nabla_1\psi(\underbrace{h(\mathbf{w};\mathbf{z})}_{\text{local}}, \underbrace{h(\mathbf{w};\mathbf{z}')}_{\text{global}}) \underbrace{\nabla h(\mathbf{w};\mathbf{z})}_{\text{local}}, \tag{10}$$

which local components can be computed by each client using its own data but global components depend on data from all clients.

To address this challenge while maintaining the cyclic participation constraint, we adopt an *active–passive* strategy. To estimate $\Delta_{i,1}$, each client $i$ computes the active parts using its own data while the passive parts are contributed by other clients through previously shared global information. Specifically, for the $k$-th group in the $e$-th epoch, we estimate the global term by sampling $h_{2,\xi}^{e-1} \in \mathcal{H}_2^{e-1}$ without replacement and compute an estimator of $\Delta_{i1}$ by

$$G_{i,t,1}^{e,k} = \nabla_1\psi(\underbrace{h(\mathbf{w}_{m,t}^{e,k};\mathbf{z}_{m,t,1}^{e,k})}_{\text{active}}, \underbrace{h_{2,\xi}^{e-1}}_{\text{passive}}) \underbrace{\nabla h(\mathbf{w}_{m,t}^{e,k};\mathbf{z}_{m,t,1}^{e,k})}_{\text{active}}, \tag{11}$$

and similarly $\Delta_{i2}$ by

$$G_{m,t,2}^{e,k} = \nabla_1\psi(\underbrace{h_{1,\xi}^{e-1}}_{\text{passive}}, \underbrace{h(\mathbf{w}_{m,t}^{e,k};\mathbf{z}_{m,t,2}^{e,k})}_{\text{active}},) \underbrace{\nabla h(\mathbf{w}_{m,t}^{e,k};\mathbf{z}_{m,t,2}^{e,k})}_{\text{active}}, \tag{12}$$

where local components in $G_{i,t,1}^{e,k}$ are referred as active parts, while global components are referred as passive parts. The passive parts $h_{2,\xi}^{e-1}$ and $h_{1,\xi}^{e-1}$ are constructed by sampling scores from the *previous epoch* $(e-1)$. And $\xi = (\tilde{k}, \tilde{m}, \tilde{t}, \mathbf{z}_{\tilde{m},\tilde{t},2}^{e-1,\tilde{k}})$ represents a random variable that captures the randomness in the sampled group $\tilde{k} \in \{1,\ldots,K\}$, sampled client $\tilde{m} \in \{1,\ldots,\mathcal{G}^{\tilde{k}}\}$, iteration index $\tilde{t} \in \{1,\ldots,I\}$, data sample $\mathbf{z}_{\tilde{m},\tilde{t},1}^{e-1,\tilde{k}} \in \mathcal{D}_1^j$ and $\mathbf{z}_{\tilde{m},\tilde{t},2}^{e-1,\tilde{k}} \in \mathcal{D}_2^j$ for estimating the global component in (10). This delayed estimation bring two challenges: a latency error of using old estimates and interdependence between randomness of different epochs and clients.

Our algorithm design and analysis differ from Guo et al. (2023a) in two key aspects: (1) the passive components are from the previous *epoch* rather than the previous *round*, introducing greater latency but necessary

for cyclic analysis; (2) the deterministic cyclic order creates more complex interdependency between clients than random sampling. Unlike our minimax analysis where we could trace back to the start of the current cycle, here we must reference the previous cycle's initial state ($\mathbf{w}^{e-1,0}$) to establish independence, which increases the analytical complexity. A single loop algorithm is shown in Algorithm 3, and a double loop algorithm for leveraging PL condition is shown in 4. The One-Stage Federated Pairwise (Algorithm 3: OSFP) algorithm executes a single stage of federated optimization by iterating over multiple epochs. Within each epoch, it sequentially processes each client group. For every group, the server samples a subset of clients and broadcasts the current global model along with reference prediction sets. Each selected client then performs a *LocalUpdate* procedure, which maintains buffers of past predictions, samples new data points, computes pairwise losses and their gradients, and updates the local model using stochastic gradient steps. After completing local computations, clients return the updated models and prediction sets to the server, which aggregates them to update the global model. Once all epochs are completed, OSFP outputs the averaged global model. Furthermore, if the PL condition is satisfied, an outer loop (Algorithm 4) repeatedly calls OSFP over multiple stages, adjusting learning rates and epoch schedules to progressively refine the model.

Our analysis relies on the following standard assumptions for pairwise optimization in problem (9).

**Assumption 5.1.** *(i)* $\psi(\cdot)$ *is differentiable,* $L_\psi$*-smooth and* $C_\psi$*-Lipschitz.* *(ii)* $h(\cdot; \mathbf{z})$ *is differentiable,* $L_h$*-smooth and* $C_h$*-Lipschitz on* $\mathbf{w}$ *for any* $\mathbf{z} \in \mathcal{S}_1 \cup \mathcal{S}_2$. *(iii)* $\mathbb{E}_{\mathbf{z} \in \mathcal{S}_1^i} \mathbb{E}_{j \in [1:N]} \mathbb{E}_{\mathbf{z}' \in \mathcal{S}_2^j} \| \nabla_1 \psi(h(\mathbf{w}; \mathbf{z}), h(\mathbf{w}; \mathbf{z}')) \nabla h(\mathbf{w}; \mathbf{z}) + \nabla_2 \psi(h(\mathbf{w}; \mathbf{z}), h(\mathbf{w}; \mathbf{z}')) \nabla h(\mathbf{w}; \mathbf{z}') - \nabla F_i(\mathbf{w}) \|^2 \leq \sigma^2$. *(iv)* $\exists D$ *such that* $\| \nabla F_i(\mathbf{w}) - \nabla F(\mathbf{w}) \|^2 \leq D^2, \forall i$.

**Theorem 5.2.** *Suppose Assumption 5.1 holds. Running Algorithm 3 ensures that*

$$\frac{1}{E} \sum_{e=1}^{E} \mathbb{E} \| \nabla F(\mathbf{w}^{e-1}) \|^2 \leq O\left( \frac{F(\mathbf{w}^{e,0}) - F(\mathbf{w}_*)}{\eta I K E} + \eta^2 I^2 K^2 D^2 + 24\eta \frac{\sigma^2}{M} \right). \tag{13}$$

**Remark.** Since $\tilde{\eta} = \eta I$. By setting $\eta = \epsilon^2 M$, $I = \frac{1}{MK\epsilon}$, $E = \frac{1}{\epsilon^3}$, the iteration complexity is $EKI = O(\frac{1}{M\epsilon^4})$, which demonstrates a linear-speedup by $M$, and communication cost is $EK = O(\frac{K}{\epsilon^3})$. These results matches the results in (Guo et al., 2023a) of full client participation setting. This demonstrates that our algorithm successfully handles the additional challenges of cyclic participation without sacrificing asymptotic efficiency. It is important to note that the prediction scores $(\mathcal{H}_1^e, \mathcal{H}_2^e)$ incur no additional computation, as they simply reuse the scores generated during local updates. Since $(\mathcal{H}_1^e, \mathcal{H}_2^e)$ stores only the prediction scores from the previous communication round, the required memory is $O(IMK)$, where $I$ is the communication interval, $K$ is the number of client groups, and $M$ is the number of simultaneously participating clients per group. This storage cost is negligible compared with the number of parameters in a modern neural network and can be adjusted in practice by tuning $M$ or $I$. Each client's local buffers, $\mathcal{B}_{i,1}$ and $\mathcal{B}_{i,2}$, are of size $O(I)$, which stores sufficient historical predictions used to construct the loss function at every local update iteration. Shuffling these buffers is essential to ensure that, over time, each client has the opportunity to interact with every other client.

Under the additional assumption that $F(\mathbf{w})$ satisfies the $\mu$-PL condition, which is commonly used in pairwise optimization (Yang et al., 2021c), we obtain significantly improved convergence rates as follows.

**Theorem 5.3.** *Suppose Assumption 5.1 holds and* $F(\cdot)$ *satisfies a* $\mu$*-PL condition. To achieve an* $\epsilon$*-stationary point by running Algorithm 4, the iteration complexity is* $\widetilde{O}\left( \frac{1}{\mu^2 M \epsilon} \right)$ *and the communication complexity is* $\widetilde{O}\left( \frac{K}{\mu^{3/2} \epsilon^{1/2}} \right)$ *by setting* $I_s = \Theta(\frac{1}{\sqrt{K \eta_s}})$, *where* $\widetilde{O}$ *suppresses logarithmic factors.*

**Remark. Linear Speedup:** Both Theorem 5.2 and 5.3 confirm linear speedup with respect to the number of simultaneously participating clients $M$, demonstrating efficient use of parallel resources even under cyclic participation.

**Impact of PL Condition:** The PL condition dramatically reduce the iteration complexity from $O(1/\epsilon^4)$ to $\widetilde{O}(1/\epsilon)$ and the communication complexity from $O(1/\epsilon^3)$ to $\widetilde{O}(1/\mu^{3/2}\epsilon^{1/2})$.

---

**Algorithm 3** One Stage Federated Pairwise (OSFP)

---

1: On Server
2: Initialize $\mathcal{H}_1^1, \mathcal{H}_2^1$ by one initial epoch
3: **for** $e \in [E]$ **do**
4:    $\mathcal{H}_1^{e+1}, \mathcal{H}_2^{e+1} = \emptyset$
5:    **for** $k \in [K]$ **do**
6:       Sample $M$ clients from $k$-th client set uniformly at random w/o replacement to get client set $\mathcal{G}^{e,k}$
7:       Send global model $\mathbf{w}^{e,k-1}$ to clients in $\mathcal{S}^{e,k}$
8:       Clients $\mathcal{S}^{e,k}$ in parallel do:
9:          Sample without replacement $\mathcal{R}_{i,1}^{e,k}, \mathcal{R}_{i,2}^{e,k}$ from $\mathcal{H}_1^e, \mathcal{H}_2^e$, respectively.
10:          Send $\mathcal{R}_{i,1}^{e,k}, \mathcal{R}_{i,2}^{e,k}$ to client $i$ for all $i \in [N]$
11:          $\mathbf{w}_m^{e,k}, \mathcal{H}_{m,1}^{e,k}, \mathcal{H}_{m,2}^{e,k} \leftarrow LocalUpdate(\mathbf{w}^{e,k-1})$
12:       Add $\cup_{m \in \mathcal{G}^{e,k}} \mathcal{H}_{m,1}^{e,k}$ to $\mathcal{H}_1^{e+1}$ and $\cup_{m \in \mathcal{G}^{e,k}} \mathcal{H}_{m,2}^{e,k}$ to $\mathcal{H}_2^{e+1}$
13:       Compute $\mathbf{w}^{e,k} = \frac{1}{|\mathcal{G}^{e,k}|} \sum_{m \in \mathcal{G}^{e,k}} \mathbf{w}_m^{e,k}$.
14:    **end for**
15: **end for**
16: Return $\frac{1}{E} \sum_e \frac{1}{K} \sum_k \mathbf{w}_{m,K}^{e,k}$

---

17: On Client *LocalUpdate*
18: On Client $i$: **Require** parameters $\eta, K$
19: Initialize model $\mathbf{w}_{i,K}^0$ and initialize Buffer $\mathcal{B}_{i,1}, \mathcal{B}_{i,2} = \emptyset$
20: Receives $\bar{\mathbf{w}}^{e,k-1}$ from the server and set $\mathbf{w}_{i,0}^{e,k} = \bar{\mathbf{w}}^{e,k-1}$
21: Receive $\mathcal{R}_{i,1}^{e,k}, \mathcal{R}_{i,2}^{e,k}$ from the server
22: Shuffle $\mathcal{R}_{i,1}^{e,k}, \mathcal{R}_{i,2}^{e,k}$ and place the results into buffers $\mathcal{B}_{i,1}, \mathcal{B}_{i,2}$
23: Sample $K$ points from $\mathcal{D}_1^i$, compute their predictions using model $\mathbf{w}_{i,K}^0$ denoted by $\mathcal{H}_{i,1}^0$
24: Sample $K$ points from $\mathcal{D}_2^i$, compute their predictions using model $\mathbf{w}_{i,K}^0$ denoted by $\mathcal{H}_{i,2}^0$
25: **for** $k = 0, .., K-1$ **do**
26:    Sample $\mathbf{z}_{m,t,1}^{e,k}$ from $\mathcal{D}_1^m$, sample $\mathbf{z}_{m,t,2}^{e,k}$ from $\mathcal{D}_2^m$         $\diamond$ or sample two mini-batches of data
27:    Take next $h_\xi^{e-1}$ and $h_\zeta^{e-1}$ from $\mathcal{B}_{m,1}$ and $\mathcal{B}_{m,2}$, resp.
28:    Compute $h(\mathbf{w}_{m,t}^{e,k}; \mathbf{z}_{m,t,1}^{e,k})$ and $h(\mathbf{w}_{m,t}^{e,k}; \mathbf{z}_{m,t,2}^{e,k})$
29:    Add $h(\mathbf{w}_{m,k}^{e,k}; \mathbf{z}_{m,t,1}^{e,k})$ into $\mathcal{H}_{m,1}^{e,k}$ and add $h(\mathbf{w}_{m,t}^{e,k}; \mathbf{z}_{m,t,2}^{e,k})$ into $\mathcal{H}_{m,2}^{e,k}$
30:    Compute $G_{m,t,1}^{e,k}$ and $G_{m,t,2}^{e,k}$ according to (11) and (12)
31:    $\mathbf{w}_{m,t+1}^{e,k} = \mathbf{w}_{m,t}^{e,k} - \eta(G_{m,t,1}^{e,k} + G_{m,t,2}^{e,k})$
32: **end for**
33: Sends $\mathbf{w}_{m,t}^{e,k}$ to the server
34: Sends $\mathcal{H}_{m,1}^{e,k}, \mathcal{H}_{m,2}^{e,k}$ to the server

---

**Algorithm 4** Federated Pairwise AUC Under PL Condition (CyCP-Pairwise)

---

Initialization: $\mathbf{w}_0$
**for** $s \in [S]$ **do**
   $\mathbf{w}_s, \alpha_s = \text{OSFP}(\mathbf{w}_{s-1}, \eta_s, E_s)$
**end for**
Return $\mathbf{w}_S, \alpha_S$

---

**General Impact:** The problem we consider is general enough to extend beyond AUC maximization to applications such as bipartite ranking, metric learning, and other client-coupled tasks. For instance, consider the contrastive loss commonly used in metric learning (Hadsell et al., 2006):

$$\psi(h(\mathbf{w};\mathbf{z}_1), h(\mathbf{w};\mathbf{z}_2)) = \frac{1}{2}\,\delta(\mathbf{z}_1,\mathbf{z}_2)\,\|h(\mathbf{w};\mathbf{z}_1)-h(\mathbf{w};\mathbf{z}_2)\|^2 + \frac{1}{2}\,(1-\delta(\mathbf{z}_1,\mathbf{z}_2))\,\big(\max(0, m-\|h(\mathbf{w};\mathbf{z}_1)-h(\mathbf{w};\mathbf{z}_2)\|)\big)^2,$$

$$(14)$$

where $\delta(\mathbf{z}_1,\mathbf{z}_2) = 1$ if $\mathbf{z}_1$ and $\mathbf{z}_2$ are similar, and 0 otherwise, and $h(\mathbf{w};\mathbf{z})$ denotes the embedding of data point $\mathbf{z}$ produced by model $\mathbf{w}$. This loss naturally fits into the formulation in Problem 9 when the data pairs are distributed across multiple clients.

**Privacy Considerations.** The two algorithmic frameworks proposed in this work require clients to share model parameters—and, for pairwise objectives, to additionally share prediction scores—to enable collaborative optimization, consistent with prior literature (Guo et al., 2023a; McMahan et al., 2017). However, such information exchange can introduce privacy risks, as model updates and related signals may potentially leak information about individual data points (Zhu et al., 2019). These risks can be mitigated through several techniques, including: 1) adding noise to ensure differential privacy (Abadi et al., 2016; McMahan et al., 2018; Truex et al., 2020; Wei et al., 2020); 2) quantization (Kang et al., 2024; Youn et al., 2023; Xu et al., 2025); 3) dropout (Jain et al., 2015); and 4) homomorphic encryption during aggregation (Jin et al., 2023; Fang & Qian, 2021).

# 6 Experiments

We evaluate our proposed method on four distinct datasets: CIFAR-10, CIFAR-100, ChestMNIST, and an Insurance fraud dataset to demonstrate the robustness of the proposed algorithms across domains (computer vision, medical imaging, and tabular data) under challenging non-IID and imbalanced conditions.

For CIFAR-10 and CIFAR-100 (Krizhevsky et al., 2009), we reformulate the multi-class tasks as binary classification problems by designating one class as positive and the rest as negative. To induce class imbalance, 95% of positive samples in CIFAR-10 and 50% in CIFAR-100 are removed. ChestMNIST (Yang et al., 2021b) is framed as a binary task predicting the presence of a mass. Training data is distributed across 100 clients using a Dirichlet distribution with concentration parameter *dir* to simulate heterogeneity. ResNet18 (He et al., 2016) is used for Cifar-10/100 and DenseNet121 Huang et al. (2017) is used for ChestMNIST.

The Insurance fraud dataset is constructed from Medicare Part B claims (2017–2022) from the Centers for Medicare & Medicaid Services (CMS) (Centers for Medicare & Medicaid Services (CMS), 2016). Providers are labeled as fraudulent based on the List of Excluded Individuals and Entities (LEIE) (Office of Inspector General, U.S. Department of Health & Human Services, 2016). Each U.S. state is treated as a separate client, resulting in 50 clients with naturally heterogeneous data. A chronological split is used: 2017–2020 for training, 2021 for validation, and 2022 for testing. The linear model is used for the insurance data.

We compare our methods with cyclic-participation baselines: CyCp-FedAVG (Cho et al., 2023), Amplified FedAVG (A-FedAVG) (Wang & Ji, 2022), and Amplified SCAFFOLD (A-SCAFFOLD) (Crawshaw & Liu, 2024). We also include Random Sampling Minimax (RS-Minimax) and Random Sampling PSM (RS-PSM). These two methods are adapted from Guo et al. (2020) and Guo et al. (2023a) and rely on randomly sampling clients in each round. Step sizes are tuned in [1e-1, 1e-2, 1e-3]. We use a surrogate loss $\psi(h1, h2) = \frac{1}{1+\exp((h1-h2)/\lambda)}$ where the scaling factor $\lambda$ is tuned in [1, 1e-1, 1e-2, 1e-3]. Regularization factor $\rho$ for minimax AUC is tuned in [1e-1, 1e-2, 1e-3]. All algorithms are trained for 100 epochs, with stagewise algorithms tuned for initial stage length [1,10,50], decay factor [0.1,0.2,0.5], and stage scaling [2,5,10]. The reported results in all tables are mean and std of three runs of an algorithm

We first set dir= 0.5 for CIFAR-10/100 and evaluate two settings: (1) *Original labels*, using naturally imbalanced labels; (2) *Flipped labels*, where 20% of positive labels are randomly flipped to negative. Test AUC results are shown in Tables 1 and 2. Our proposed methods (Minimax and PSM) consistently outperform all baselines across datasets and settings. For instance, on CIFAR-100, PSM improves AUC by 5–6% over the best baseline; on the Insurance dataset, PSM surpasses FedAVG by over 4%. Under label flips, Minimax and PSM also significantly outperform baselines. Our proposed methods, CyCP-Minimax and

CyCP-PSM, achieve superior performance to these random-sampling baselines in most cases since random sampling does not ensure comprehensive population coverage, which is essential for mitigating non-IID bias. We also emphasize that random sampling implicitly assumes that all sampled clients are available to participate, an assumption that often breaks down in real-world federated environments. In contrast, cyclic client participation overcomes these limitations by utilizing predictable participation.

**Sensitivity to Heterogeneity and Label Noise** We further examine the impact of heterogeneity (Dirichlet concentration $dir$) and label noise (flip ratio $flip$) on CIFAR-10, CIFAR-100, and ChestMNIST (Tables 3, 4 and 5). The trends are consistent: 1) As heterogeneity increases (smaller Dirichlet $\alpha$), baseline methods exhibit significant drops in AUC, while Minimax and PSM remain stable. 2) Under high label-flip rates ($flip$=0.2), our methods maintain competitive performance, sometimes exceeding the clean-label baselines (e.g., PSM on CIFAR-10). 3) These observations confirm that the proposed algorithms better align client objectives despite data inconsistency and label corruption, validating their theoretical design for robust federated AUC optimization.

**Ablation on Communication Efficiency** Figure 1 presents an ablation study on the communication interval $I$, which controls the number of local updates before synchronization. Both Minimax and PSM maintain stable test AUC even as $I$ increases, demonstrating that the proposed methods can tolerate infrequent communication without loss of convergence quality. This property makes them well-suited for bandwidth-constrained federated environments, where communication cost dominates training time.

Overall, the experimental results demonstrate that: 1) Performance: Minimax and PSM achieve consistent and significant AUC improvements over state-of-the-art baselines across diverse modalities. 2) Robustness: They remain resilient under severe class imbalance and label noise. 3) Scalability: The methods scale effectively to large heterogeneous client populations and tolerate sparse communication. Together, these results substantiate the practical advantages of our proposed algorithms for real-world federated learning scenarios involving non-IID, imbalanced, and noisy data distributions.

Table 1: Experimental Results on Original Labels. ($dir$=0.5 for Cifar-10/100)

|  | Cifar-10 | Cifar-100 | ChestMNIST | Insurance |
|---|---|---|---|---|
| CyCp-FedAVG | $0.7836 \pm 0.0020$ | $0.8894 \pm 0.0012$ | $0.6012 \pm 0.0016$ | $0.7339 \pm 0.0035$ |
| A-FedAVG | $0.7950 \pm 0.0014$ | $0.9137 \pm 0.0028$ | $0.5994 \pm 0.0025$ | $0.7384 \pm 0.0030$ |
| A-SCAFFOLD | $0.7993 \pm 0.0018$ | $0.9105 \pm 0.0017$ | $0.6015 \pm 0.0022$ | $0.7412 \pm 0.0021$ |
| RS-Minimax | $0.8153 \pm 0.0025$ | $0.9392 \pm 0.0029$ | $0.5697 \pm 0.0031$ | $0.7419 \pm 0.0013$ |
| RS-Pairwise | $0.8357 \pm 0.0012$ | $\mathbf{0.9688 \pm 0.0010}$ | $0.6118 \pm 0.0020$ | $\mathbf{0.7619 \pm 0.0011}$ |
| CyCp-Minimax | $\mathbf{0.8446 \pm 0.0015}$ | $0.9318 \pm 0.0014$ | $\mathbf{0.6163 \pm 0.0026}$ | $0.7616 \pm 0.0017$ |
| CyCp-Pairwise | $\mathbf{0.8480 \pm 0.0008}$ | $\mathbf{0.9694 \pm 0.0013}$ | $\mathbf{0.6256 \pm 0.0015}$ | $\mathbf{0.7778 \pm 0.0002}$ |

Table 2: Experimental Results on Flipped Labels ($dir$=0.5 for Cifar-10/100, $flip$=20%)

|  | Cifar-10 | Cifar-100 | ChestMNIST | Insurance |
|---|---|---|---|---|
| CyCp-FedAVG | $0.7739 \pm 0.0031$ | $0.8781 \pm 0.0019$ | $0.5927 \pm 0.0008$ | $0.7292 \pm 0.0029$ |
| A-FedAVG | $0.7891 \pm 0.0020$ | $0.8951 \pm 0.0023$ | $0.5901 \pm 0.0013$ | $0.7298 \pm 0.0035$ |
| A-SCAFFOLD | $0.7950 \pm 0.0025$ | $0.9066 \pm 0.0024$ | $0.5987 \pm 0.0011$ | $0.7307 \pm 0.0036$ |
| RS-Minimax | $0.8170 \pm 0.0021$ | $0.9185 \pm 0.0019$ | $0.5587 \pm 0.0026$ | $\mathbf{0.7517 \pm 0.0038}$ |
| RS-Pairwise | $0.8201 \pm 0.0017$ | $\mathbf{0.9480 \pm 0.0020}$ | $0.6004 \pm 0.0009$ | $0.7173 \pm 0.0043$ |
| Minimax | $\mathbf{0.8316 \pm 0.0006}$ | $0.9275 \pm 0.0015$ | $\mathbf{0.6150 \pm 0.0007}$ | $0.7505 \pm 0.0041$ |
| CyCp-Pairwise | $\mathbf{0.8424 \pm 0.0038}$ | $\mathbf{0.9455 \pm 0.0020}$ | $\mathbf{0.6077 \pm 0.0012}$ | $\mathbf{0.7722 \pm 0.0018}$ |

We show ablation experiments over the communication interval in Figure 1. We can see that the algorithms can tolerate a large number of local update steps $I$ without degrading the performance, which verifies the communication-efficiency of the proposed algorithms.

For more ablation experiments, please refer to Appendix F.

Table 3: Results with different Dirichlet parameter $dir$ and flip ratio $flip$ on Cifar-10

|  | $dir$=0.1,$flip$=0 | $dir$=0.1,$flip$=0.2 | $dir$=10,$flip$=0 | $dir$=10,$flip$=0.2 |
|---|---|---|---|---|
| CyCp-FedAVG | $0.7581 \pm 0.0023$ | $0.6800 \pm 0.0030$ | $0.8227 \pm 0.0025$ | $0.8199 \pm 0.0019$ |
| A-FedAVG | $0.7732 \pm 0.0029$ | $0.7015 \pm 0.0028$ | $0.8354 \pm 0.0033$ | $0.8254 \pm 0.0015$ |
| A-SCAFFOLD | $0.7815 \pm 0.0016$ | $0.7119 \pm 0.0024$ | $0.8336 \pm 0.0019$ | $0.8315 \pm 0.0011$ |
| RS-Minimax | $0.7976 \pm 0.0021$ | $\mathbf{0.8083 \pm 0.0025}$ | $0.8154 \pm 0.0031$ | $0.8086 \pm 0.0016$ |
| RS-Pairwise | $0.8059 \pm 0.0014$ | $0.7712 \pm 0.0027$ | $0.8304 \pm 0.0032$ | $0.8217 \pm 0.0023$ |
| CyCp-Minimax | $\mathbf{0.8184 \pm 0.0018}$ | $\mathbf{0.8095 \pm 0.0017}$ | $\mathbf{0.8702 \pm 0.0023}$ | $\mathbf{0.8511 \pm 0.0008}$ |
| CyCp-Pairwise | $\mathbf{0.8283 \pm 0.0012}$ | $0.7828 \pm 0.0020$ | $\mathbf{0.8626 \pm 0.0027}$ | $\mathbf{0.8540 \pm 0.0014}$ |

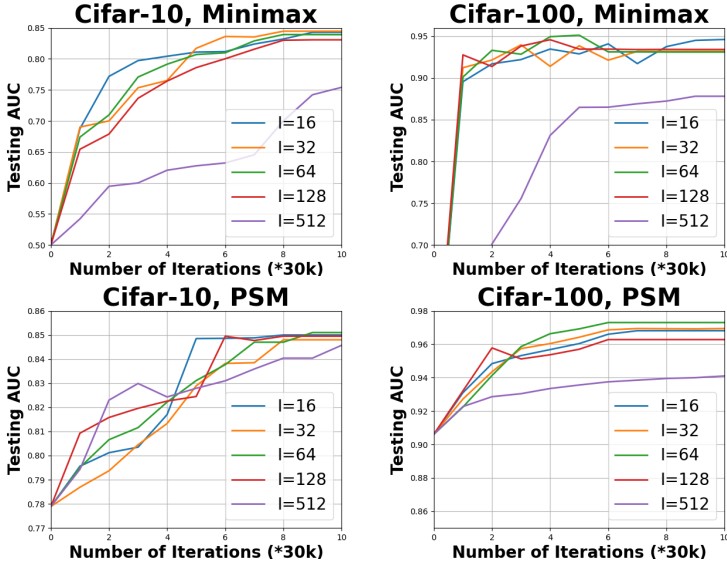

Figure 1: Ablation Study: The effect of communication interval $I$.

Table 4: Results with different Dirichlet parameter $dir$ and flip ratio $flip$ on Cifar-100

|  | $dir$=0.1,$flip$=0 | $dir$=0.1,$flip$=0.2 | $dir$=10,$flip$=0 | $dir$=10,$flip$=0.2 |
|---|---|---|---|---|
| CyCp-FedAVG | $0.9203 \pm 0.0009$ | $0.8774 \pm 0.0018$ | $0.9587 \pm 0.0025$ | $0.9363 \pm 0.0024$ |
| A-FedAVG | $0.9230 \pm 0.0014$ | $0.8825 \pm 0.0026$ | $0.9605 \pm 0.0030$ | $0.9402 \pm 0.0023$ |
| A-SCAFFOLD | $0.9199 \pm 0.0021$ | $0.8906 \pm 0.0015$ | $\mathbf{0.9627 \pm 0.0027}$ | $0.9421 \pm 0.0016$ |
| RS-Minimax | $0.9216 \pm 0.0011$ | $0.9123 \pm 0.0022$ | $0.9520 \pm 0.0018$ | $0.9266 \pm 0.0030$ |
| RS-Pairwise | $0.9243 \pm 0.0015$ | $\mathbf{0.9180 \pm 0.0024}$ | $0.9601 \pm 0.0017$ | $0.9475 \pm 0.0026$ |
| CyCp-Minimax | $\mathbf{0.9299 \pm 0.0013}$ | $0.9148 \pm 0.0017$ | $0.9610 \pm 0.0023$ | $\mathbf{0.9564 \pm 0.0019}$ |
| CyCp-Pairwise | $\mathbf{0.9308 \pm 0.0016}$ | $0.9123 \pm 0.0028$ | $\mathbf{0.9669 \pm 0.0015}$ | $\mathbf{0.9525 \pm 0.0021}$ |

Table 5: Results with different Dirichlet parameter $dir$ and flip ratio $flip$ on ChestMNIST

|  | $dir$=0.1,$flip$=0 | $dir$=0.1,$flip$=0.2 | $dir$=10,$flip$=0 | $dir$=10,$flip$=0.2 |
|---|---|---|---|---|
| CyCp-FedAVG | $\mathbf{0.5584 \pm 0.0006}$ | $0.5329 \pm 0.0003$ | $0.6561 \pm 0.0013$ | $0.6197 \pm 0.0017$ |
| A-FedAVG | $0.5580 \pm 0.0004$ | $0.5334 \pm 0.0006$ | $\mathbf{0.6572 \pm 0.0015}$ | $0.6201 \pm 0.0013$ |
| A-SCAFFOLD | $0.5573 \pm 0.0004$ | $0.5341 \pm 0.0002$ | $0.6568 \pm 0.0016$ | $0.6233 \pm 0.0019$ |
| RS-Minimax | $0.5245 \pm 0.0011$ | $0.5155 \pm 0.0004$ | $0.6541 \pm 0.0008$ | $0.6232 \pm 0.0017$ |
| RS-Pairwise | $0.5502 \pm 0.0007$ | $0.5338 \pm 0.0009$ | $0.6545 \pm 0.0012$ | $0.6293 \pm 0.0014$ |
| CyCp-Minimax | $0.5553 \pm 0.0010$ | $\mathbf{0.5362 \pm 0.0005}$ | $0.6431 \pm 0.0012$ | $\mathbf{0.6318 \pm 0.0018}$ |
| CyCp-Pairwise | $\mathbf{0.5598 \pm 0.0009}$ | $\mathbf{0.5498 \pm 0.0004}$ | $\mathbf{0.6595 \pm 0.0010}$ | $\mathbf{0.6356 \pm 0.0023}$ |

## 7 Conclusion

This paper studies federated optimization of non-ERM objectives, with a focus on AUC maximization under the practical constraint of cyclic client participation. We propose communication-efficient algorithms designed for this setting. For minimax-reformulated AUC, we introduce a stagewise method with auxiliary sequences to manage cyclic dependencies. For general pairwise losses, we develop an active-passive strategy that shares prediction scores across clients. We provide theoretical guarantees on both iteration and communication complexity, and validate the effectiveness of our algorithms on diverse tasks involving 50–100 clients.

## 8 Limitations and Future Work

Our work has several limitations. First, the fast communication rate of $\widetilde{O}(1/\epsilon^{1/2})$ depends on the PL condition, which may limit the generalizability of the associated algorithms. Second, the convergence rates in both the PL and non-PL regimes have not yet reached known lower bounds, leaving room for improvement. Third, developing methods to formally ensure differential privacy remains an open question. Finally, extending our approach beyond AUC maximization to a broader class of non-ERM objectives constitutes an important direction for future research.

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

# A Notations

Table 6: Notations

| | |
|---|---|
| $\mathbf{w} \in \mathbb{R}^d$ | Model parameters of the neural network, variables to be trained |
| $\mathbf{w}_{m,t}^{e,k} \in \mathbb{R}^d$ | Model parameters of machine $m$ of group $k$ at epoch $e$, iteration $t$, stage index is omitted when context ic clear |
| $a, b, \alpha \in \mathbb{R}^1$ | Introduced variables in minimax objective function |
| $\mathbf{v} = (\mathbf{w}, a, b) \in \mathbb{R}^{d+2}$ | Primal variable in minimax objective function. |
| $\mathbf{w}^{e,k}, \mathbf{v}^{e,k}, \alpha^{e,k}$ | Outputs of group/round $k$ of epoch $e$. Averages over all participating clients. |
| $\mathbf{z}$ | A data point |
| $\mathbf{z}_i$ | A data point from machine $i$ |
| $\mathbf{z}_{m,t}^{e,k}$ | A data point sampled on machine $m$ of group $k$ at epoch $e$, iteration $t$ |
| $\mathbf{z}_{m,t,1}^{e,k}, \mathbf{z}_{m,t,2}^{e,k}$ | Two independent data points (positive and negative, respectively) sampled on machine $m$ of group $k$ at epoch $e$, iteration $t$ |
| $h(\mathbf{w}; \mathbf{z}) \in \mathbb{R}^1$ | The prediction score of data $\mathbf{z}$ by network $\mathbf{w}$ |
| $[X]$ | The set of integers $\{1, 2, \ldots, X\}$, where $X$ is an integer. |
| $G_{m,t,1}^{e,k}, G_{m,t,2}^{e,k}$ | Local stochastic estimators of components of gradient |
| $\mathcal{H}_1^e, \mathcal{H}_2^e$ | Global buffers of historical prediction scores for positive and negative data. |
| $\mathcal{H}_{m,1}^{e,k}, \mathcal{H}_{m,2}^{e,k}$ | Collected historical prediction scores on machine $m$ of group/round $k$ at epoch $e$ |
| $\mathcal{R}_{i,1}^{e,k}, \mathcal{R}_{i,2}^{e,k}$ | Trasmitted buffer from server to client |
| $h_\epsilon^{e-1}, h_\zeta^{e-1}$ | Predictions scores sampled from the collected scores of epoch $e-1$ |

# B Analysis of CyCp Minimax

## B.1 Auxiliary Lemmas

For the stagewise algorithm for minimax problem, we define the duality gap of $s$-th stage at a point $(\mathbf{v}, \alpha)$ as

$$Gap_s(\mathbf{v}, \alpha) = \max_{\alpha'} f^s(\mathbf{v}, \alpha') - \min_{\mathbf{v}'} f^s(\mathbf{v}', \alpha). \tag{15}$$

Before we show the proofs, we first present the auxiliary lemmas from Yan et al. (2020).

**Lemma B.1** (Lemma 1 of Yan et al. (2020)). *Suppose a function $f(\mathbf{v}, \alpha)$ is $\lambda_1$-strongly convex in $\mathbf{v}$ and $\lambda_2$-strongly concave in $\alpha$. Consider the following problem*

$$\min_{\mathbf{v} \in X} \max_{\alpha \in Y} f(\mathbf{v}, \alpha),$$

*where $X$ and $Y$ are convex compact sets. Denote $\hat{\mathbf{v}}_f(\alpha) = \arg\min_{\mathbf{v}' \in X} f(\mathbf{v}', \alpha)$ and $\hat{\alpha}_f(\mathbf{v}) = \arg\max_{\alpha' \in Y} f(\mathbf{v}, \alpha')$. Suppose we have two solutions $(\mathbf{v}_0, \alpha_0)$ and $(\mathbf{v}_1, \alpha_1)$. Then the following relation between variable distance and duality gap holds*

$$\frac{\lambda_1}{4}\|\hat{\mathbf{v}}_f(\alpha_1) - \mathbf{v}_0\|^2 + \frac{\lambda_2}{4}\|\hat{\alpha}_f(\mathbf{v}_1) - \alpha_0\|^2 \leq \max_{\alpha' \in Y} f(\mathbf{v}_0, \alpha') - \min_{\mathbf{v}' \in X} f(\mathbf{v}', \alpha_0) \\ + \max_{\alpha' \in Y} f(\mathbf{v}_1, \alpha') - \min_{\mathbf{v}' \in X} f(\mathbf{v}', \alpha_1). \tag{16}$$

$\square$

**Lemma B.2** (Lemma 5 of Yan et al. (2020)). *We have the following lower bound for $Gap_s(\mathbf{v}_s, \alpha_s)$*

$$Gap_s(\mathbf{v}_s, \alpha_s) \geq \frac{3}{50} Gap_{s+1}(\mathbf{v}_0^{s+1}, \alpha_0^{s+1}) + \frac{4}{5}(\phi(\mathbf{v}_0^{s+1}) - \phi(\mathbf{v}_0^s)),$$

*where $\mathbf{v}_0^{s+1} = \mathbf{v}_s$ and $\alpha_0^{s+1} = \alpha_s$, i.e., the initialization of $(s+1)$-th stage is the output of the $s$-th stage.*

## B.2 Lemmas

Utilizing the strong convexity/concavity and smoothness of $f(\cdot, \cdot)$, we have the following lemma.

**Lemma B.3.** *Suppose Assumption 4.1 holds and by running Algorithm 1 with input $(\mathbf{v}^{0,0}, \alpha^{0,0})$. For any $(\mathbf{v}, \alpha)$, the outputs $(\bar{\mathbf{v}}, \bar{\alpha})$ satisfies*

$$
f^s(\bar{\mathbf{v}}, \alpha) - f^s(\mathbf{v}, \bar{\alpha}) \leq \frac{1}{E} \sum_{e=0}^{E-1} [f^s(\mathbf{v}^{e+1,0}, \alpha) - f^s(\mathbf{v}, \alpha^{e+1,0})]
$$

$$
\leq \frac{1}{E} \sum_{e=0}^{E-1} \left[ \underbrace{\langle \partial_{\mathbf{v}} f^s(\mathbf{v}^{e,0}, \alpha^{e,0}), \mathbf{v}^{e+1,0} - \mathbf{v} \rangle}_{A_1} + \underbrace{\langle \partial_{\alpha} f^s(\mathbf{v}^{e,0}, \alpha^{e,0}), \alpha - \alpha^{e+1,0} \rangle}_{A_2} \right.
$$
$$
\left. + \underbrace{\frac{3\ell + 3\ell^2/\mu_2}{2} \|\mathbf{v}^{e+1,0} - \mathbf{v}^{e,0}\|^2 + 2\ell(\alpha^{e+1,0} - \alpha^{e,0})^2}_{A_3} - \frac{\ell}{3} \|\mathbf{v} - \mathbf{v}^{e,0}\|^2 - \frac{\mu_2}{3} (\alpha^{e,0} - \alpha)^2 \right].
$$

*Proof.* For any $\mathbf{v}$ and $\alpha$, using Jensen's inequality and the fact that $f^s(\mathbf{v}, \alpha)$ is convex in $\mathbf{v}$ and concave in $\alpha$,

$$
f^s(\bar{\mathbf{v}}, \alpha) - f^s(\mathbf{v}, \bar{\alpha}) \leq \frac{1}{E} \sum_{e=1}^{E} \left( f^s(\mathbf{v}^{e,0}, \alpha) - f^s(\mathbf{v}, \alpha^{e,0}) \right). \tag{17}
$$

By $\ell$-strongly convexity of $f^s(\mathbf{v}, \alpha)$ in $\mathbf{v}$, we have

$$
f^s(\mathbf{v}^{e,0}, \alpha^{e,0}) + \langle \partial_{\mathbf{v}} f^s(\mathbf{v}^{e,0}, \alpha^{e,0}), \mathbf{v} - \mathbf{v}^{e,0} \rangle + \frac{\ell}{2} \|\mathbf{v}^{e,0} - \mathbf{v}\|^2 \leq f(\mathbf{v}, \alpha^{e,0}). \tag{18}
$$

By $3\ell$-smoothness of $f^s(\mathbf{v}, \alpha)$ in $\mathbf{v}$, we have

$$
f^s(\mathbf{v}^{e+1,0}, \alpha) \leq f^s(\mathbf{v}^{e,0}, \alpha) + \langle \partial_{\mathbf{v}} f^s(\mathbf{v}^{e,0}, \alpha), \mathbf{v}^{e+1,0} - \mathbf{v}^{e,0} \rangle + \frac{3\ell}{2} \|\mathbf{v}^{e+1,0} - \mathbf{v}^{e,0}\|^2
$$
$$
= f^s(\mathbf{v}^{e,0}, \alpha) + \langle \partial_{\mathbf{v}} f^s(\mathbf{v}^{e,0}, \alpha^{e,0}), \mathbf{v}^{e+1,0} - \mathbf{v}^{e,0} \rangle + \frac{3\ell}{2} \|\mathbf{v}^{e+1,0} - \mathbf{v}^{e,0}\|^2
$$
$$
\quad + \langle \partial_{\mathbf{v}} f^s(\mathbf{v}^{e,0}, \alpha) - \partial_{\mathbf{v}} f^s(\mathbf{v}^{e,0}, \alpha^{e,0}), \mathbf{v}^{e+1,0} - \mathbf{v}^{e,0} \rangle
$$
$$
\overset{(a)}{\leq} f^s(\mathbf{v}^{e,0}, \alpha) + \langle \partial_{\mathbf{v}} f^s(\mathbf{v}^{e,0}, \alpha^{e,0}), \mathbf{v}^{e+1,0} - \mathbf{v}^{e,0} \rangle + \frac{3\ell}{2} \|\mathbf{v}^{e+1,0} - \mathbf{v}^{e,0}\|^2 \tag{19}
$$
$$
\quad + \ell |\alpha^{e,0} - \alpha| \|\mathbf{v}^{e+1,0} - \mathbf{v}^{e,0}\|
$$
$$
\overset{(b)}{\leq} f^s(\mathbf{v}^{e,0}, \alpha) + \langle \partial_{\mathbf{v}} f^s(\mathbf{v}^{e,0}, \alpha^{e,0}), \mathbf{v}^{e+1,0} - \mathbf{v}^{e,0} \rangle + \frac{3\ell}{2} \|\mathbf{v}^{e+1,0} - \mathbf{v}^{e,0}\|^2
$$
$$
\quad + \frac{\mu_2}{6} (\alpha^{e,0} - \alpha)^2 + \frac{3\ell^2}{2\mu_2} \|\bar{\mathbf{v}}^{e+1,0} - \mathbf{v}^{e,0}\|^2,
$$

where $(a)$ holds because that we know $\partial_{\mathbf{v}} f(\mathbf{v}, \alpha)$ is $\ell$-Lipschitz in $\alpha$ since $f(\mathbf{v}, \alpha)$ is $\ell$-smooth, $(b)$ holds by Young's inequality, and $\mu_2 = 2p(1-p)$ is the strong concavity coefficient of $f^s$ in $\alpha$.

Adding (18) and (19), rearranging terms, we have

$$
f^s(\mathbf{v}^{e,0}, \alpha^{e,0}) + f^s(\mathbf{v}^{e+1,0}, \alpha)
$$
$$
\leq f(\mathbf{v}, \alpha^{e,0}) + f(\mathbf{v}^{e,0}, \alpha) + \langle \partial_{\mathbf{v}} f(\mathbf{v}^{e,0}, \alpha^{e,0}), \mathbf{v}^{e+1,0} - \mathbf{v} \rangle + \frac{3\ell + 3\ell^2/\mu_2}{2} \|\mathbf{v}^{e+1,0} - \mathbf{v}^{e,0}\|^2 \tag{20}
$$
$$
\quad - \frac{\ell}{2} \|\mathbf{v}^{e,0} - \mathbf{v}\|^2 + \frac{\mu_2}{6} (\alpha^{e,0} - \alpha)^2.
$$

We know that $-f(\mathbf{v}, \alpha)$ is $\mu_2$-strong convexity of in $\alpha$. Thus, we have

$$
-f^s(\mathbf{v}^{e,0}, \alpha^{e,0}) - \partial_{\alpha} f^s(\mathbf{v}^{e,0}, \alpha^{e,0})^\top (\alpha - \alpha^{e,0}) + \frac{\mu_2}{2} (\alpha - \alpha^{e,0})^2 \leq -f^s(\mathbf{v}^{e,0}, \alpha). \tag{21}
$$

Since $f(\mathbf{v}, \alpha)$ is $\ell$-smooth in $\alpha$, we get

$$
\begin{aligned}
- f^s(\mathbf{v}, \alpha^{e+1,0}) &\leq -f^s(\mathbf{v}, \alpha^{e,0}) - \langle \partial_\alpha f^s(\mathbf{v}, \alpha^{e,0}), \alpha^{e+1,0} - \alpha^{e,0} \rangle + \frac{\ell}{2}(\alpha^{e+1,0} - \alpha^{e,0})^2 \\
&= -f^s(\mathbf{v}, \alpha^{e,0}) - \langle \partial_\alpha f^s(\mathbf{v}^{e,0}, \alpha^{e,0}), \alpha^{e+1,0} - \alpha^{e,0} \rangle + \frac{\ell}{2}(\alpha^{e+1,0} - \alpha^{e,0})^2 \\
&\quad - \langle \partial_\alpha f^s(\mathbf{v}, \alpha^{e,0}) - \partial_\alpha f^s(\mathbf{v}^{e,0}, \alpha^{e,0}), \alpha^{e+1,0} - \alpha^{e,0} \rangle \\
&\overset{(a)}{\leq} -f^s(\mathbf{v}, \alpha^{e,0}) - \langle \partial_\alpha f^s(\mathbf{v}^{e,0}, \alpha^{e,0}), \alpha^{e+1,0} - \alpha^{e,0} \rangle + \frac{\ell}{2}(\alpha^{e+1,0} - \alpha^{e,0})^2 + \ell \|\mathbf{v} - \mathbf{v}^{e,0}\| |\alpha^{e+1,0} - \alpha^{e,0}| \\
&\leq -f^s(\mathbf{v}, \alpha^{e,0}) - \langle \partial_\alpha f^s(\mathbf{v}^{e,0}, \alpha^{e,0}), \alpha^{e+1,0} - \alpha^{e,0} \rangle + \frac{\ell}{2}(\alpha^{e+1,0} - \alpha^{e,0})^2 + \frac{\ell}{6}\|\mathbf{v}^{e,0} - \mathbf{v}\|^2 + \frac{3\ell}{2}(\alpha^{e+1,0} - \alpha^{e,0})^2,
\end{aligned}
\tag{22}
$$

where (a) holds because that $\partial_\alpha f^s(\mathbf{v}, \alpha)$ is $\ell$-Lipschitz in $\mathbf{v}$.

Adding (21), (22) and arranging terms, we have

$$
\begin{aligned}
-f^s(\mathbf{v}^{e,0}, \alpha^{e,0}) - f^s(\mathbf{v}, \alpha^{e+1,0}) &\leq -f^s(\mathbf{v}^{e,0}, \alpha) - f^s(\mathbf{v}, \alpha^{e,0}) - \langle \partial_\alpha f^s(\mathbf{v}^{e,0}, \alpha^{e,0}), \alpha^{e+1,0} - \alpha \rangle \\
&\quad + 2\ell(\alpha^{e+1,0} - \alpha^{e,0})^2 + \frac{\ell}{6}\|\mathbf{v}^{e,0} - \mathbf{v}\|^2 - \frac{\mu_2}{2}(\alpha - \alpha^{e,0})^2.
\end{aligned}
\tag{23}
$$

Adding (20) and (23), we get

$$
\begin{aligned}
f^s(\mathbf{v}^{e+1,0}, \alpha) - f^s(\mathbf{v}, \alpha^{e+1,0}) &\leq \langle \partial_\mathbf{v} f(\mathbf{v}^{e,0}, \alpha^{e,0}), \mathbf{v}^{e+1,0} - \mathbf{v} \rangle - \langle \partial_\alpha f(\mathbf{v}^{e,0}, \alpha^{e,0}), \alpha^{e+1,0} - \alpha \rangle \\
&\quad + \frac{3\ell + 3\ell^2/\mu_2}{2}\|\mathbf{v}^{e+1,0} - \mathbf{v}^{e,0}\|^2 + 2\ell(\alpha^{e+1,0} - \alpha^{e-1,0})^2 - \frac{\ell}{3}\|\mathbf{v}^{e,0} - \mathbf{v}\|^2 - \frac{\mu_2}{3}(\alpha^{e,0} - \alpha)^2.
\end{aligned}
\tag{24}
$$

Taking average over $e, k, t$, we get

$$
\begin{aligned}
f^s(\bar{\mathbf{v}}, \alpha) - f^s(\mathbf{v}, \bar{\alpha}) &\leq \frac{1}{E}\sum_{e=0}^{E-1}[f^s(\mathbf{v}^{e+1,0}, \alpha) - f^s(\mathbf{v}, \alpha^{e+1,0})] \\
&\leq \frac{1}{E}\sum_{e=0}^{E-1}\bigg[ \underbrace{\langle \partial_\mathbf{v} f^s(\mathbf{v}^{e,0}, \alpha^{e,0}), \mathbf{v}^{e+1,0} - \mathbf{v} \rangle}_{A_1} + \underbrace{\langle \partial_\alpha f^s(\mathbf{v}^{e,0}, \alpha^{e,0}), \alpha - \alpha^{e+1,0} \rangle}_{A_2} \\
&\quad + \underbrace{\frac{3\ell + 3\ell^2/\mu_2}{2}\|\mathbf{v}^{e+1,0} - \mathbf{v}^{e,0}\|^2 + 2\ell(\alpha^{e+1,0} - \alpha^{e,0})^2}_{A_3} - \frac{\ell}{3}\|\mathbf{v} - \mathbf{v}^{e,0}\|^2 - \frac{\mu_2}{3}(\alpha^{e,0} - \alpha)^2 \bigg].
\end{aligned}
$$

$\square$

In the following, we will bound the term $A_1$ by Lemma 4.3, $A_2$ by Lemma B.4.

**Proof of Lemma 4.3** .

*Proof.*

$$
\begin{aligned}
\langle \nabla_\mathbf{v} f^s(\mathbf{v}^{e,0}, \alpha^{e,0}), \mathbf{v}^{e+1,0} - \mathbf{v} \rangle &= \bigg\langle \frac{1}{K}\sum_{k=1}^{K}\frac{1}{M}\sum_{m \in \mathcal{G}^{e,k}} \nabla_\mathbf{v} f_{k,m}^s(\mathbf{v}^{e,0}, \alpha^{e,0}), \mathbf{v}^{e+1,0} - \mathbf{v} \bigg\rangle \\
&= \bigg\langle \frac{1}{K}\sum_{k=1}^{K}\frac{1}{M}\sum_{m \in \mathcal{G}^{e,k}} [\frac{1}{I}\sum_t \nabla_\mathbf{v} f_{k,m}^s(\mathbf{v}^{e,0}, \alpha^{e,0})] - \frac{1}{K}\sum_{k=1}^{K}\frac{1}{M}\sum_{m \in \mathcal{G}^{e,k}} [\frac{1}{I}\sum_t \nabla_\mathbf{v} F_{k,m}^s(\mathbf{v}_{m,t}^{e,k}, \alpha_{m,t}^{e,k}; \mathbf{z}_{m,t}^{e,k})], \mathbf{v}^{e+1,0} - \mathbf{v} \bigg\rangle \quad ③ \\
&\quad + \bigg\langle \frac{1}{K}\sum_{k=1}^{K}\frac{1}{M}\sum_{m \in \mathcal{G}^{e,k}} [\frac{1}{I}\sum_t \nabla_\mathbf{v} F_{k,m}^s(\mathbf{v}_{m,t}^{e,k}, \alpha_{m,t}^{e,k}; \mathbf{z}_{m,t}^{e,k})], \mathbf{v}^{e+1,0} - \mathbf{v} \bigg\rangle \quad ④.
\end{aligned}
\tag{25}
$$

Using $\hat{\mathbf{v}}^{e+1,0} = \mathbf{v}^{e,0} - \eta I \sum_{k=1}^{K} \frac{1}{M} \sum_{m \in \mathcal{G}^{e,k}} \nabla_{\mathbf{v}} f^s(\mathbf{v}^{e,0}, \alpha^{e,0})$, then we have

$$\mathbf{v}^{e+1,0} - \hat{\mathbf{v}}^{e+1,0} = \eta \left( \sum_{k=1}^{K} \frac{1}{M} \sum_{m \in \mathcal{G}^{e,k}} \sum_{t=1}^{I} \nabla_{\mathbf{v}} f_{k,m}^s(\mathbf{v}^{e,0}, \alpha^{e,0}) - \sum_{k=1}^{K} \frac{1}{M} \sum_{m \in \mathcal{G}^{e,k}} \sum_{t=1}^{I} \nabla_{\mathbf{v}} f_{k,m}^s(\mathbf{v}_{m,t}^{e,k}, \alpha_{m,t}^{e,k}; \mathbf{z}_{m,t}^{e,k}) \right). \tag{26}$$

Hence we get

$$\begin{aligned}
\text{③} &= \left\langle \frac{1}{K} \sum_{k=1}^{K} \frac{1}{M} \sum_{m \in \mathcal{G}^{e,k}} [\frac{1}{I} \sum_t \nabla_{\mathbf{v}} f_{k,m}^s(\mathbf{v}^{e,0}, \alpha^{e,0})] - \frac{1}{K} \sum_{k=1}^{K} \frac{1}{M} \sum_{m \in \mathcal{G}^{e,k}} [\frac{1}{I} \sum_t \nabla_{\mathbf{v}} F_{k,m}^s(\mathbf{v}_{m,t}^{e,k}, \alpha_{m,t}^{e,k}; \mathbf{z}_{m,t}^{e,k})], \mathbf{v}^{e+1,0} - \hat{\mathbf{v}}^{e+1,0} \right\rangle \\
&\quad + \left\langle \frac{1}{K} \sum_{k=1}^{K} \frac{1}{M} \sum_{m \in \mathcal{G}^{e,k}} [\frac{1}{I} \sum_t \nabla_{\mathbf{v}} f_{k,m}^s(\mathbf{v}^{e,0}, \alpha^{e,0})] - \frac{1}{K} \sum_{k=1}^{K} \frac{1}{M} \sum_{m \in \mathcal{G}^{e,k}} [\frac{1}{I} \sum_t \nabla_{\mathbf{v}} F_{k,m}^s(\mathbf{v}_{m,t}^{e,k}, \alpha_{m,t}^{e,k}; \mathbf{z}_{m,t}^{e,k})], \hat{\mathbf{v}}^{e+1,0} - \mathbf{v} \right\rangle \\
&= \eta K I \left\| \frac{1}{K} \sum_{k=1}^{K} \frac{1}{M} \sum_{m \in \mathcal{G}^{e,k}} \frac{1}{I} \sum_t [\nabla_{\mathbf{v}} f_{k,m}^s(\mathbf{v}^{e,0}, \alpha^{e,0}) - \nabla_{\mathbf{v}} F_{k,m}^s(\mathbf{v}_{m,t}^{e,k}, \alpha_{m,t}^{e,k}; z_{m,t}^{e,k})] \right\|^2 \\
&\quad + \left\langle \frac{1}{K} \sum_{k=1}^{K} \frac{1}{M} \sum_{m \in \mathcal{G}^{e,k}} [\frac{1}{I} \sum_t \nabla_{\mathbf{v}} f_{k,m}^s(\mathbf{v}^{e,0}, \alpha^{e,0})] - \frac{1}{K} \sum_{k=1}^{K} \frac{1}{M} \sum_{m \in \mathcal{G}^{e,k}} [\frac{1}{I} \sum_t \nabla_{\mathbf{v}} F_{k,m}^s(\mathbf{v}^{e,0}, \alpha^{e,0}; \mathbf{z}_{m,t}^{e,k})], \hat{\mathbf{v}}^{e+1,0} - \mathbf{v} \right\rangle \\
&\quad + \left\langle \frac{1}{K} \sum_{k=1}^{K} \frac{1}{M} \sum_{m \in \mathcal{G}^{e,k}} [\frac{1}{I} \sum_t \nabla_{\mathbf{v}} F_{k,m}^s(\mathbf{v}^{e,0}, \alpha^{e,0}; \mathbf{z}_{m,t}^{e,k})] - \frac{1}{K} \sum_{k=1}^{K} \frac{1}{M} \sum_{m \in \mathcal{G}^{e,k}} [\frac{1}{I} \sum_t \nabla_{\mathbf{v}} F_{k,m}^s(\mathbf{v}_{m,t}^{e,k}, \alpha_{m,t}^{e,k}; \mathbf{z}_{m,t}^{e,k})], \hat{\mathbf{v}}^{e+1,0} - \mathbf{v} \right\rangle,
\end{aligned} \tag{27}$$

where

$$\begin{aligned}
&\left\langle \frac{1}{K} \sum_{k=1}^{K} \frac{1}{M} \sum_{m \in \mathcal{G}^{e,k}} [\frac{1}{I} \sum_t \nabla_{\mathbf{v}} F_{k,m}^s(\mathbf{v}^{e,0}, \alpha^{e,0}; \mathbf{z}_{m,t}^{e,k})] - \frac{1}{K} \sum_{k=1}^{K} \frac{1}{M} \sum_{m \in \mathcal{G}^{e,k}} [\frac{1}{I} \sum_t \nabla_{\mathbf{v}} F_{k,m}^s(\mathbf{v}_{m,t}^{e,k}, \alpha_{m,t}^{e,k}; \mathbf{z}_{m,t}^{e,k})], \hat{\mathbf{v}}^{e+1,0} - \mathbf{v} \right\rangle \\
&\leq 3\ell \frac{1}{K} \sum_{k=1}^{K} \frac{1}{M} \sum_{m \in \mathcal{G}^{e,k}} \frac{1}{I} \sum_t \|\mathbf{v}^{e,0} - \mathbf{v}_{m,t}^{e,k}\|^2 + \frac{\ell}{6} \|\hat{\mathbf{v}}^{e+1,0} - \mathbf{v}\|^2.
\end{aligned} \tag{28}$$

Define another auxiliary sequence as

$$\tilde{\mathbf{v}}^{e+1,0} = \tilde{\mathbf{v}}^{e,0} - \frac{\eta}{M} \sum_{k=1}^{K} \sum_{m \in \mathcal{G}^{e,k}} \sum_{t=1}^{I} \left( \nabla_{\mathbf{v}} F_{k,m}^s(\mathbf{v}^{e,0}, \alpha^{e,0}; z_{m,t}^{e,k}) - \nabla_{\mathbf{v}} f_{k,m}^s(\mathbf{v}^{e,0}, \alpha^{e,0}) \right), \text{ for } t > 0; \tilde{\mathbf{v}}_0 = \mathbf{v}_0. \tag{29}$$

Denote

$$\Theta_e(\mathbf{v}) = \left( \frac{1}{M} \sum_{k=1}^{K} \sum_{m \in \mathcal{G}^{e,k}} \sum_{t=1}^{I} \left( \nabla_{\mathbf{v}} f_{k,m}^s(\mathbf{v}^{e,0}, \alpha^{e,0}) - \nabla_{\mathbf{v}} F_{k,m}^s(\mathbf{v}^{e,0}, \alpha^{e,0}; z_{m,t}^{e,k}) \right) \right)^\top \mathbf{v} + \frac{1}{2\eta} \|\mathbf{v} - \tilde{\mathbf{v}}^{e,0}\|^2. \tag{30}$$

Hence, for the auxiliary sequence $\tilde{\alpha}_t$, we can verify that

$$\tilde{\mathbf{v}}^{e+1,0} = \arg\min_{\mathbf{v}} \Theta_e(\mathbf{v}). \tag{31}$$

Since $\Theta_e(\mathbf{v})$ is $\frac{1}{\eta}$-strongly convex, we have

$$
\begin{aligned}
\frac{1}{2\eta}\|\mathbf{v} - \tilde{\mathbf{v}}^{e+1,0}\|^2 &\leq \Theta_e(\mathbf{v}) - \Theta_e(\tilde{\mathbf{v}}^{e+1,0}) \\
&= \left( \frac{1}{M} \sum_{k=1}^{K} \sum_{m \in \mathcal{G}^{e,k}} \sum_{t=1}^{I} \left( \nabla_{\mathbf{v}} f_{k,m}^s(\mathbf{v}^{e,0}, \alpha^{e,0}) - \nabla_{\mathbf{v}} F_{k,m}^s(\mathbf{v}^{e,0}, \alpha^{e,0}; z_{m,t}^{e,k}) \right) \right)^{\top} \mathbf{v} + \frac{1}{2\eta}\|\mathbf{v} - \tilde{\mathbf{v}}^{e,0}\|^2 \\
&\quad - \left( \frac{1}{M} \sum_{k=1}^{K} \sum_{m \in \mathcal{G}^{e,k}} \sum_{t=1}^{I} \left( \nabla_{\mathbf{v}} f_{k,m}^s(\mathbf{v}^{e,0}, \alpha^{e,0}) - \nabla_{\mathbf{v}} F_{k,m}^s(\mathbf{v}^{e,0}, \alpha^{e,0}; z_{m,t}^{e,k}) \right) \right)^{\top} \tilde{\mathbf{v}}^{e+1,0} - \frac{1}{2\eta}\|\tilde{\mathbf{v}}^{e+1,0} - \tilde{\mathbf{v}}^{e,0}\|^2 \\
&= \left( \frac{1}{M} \sum_{k=1}^{K} \sum_{m \in \mathcal{G}^{e,k}} \sum_{t=1}^{I} \left( \nabla_{\mathbf{v}} f_{k,m}^s(\mathbf{v}^{e,0}, \alpha^{e,0}) - \nabla_{\mathbf{v}} F_{k,m}^s(\mathbf{v}^{e,0}, \alpha^{e,0}; z_{m,t}^{e,k}) \right) \right)^{\top} (\mathbf{v} - \tilde{\mathbf{v}}^{e,0}) + \frac{1}{2\eta}\|\mathbf{v} - \tilde{\mathbf{v}}^{e,0}\|^2 \\
&\quad - \left( \frac{1}{M} \sum_{k=1}^{K} \sum_{m \in \mathcal{G}^{e,k}} \sum_{t=1}^{I} \left( \nabla_{\mathbf{v}} f_{k,m}^s(\mathbf{v}^{e,0}, \alpha^{e,0}) - \nabla_{\mathbf{v}} F_{k,m}^s(\mathbf{v}^{e,0}, \alpha^{e,0}; z_{m,t}^{e,k}) \right) \right)^{\top} (\tilde{\mathbf{v}}^{e+1,0} - \tilde{\mathbf{v}}^{e,0}) - \frac{1}{2\eta}\|\tilde{\mathbf{v}}^{e+1,0} - \tilde{\mathbf{v}}^{e,0}\|^2 \\
&\leq \left( \frac{1}{M} \sum_{k=1}^{K} \sum_{m \in \mathcal{G}^{e,k}} \sum_{t=1}^{I} \left( \nabla_{\mathbf{v}} f_{k,m}^s(\mathbf{v}^{e,0}, \alpha^{e,0}) - \nabla_{\mathbf{v}} F_{k,m}^s(\mathbf{v}^{e,0}, \alpha^{e,0}; z_{m,t}^{e,k}) \right) \right)^{\top} (\mathbf{v} - \tilde{\mathbf{v}}^{e,0}) + \frac{1}{2\eta}\|\mathbf{v} - \tilde{\mathbf{v}}^{e,0}\|^2 \\
&\quad + \frac{\eta}{2} \left\| \frac{1}{M} \sum_{k=1}^{K} \sum_{m \in \mathcal{G}^{e,k}} \sum_{t=1}^{I} \left( \nabla_{\mathbf{v}} f_{k,m}^s(\mathbf{v}^{e,0}, \alpha^{e,0}) - \nabla_{\mathbf{v}} F_{k,m}^s(\mathbf{v}^{e,0}, \alpha^{e,0}; z_{m,t}^{e,k}) \right) \right\|^2.
\end{aligned}
\tag{32}
$$

Multiplying $1/KI$ on both sides and adding this with (27), we get

$$
\begin{aligned}
③ &\leq \eta K I \left\| \frac{1}{MKI} \sum_{k=1}^{K} \sum_{m \in \mathcal{G}^{e,k}} \sum_{t=1}^{I} \left( \nabla_{\mathbf{v}} f_k^s(\mathbf{v}^{e,0}, \alpha^{e,0}) - \nabla_{\mathbf{v}} F_k^s(\mathbf{v}_{m,t}^{e,k}, \alpha_{m,t}^{e,k}; z_{m,t}^{e,k}) \right) \right\|^2 \\
&\quad + \frac{\eta K I}{2} \left\| \frac{1}{MKI} \sum_{k=1}^{K} \sum_{m \in \mathcal{G}^{e,k}} \sum_{t=1}^{I} \left( \nabla_{\mathbf{v}} f_k^s(\mathbf{v}^{e,0}, \alpha^{e,0}) - \nabla_{\mathbf{v}} F_k^s(\mathbf{v}^{e,0}, \alpha^{e,0}; z_{m,t}^{e,k}) \right) \right\|^2 \\
&\quad + \frac{1}{2\eta K I}\|\mathbf{v} - \tilde{\mathbf{v}}^{e,0}\|^2 - \frac{1}{2\eta K I}\|\mathbf{v} - \tilde{\mathbf{v}}^{e+1,0}\|^2 \\
&\quad + \left\langle \frac{1}{MKI} \sum_{k=1}^{K} \sum_{m \in \mathcal{G}^{e,k}} \sum_{t=1}^{I} \left( \nabla_{\mathbf{v}} f_k^s(\mathbf{v}^{e,0}, \alpha^{e,0}) - \nabla_{\mathbf{v}} F_k^s(\mathbf{v}^{e,0}, \alpha^{e,0}; z_{m,t}^{e,k}) \right), \hat{\mathbf{v}}^{e+1,0} - \tilde{\mathbf{v}}^{e,0} \right\rangle.
\end{aligned}
\tag{33}
$$

where

$$
\mathbb{E} \left\langle \frac{1}{MKI} \sum_{k=1}^{K} \sum_{m \in \mathcal{G}^{e,k}} \sum_{t=1}^{I} \left( \nabla_{\mathbf{v}} f_k^s(\mathbf{v}^{e,0}, \alpha^{e,0}) - \nabla_{\mathbf{v}} F_k^s(\mathbf{v}^{e,0}, \alpha^{e,0}; z_{m,t}^{e,k}) \right), \hat{\mathbf{v}}^{e+1,0} - \tilde{\mathbf{v}}^{e,0} \right\rangle = 0.
\tag{34}
$$

④ can be bounded as

$$
④ = -\frac{1}{\eta K I}\langle \mathbf{v}^{e+1,0} - \mathbf{v}^{e,0}, \mathbf{v}^{e+1,0} - \mathbf{v} \rangle = \frac{1}{2\eta K I}(\|\mathbf{v}^{e,0} - \mathbf{v}\|^2 - \|\mathbf{v}^{e,0} - \mathbf{v}^{e+1,0}\|^2 - \|\mathbf{v}^{e+1,0} - \mathbf{v}\|^2).
\tag{35}
$$

Plugging (33) and (35) into (25) and noting $\eta KI \leq O(1)$, we get

$$\mathbb{E}_{e,0} \left\langle \nabla_{\mathbf{v}} f(\mathbf{v}^{e,0}, \alpha^{e,0}), \mathbf{v}^{e+1,0} - \mathbf{v} \right\rangle$$

$$\leq 3\eta KI \mathbb{E} \left\| \frac{1}{MKI} \sum_{k=1}^{K} \sum_{m \in \mathcal{G}^{e,k}} \sum_{t=1}^{I} \left( \nabla_{\mathbf{v}} F_k^s(\mathbf{v}^{e,0}, \alpha^{e,0}; z_{m,t}^{e,k}) - \nabla_{\mathbf{v}} f_k^s(\mathbf{v}^{e,0}, \alpha^{e,0}) \right) \right\|^2$$

$$+ \frac{5\ell}{KMI} \sum_{k} \sum_{m \in \mathcal{G}^{e,k}} \sum_{t} \mathbb{E}(\|\mathbf{v}^{e,0} - \mathbf{v}_{m,t}^{e,k}\|^2 + \|\alpha^{e,0} - \alpha_{m,t}^{e,k}\|^2)$$

$$+ \frac{1}{2\eta KI} \mathbb{E}(\|\mathbf{v}^{e,0} - \mathbf{v}\|^2 - \|\mathbf{v}^{e,0} - \mathbf{v}^{e+1,0}\|^2 - \|\mathbf{v}^{e+1,0} - \mathbf{v}\|^2)$$

$$+ \frac{1}{2\eta KI}(\|\mathbf{v} - \tilde{\mathbf{v}}^{e,0}\|^2 - \|\mathbf{v} - \tilde{\mathbf{v}}^{e+1,0}\|^2) + \frac{\ell}{3} \mathbb{E}\|\hat{\mathbf{v}}^{e+1,0} - \mathbf{v}\|^2$$

$$\leq \frac{6\tilde{\eta} K \sigma^2}{MKI} + \frac{6\ell}{KMI} \sum_{k} \sum_{m \in \mathcal{G}^{e,k}} \sum_{t} \mathbb{E}(\|\mathbf{v}^{e,0} - \mathbf{v}_{m,t}^{e,k}\|^2 + \|\alpha^{e,0} - \alpha_{m,t}^{e,k}\|^2) + \frac{\ell}{3}\|\mathbf{v}^{e+1,0} - \mathbf{v}\|^2$$

$$+ \frac{1}{2\eta KI} \mathbb{E}(\|\mathbf{v}^{e,0} - \mathbf{v}\|^2 - \|\mathbf{v}^{e,0} - \mathbf{v}^{e+1,0}\|^2 - \|\mathbf{v}^{e+1,0} - \mathbf{v}\|^2) + \frac{1}{2\eta KI}(\|\mathbf{v} - \tilde{\mathbf{v}}^{e,0}\|^2 - \|\mathbf{v} - \tilde{\mathbf{v}}^{e+1,0}\|^2),$$

where the last inequality uses

$$\|\hat{\mathbf{v}}^{e+1,0} - \mathbf{v}\|^2 \leq 2\|\hat{\mathbf{v}}^{e+1,0} - \mathbf{v}^{e+1,0}\|^2 + 2\|\mathbf{v}^{e+1,0} - \mathbf{v}\|^2, \tag{36}$$

and $\|\hat{\mathbf{v}}^{e+1,0} - \mathbf{v}^{e+1,0}\|^2$ is addressed similarly as before in ③. □

Similarly, $A_2$ can be bounded as

**Lemma B.4.** *Define* $\hat{\alpha}^{e+1,0} = \alpha^{e,0} + \eta \sum_{k=1}^{K} \frac{1}{M} \sum_{m \in \mathcal{G}^{e,k}} \sum_{t=1}^{I} \nabla_\alpha f_{k,m}^s(\mathbf{v}^{e,0}, \alpha^{e,0}).$

$$\tilde{\alpha}^{e+1,0} = \tilde{\alpha}^{e,0} + \frac{\eta}{M} \sum_{k=1}^{K} \sum_{m \in \mathcal{G}^{e,k}} \sum_{t=1}^{I} \left( \nabla_\alpha F_k^s(\mathbf{v}^{e,0}, \alpha^{e,0}; z_{m,t}^{e,k}) + \nabla_\alpha f_k^s(\mathbf{v}^{e,0}, \alpha^{e,0}) \right), \text{ for } t > 0; \tilde{\alpha}_0 = \alpha_0. \tag{37}$$

$$\mathbb{E}_{e,0} \left\langle \nabla_\alpha f(\mathbf{v}^{e,0}, \alpha^{e,0}), \alpha^{e+1,0} - \alpha \right\rangle$$

$$\leq \frac{6\tilde{\eta} K \sigma^2}{MKI} + \frac{6\ell}{KMI} \sum_{k} \sum_{m \in \mathcal{G}^{e,k}} \sum_{t} \mathbb{E}(\|\mathbf{v}^{e,0} - \mathbf{v}_{m,t}^{e,k}\|^2 + \|\alpha^{e,0} - \alpha_{m,t}^{e,k}\|^2) + \frac{\ell}{3} \mathbb{E}\|\alpha^{e+1,0} - \alpha\|^2$$

$$+ \frac{1}{2\eta KI} \mathbb{E}(\|\alpha^{e,0} - \alpha\|^2 - \|\alpha^{e,0} - \alpha^{e+1,0}\|^2 - \|\alpha^{e+1,0} - \alpha\|^2) + \frac{1}{2\eta KI} \mathbb{E}(\|\alpha - \tilde{\alpha}^{e,0}\|^2 - \|\alpha - \tilde{\alpha}^{e+1,0}\|^2).$$

□

With the above lemmas, we are ready to give the convergence in one stage.

### B.3  Convergence Analysis of a Single Stage in CyCp-Minimax

We have the following lemma to bound the convergence for the subproblem in each $s$-th stage.

**Lemma B.5.** *(One call of Algorithm 1) Let* $(\bar{\mathbf{v}}, \bar{\alpha})$ *be the output of Algorithm 1. Suppose Assumption 4.1 hold. By running Algorithm 1 with given input* $\mathbf{v}_0, \alpha_0$ *for* $T$ *iterations,* $\gamma = 2\ell$, *and* $\eta \leq \min(\frac{1}{3\ell+3\ell^2/\mu_2}, \frac{1}{4\ell})$, *we have for any* $\mathbf{v}$ *and* $\alpha$

$$\mathbb{E}[f^s(\bar{\mathbf{v}}, \alpha) - f^s(\mathbf{v}, \bar{\alpha})] \leq \frac{1}{\eta T}\|\mathbf{v}_0 - \mathbf{v}\|^2 + \frac{1}{\eta T}(\alpha_0 - \alpha)^2 + \underbrace{\left( \frac{3\ell^2}{2\mu_2} + \frac{3\ell}{2} \right) 36\eta^2 I^2 D^2}_{A_1} + \frac{3\eta\sigma^2}{K},$$

where $\mu_2 = 2p(1-p)$ is the strong concavity coefficient of $f(\mathbf{v}, \alpha)$ in $\alpha$.

*Proof.* By the updates of $\mathbf{w}$, we obtain

$$
\begin{aligned}
&\mathbb{E}\|\mathbf{v}^{e+1,0} - \mathbf{v}^{e,0}\|^2 = \tilde{\eta}^2 \mathbb{E}\left\|\frac{1}{MI}\sum_{k=1}^{K}\sum_{m\in\mathcal{G}^{e,k}}\sum_t (\nabla_{\mathbf{v}} f^s_{k,m}(\mathbf{v}^{e,k}_{m,t}, \alpha^{e,k}_{m,t}; \mathbf{z}^{e,k}_{m,t}))\right\|^2 \leq \tilde{\eta}^2 K^2 G^2, \\
&\mathbb{E}\|\alpha^{e+1,0} - \alpha^{e,0}\|^2 \leq \tilde{\eta}^2 K^2 G^2, \\
&\mathbb{E}\|\mathbf{v}^{e,k}_{m,t} - \mathbf{v}^{e,0}\|^2 \leq \tilde{\eta}^2 K^2 G^2, \\
&\mathbb{E}\|\alpha^{e,k}_{m,t} - \alpha^{e,0}\|^2 \leq \tilde{\eta}^2 K^2 G^2.
\end{aligned}
\tag{38}
$$

Plugging Lemma 4.3 and Lemma B.4 into Lemma B.3, and taking expectation, we get

$$
\begin{aligned}
&\mathbb{E}[f^s(\bar{\mathbf{v}}, \alpha) - f^s(\mathbf{v}, \bar{\alpha})] \\
&\leq \frac{1}{E}\sum_{e=1}^{E}\mathbb{E}\Bigg[\underbrace{\left(\frac{3\ell + 3\ell^2/\mu_2}{2} - \frac{1}{2\eta KI}\right)\|\mathbf{v}^{e,0} - \mathbf{v}^{e+1,0}\|^2 + \left(2\ell - \frac{1}{2\eta KI}\right)\|\alpha^{e+1,0} - \alpha^{e,0}\|^2}_{C_1} \\
&\quad + \underbrace{\left(\frac{1}{2\eta KI} - \frac{\mu_2}{3}\right)\|\alpha^{e,0} - \alpha\|^2 - \left(\frac{1}{2\eta KI} - \frac{\mu_2}{3}\right)(\alpha^{e+1,0} - \alpha)^2}_{C_2} \\
&\quad + \underbrace{\left(\frac{1}{2\eta KI} - \frac{\ell}{3}\right)\|\mathbf{v}^{e,0} - \mathbf{v}\|^2 - \left(\frac{1}{2\eta KI} - \frac{\ell}{3}\right)\|\mathbf{v}^{e+1,0} - \mathbf{v}\|^2}_{C_3} \\
&\quad + \underbrace{\frac{1}{2\eta KI}((\alpha - \tilde{\alpha}^{e,0})^2 - (\alpha - \tilde{\alpha}^{e+1,0})^2)}_{C_4} + \underbrace{\frac{1}{2\eta KI}(\|\mathbf{v} - \tilde{\mathbf{v}}^{e,0}\|^2 - \|\mathbf{v} - \tilde{\mathbf{v}}^{e+1,0}\|^2)}_{C_5} \\
&\quad + \underbrace{\left(\frac{3\ell^2}{2\mu_2} + \frac{3\ell}{2}\right)\frac{1}{K}\sum_{k=1}^{K}\frac{1}{M}\sum_{m\in\mathcal{G}^{e,k}}\frac{1}{I}\sum_{t=1}^{I}\|\mathbf{v}^{e,0} - \mathbf{v}^{e,k}_{m,t}\|^2 + \left(\frac{3\ell}{2} + \frac{3\ell^2}{2\mu_2}\right)\frac{1}{K}\sum_{k=1}^{K}\frac{1}{M}\sum_{m\in\mathcal{G}^{e,k}}\frac{1}{I}\sum_{t=1}^{I}(\alpha^{e,0} - \alpha^{e,k}_{m,t})^2}_{C_6} \\
&\quad + \frac{3\tilde{\eta}K\sigma^2}{MKI}
\end{aligned}
\tag{39}
$$

Since $\eta \leq \min(\frac{1}{3\ell + 3\ell^2/\mu_2}, \frac{1}{4\ell})$, thus in the RHS of (39), $C_1$ can be canceled. $C_2, C_3, C_4$ and $C_5$ will be handled by telescoping sum. $C_6$ can be bounded by (38).

Taking telescoping sum, it yields

$$
\begin{aligned}
&\mathbb{E}[f^s(\bar{\mathbf{v}}, \alpha) - f^s(\mathbf{v}, \bar{\alpha}) \\
&\leq \frac{1}{4\eta EKI}\|\mathbf{v}_0 - \mathbf{v}\|^2 + \frac{1}{4\eta EKI}\|\alpha_0 - \alpha\|^2 + \left(\frac{3\ell^2}{2\mu_2} + \frac{3\ell}{2}\right)36\eta^2 I^2 K^2 G^2 + \frac{3\eta\sigma^2}{M}.
\end{aligned}
$$

$\square$

### B.4 Main Proof of Theorem 4.5

*Proof.* Since $f(\mathbf{v}, \alpha)$ is $\ell$-smooth (thus $\ell$-weakly convex) in $\mathbf{v}$ for any $\alpha$, $\phi(\mathbf{v}) = \max_{\alpha'} f(\mathbf{v}, \alpha')$ is also $\ell$-weakly convex. Taking $\gamma = 2\ell$, we have

$$
\begin{aligned}
\phi(\mathbf{v}_{s-1}) &\geq \phi(\mathbf{v}_s) + \langle \partial\phi(\mathbf{v}_s), \mathbf{v}_{s-1} - \mathbf{v}_s \rangle - \frac{\ell}{2}\|\mathbf{v}_{s-1} - \mathbf{v}_s\|^2 \\
&= \phi(\mathbf{v}_s) + \langle \partial\phi(\mathbf{v}_s) + 2\ell(\mathbf{v}_s - \mathbf{v}_{s-1}), \mathbf{v}_{s-1} - \mathbf{v}_s \rangle + \frac{3\ell}{2}\|\mathbf{v}_{s-1} - \mathbf{v}_s\|^2 \\
&\overset{(a)}{=} \phi(\mathbf{v}_s) + \langle \partial\phi_s(\mathbf{v}_s), \mathbf{v}_{s-1} - \mathbf{v}_s \rangle + \frac{3\ell}{2}\|\mathbf{v}_{s-1} - \mathbf{v}_s\|^2 \\
&\overset{(b)}{=} \phi(\mathbf{v}_s) - \frac{1}{2\ell}\langle \partial\phi_s(\mathbf{v}_s), \partial\phi_s(\mathbf{v}_s) - \partial\phi(\mathbf{v}_s) \rangle + \frac{3}{8\ell}\|\partial\phi_s(\mathbf{v}_s) - \partial\phi(\mathbf{v}_s)\|^2 \\
&= \phi(\mathbf{v}_s) - \frac{1}{8\ell}\|\partial\phi_s(\mathbf{v}_s)\|^2 - \frac{1}{4\ell}\langle \partial\phi_s(\mathbf{v}_s), \partial\phi(\mathbf{v}_s) \rangle + \frac{3}{8\ell}\|\partial\phi(\mathbf{v}_s)\|^2,
\end{aligned}
\tag{40}
$$

where $(a)$ and $(b)$ hold by the definition of $\phi_s(\mathbf{v})$.

Rearranging the terms in (40) yields

$$
\begin{aligned}
\phi(\mathbf{v}_s) - \phi(\mathbf{v}_{s-1}) &\leq \frac{1}{8\ell}\|\partial\phi_s(\mathbf{v}_s)\|^2 + \frac{1}{4\ell}\langle \partial\phi_s(\mathbf{v}_s), \partial\phi(\mathbf{v}_s) \rangle - \frac{3}{8\ell}\|\partial\phi(\mathbf{v}_s)\|^2 \\
&\overset{(a)}{\leq} \frac{1}{8\ell}\|\partial\phi_s(\mathbf{v}_s)\|^2 + \frac{1}{8\ell}(\|\partial\phi_s(\mathbf{v}_s)\|^2 + \|\partial\phi(\mathbf{v}_s)\|^2) - \frac{3}{8\ell}\|\phi(\mathbf{v}_s)\|^2 \\
&= \frac{1}{4\ell}\|\partial\phi_s(\mathbf{v}_s)\|^2 - \frac{1}{4\ell}\|\partial\phi(\mathbf{v}_s)\|^2 \\
&\overset{(b)}{\leq} \frac{1}{4\ell}\|\partial\phi_s(\mathbf{v}_s)\|^2 - \frac{\mu}{2\ell}(\phi(\mathbf{v}_s) - \phi(\mathbf{v}_*))
\end{aligned}
\tag{41}
$$

where $(a)$ holds by using $\langle \mathbf{a}, \mathbf{b} \rangle \leq \frac{1}{2}(\|\mathbf{a}\|^2 + \|\mathbf{b}\|^2)$, and $(b)$ holds by the $\mu$-PL property of $\phi(\mathbf{v})$.

Thus, we have

$$
(4\ell + 2\mu)(\phi(\mathbf{v}_s) - \phi(\mathbf{v}_*)) - 4\ell(\phi(\mathbf{v}_{s-1}) - \phi(\mathbf{v}_*)) \leq \|\partial\phi_s(\mathbf{v}_s)\|^2.
\tag{42}
$$

Since $\gamma = 2\ell$, $f^s(\mathbf{v}, \alpha)$ is $\ell$-strongly convex in $\mathbf{v}$ and $\mu_2 = 2p(1-p)$ strong concave in $\alpha$. Apply Lemma B.1 to $f^s$, we know that

$$
\frac{\ell}{4}\|\hat{\mathbf{v}}_s(\alpha_s) - \mathbf{v}_0^s\|^2 + \frac{\mu_2}{4}\|\hat{\alpha}_s(\mathbf{v}_s) - \alpha_0^s\|^2 \leq \mathrm{Gap}_s(\mathbf{v}_0^s, \alpha_0^s) + \mathrm{Gap}_s(\mathbf{v}_s, \alpha_s).
\tag{43}
$$

By the setting of $\eta_s = \eta_0 \exp\left(-(s-1)\frac{2\mu}{c+2\mu}\right)$, and $T_s = E_s K I_s = \frac{212}{\eta_0 \min\{\ell, \mu_2\}} \exp\left((s-1)\frac{2\mu}{c+2\mu}\right)$, we note that $\frac{1}{\eta_s T_s} \leq \frac{\min\{\ell, \mu_2\}}{212}$. Set $I_s$ such that $\left(\frac{3\ell^2}{2\mu_2} + \frac{3\ell}{2}\right)36\eta_s^2 I_s^2 K^2 G^2 \leq \frac{\eta_s \sigma^2}{M}$, where the specific choice of $I_s$ will be made later. Applying Lemma B.5 with $\hat{\mathbf{v}}_s(\alpha_s) = \arg\min_{\mathbf{v}'} f^s(\mathbf{v}', \alpha_s)$ and $\hat{\alpha}_s(\mathbf{v}_s) = \arg\max_{\alpha'} f^s(\mathbf{v}_s, \alpha')$, we have

$$
\begin{aligned}
\mathbb{E}[\mathrm{Gap}_s(\mathbf{v}_s, \alpha_s)] &\leq \frac{4\eta_s \sigma^2}{M} + \frac{1}{53}\mathbb{E}\left[\frac{\ell}{4}\|\hat{\mathbf{v}}_s(\alpha_s) - \mathbf{v}_0^s\|^2 + \frac{\mu_2}{4}\|\hat{\alpha}_s(\mathbf{v}_s) - \alpha_0^s\|^2\right] \\
&\leq \frac{4\eta_s \sigma^2}{M} + \frac{1}{53}\mathbb{E}\left[\mathrm{Gap}_s(\mathbf{v}_0^s, \alpha_0^s) + \mathrm{Gap}_s(\mathbf{v}_s, \alpha_s)\right].
\end{aligned}
\tag{44}
$$

Since $\phi(\mathbf{v})$ is $L$-smooth and $\gamma = 2\ell$, then $\phi_s(\mathbf{v})$ is $\hat{L} = (L + 2\ell)$-smooth. According to Theorem 2.1.5 of (Nesterov, 2004), we have

$$
\begin{aligned}
\mathbb{E}[\|\partial\phi_s(\mathbf{v}_s)\|^2] &\leq 2\hat{L}\mathbb{E}(\phi_s(\mathbf{v}_s) - \min_{x\in\mathbb{R}^d}\phi_s(\mathbf{v})) \leq 2\hat{L}\mathbb{E}[\mathrm{Gap}_s(\mathbf{v}_s, \alpha_s)] \\
&= 2\hat{L}\mathbb{E}[4\mathrm{Gap}_s(\mathbf{v}_s, \alpha_s) - 3\mathrm{Gap}_s(\mathbf{v}_s, \alpha_s)] \\
&\leq 2\hat{L}\mathbb{E}\left[4\left(\frac{4\eta_s\sigma^2}{M} + \frac{1}{53}\left(\mathrm{Gap}_s(\mathbf{v}_0^s, \alpha_0^s) + \mathrm{Gap}_s(\mathbf{v}_s, \alpha_s)\right)\right) - 3\mathrm{Gap}_s(\mathbf{v}_s, \alpha_s)\right] \\
&= 2\hat{L}\mathbb{E}\left[\frac{16\eta_s\sigma^2}{M} + \frac{4}{53}\mathrm{Gap}_s(\mathbf{v}_0^s, \alpha_0^s) - \frac{155}{53}\mathrm{Gap}_s(\mathbf{v}_s, \alpha_s)\right].
\end{aligned}
\tag{45}
$$

Applying Lemma B.2 to (45), we have

$$
\begin{aligned}
\mathbb{E}[\|\partial\phi_s(\mathbf{v}_s)\|^2] &\leq 2\hat{L}\mathbb{E}\left[\frac{16\eta_s\sigma^2}{M} + \frac{4}{53}\mathrm{Gap}_s(\mathbf{v}_0^s, \alpha_0^s)\right. \\
&\qquad\qquad\left. - \frac{155}{53}\left(\frac{3}{50}\mathrm{Gap}_{s+1}(\mathbf{v}_0^{s+1}, \alpha_0^{s+1}) + \frac{4}{5}(\phi(\mathbf{v}_0^{s+1}) - \phi(\mathbf{v}_0^s))\right)\right] \\
&= 2\hat{L}\mathbb{E}\left[\frac{16\eta_s\sigma^2}{M} + \frac{4}{53}\mathrm{Gap}_s(\mathbf{v}_0^s, \alpha_0^s) - \frac{93}{530}\mathrm{Gap}_{s+1}(\mathbf{v}_0^{s+1}, \alpha_0^{s+1}) - \frac{124}{53}(\phi(\mathbf{v}_0^{s+1}) - \phi(\mathbf{v}_0^s))\right].
\end{aligned}
\tag{46}
$$

Combining this with (42), rearranging the terms, and defining a constant $c = 4\ell + \frac{248}{53}\hat{L} \in O(L + \ell)$, we get

$$
\begin{aligned}
(c + 2\mu)\,&\mathbb{E}[\phi(\mathbf{v}_0^{s+1}) - \phi(\mathbf{v}_*)] + \frac{93}{265}\hat{L}\mathbb{E}[\mathrm{Gap}_{s+1}(\mathbf{v}_0^{s+1}, \alpha_0^{s+1})] \\
&\leq \left(4\ell + \frac{248}{53}\hat{L}\right)\mathbb{E}[\phi(\mathbf{v}_0^s) - \phi(\mathbf{v}_*)] + \frac{8\hat{L}}{53}\mathbb{E}[\mathrm{Gap}_s(\mathbf{v}_0^s, \alpha_0^s)] + \frac{32\eta_s\hat{L}\sigma^2}{M} \\
&\leq c\mathbb{E}\left[\phi(\mathbf{v}_0^s) - \phi(\mathbf{v}_*) + \frac{8\hat{L}}{53c}\mathrm{Gap}_s(\mathbf{v}_0^s, \alpha_0^s)\right] + \frac{32\eta_s\hat{L}\sigma^2}{M}.
\end{aligned}
\tag{47}
$$

Using the fact that $\hat{L} \geq \mu$,

$$
(c + 2\mu)\frac{8\hat{L}}{53c} = \left(4\ell + \frac{248}{53}\hat{L} + 2\mu\right)\frac{8\hat{L}}{53(4\ell + \frac{248}{53}\hat{L})} \leq \frac{8\hat{L}}{53} + \frac{16\mu\hat{L}}{248\hat{L}} \leq \frac{93}{265}\hat{L}.
\tag{48}
$$

Then, we have

$$
\begin{aligned}
(c + 2\mu)\mathbb{E}\left[\phi(\mathbf{v}_0^{s+1}) - \phi(\mathbf{v}_*) + \frac{8\hat{L}}{53c}\mathrm{Gap}_{s+1}(\mathbf{v}_0^{s+1}, \alpha_0^{s+1})\right] \\
\leq c\mathbb{E}\left[\phi(\mathbf{v}_0^s) - \phi(\mathbf{v}_*) + \frac{8\hat{L}}{53c}\mathrm{Gap}_s(\mathbf{v}_0^s, \alpha_0^s)\right] + \frac{32\eta_s\hat{L}\sigma^2}{M}.
\end{aligned}
\tag{49}
$$

Defining $\Delta_s = \phi(\mathbf{v}_0^s) - \phi(\mathbf{v}_*) + \frac{8\hat{L}}{53c}\mathrm{Gap}_s(\mathbf{v}_0^s, \alpha_0^s)$, then

$$
\mathbb{E}[\Delta_{s+1}] \leq \frac{c}{c + 2\mu}\mathbb{E}[\Delta_s] + \frac{32\eta_s\hat{L}\sigma^2}{(c + 2\mu)M}
\tag{50}
$$

Using this inequality recursively, it yields

$$
E[\Delta_{S+1}] \leq \left(\frac{c}{c + 2\mu}\right)^S E[\Delta_1] + \frac{32\hat{L}\sigma^2}{(c + 2\mu)M}\sum_{s=1}^{S}\left(\eta_s\left(\frac{c}{c + 2\mu}\right)^{S-s}\right).
\tag{51}
$$

By definition,

$$
\begin{aligned}
\Delta_1 &= \phi(\mathbf{v}_0^1) - \phi(\mathbf{v}^*) + \frac{8\hat{L}}{53c}\widehat{Gap}_1(\mathbf{v}_0^1, \alpha_0^1) \\
&= \phi(\mathbf{v}_0) - \phi(\mathbf{v}^*) + \frac{8\hat{L}}{53c}\left( f(\mathbf{v}_0, \hat{\alpha}_1(\mathbf{v}_0)) + \frac{\gamma}{2}\|\mathbf{v}_0 - \mathbf{v}_0\|^2 - f(\hat{\mathbf{v}}_1(\alpha_0), \alpha_0) - \frac{\gamma}{2}\|\hat{\mathbf{v}}_1(\alpha_0) - \mathbf{v}_0\|^2 \right) \\
&\leq \epsilon_0 + \frac{8\hat{L}}{53c}\left( f(\mathbf{v}_0, \hat{\alpha}_1(\mathbf{v}_0)) - f(\hat{\mathbf{v}}(\alpha_0), \alpha_0) \right) \leq 2\epsilon_0.
\end{aligned}
\tag{52}
$$

Using inequality $1 - x \leq \exp(-x)$, we have

$$
\begin{aligned}
\mathbb{E}[\Delta_{S+1}] &\leq \exp\left( \frac{-2\mu S}{c+2\mu} \right)\mathbb{E}[\Delta_1] + \frac{32\eta_0\hat{L}\sigma^2}{(c+2\mu)M}\sum_{s=1}^S \exp\left( -\frac{2\mu S}{c+2\mu} \right) \\
&\leq 2\epsilon_0 \exp\left( \frac{-2\mu S}{c+2\mu} \right) + \frac{32\eta_0\hat{L}\sigma^2}{(c+2\mu)M}S\exp\left( -\frac{2\mu S}{(c+2\mu)} \right).
\end{aligned}
$$

To make this less than $\epsilon$, it suffices to make

$$
\begin{aligned}
2\epsilon_0\exp\left( \frac{-2\mu S}{c+2\mu} \right) &\leq \frac{\epsilon}{2}, \\
\frac{32\eta_0\hat{L}\sigma^2}{(c+2\mu)M}S\exp\left( -\frac{2\mu S}{c+2\mu} \right) &\leq \frac{\epsilon}{2}.
\end{aligned}
\tag{53}
$$

Let $S$ be the smallest value such that $\exp\left( \frac{-2\mu S}{c+2\mu} \right) \leq \min\{\frac{\epsilon}{4\epsilon_0}, \frac{(c+2\mu)M\epsilon}{64\eta_0\hat{L}S\sigma^2}\}$. We can set $S = \max\left\{ \frac{c+2\mu}{2\mu}\log\frac{4\epsilon_0}{\epsilon}, \frac{c+2\mu}{2\mu}\log\frac{64\eta_0\hat{L}S\sigma^2}{(c+2\mu)M\epsilon} \right\}$.

Then, the total iteration complexity is

$$
\begin{aligned}
\sum_{s=1}^S T_s &\leq O\left( \frac{424}{\eta_0\min\{\ell, \mu_2\}}\sum_{s=1}^S \exp\left( (s-1)\frac{2\mu}{c+2\mu} \right) \right) \\
&\leq O\left( \frac{1}{\eta_0\min\{\ell, \mu_2\}}\frac{\exp(S\frac{2\mu}{c+2\mu}) - 1}{\exp(\frac{2\mu}{c+2\mu}) - 1} \right) \\
&\overset{(a)}{\leq} \widetilde{O}\left( \frac{c}{\eta_0\mu\min\{\ell, \mu_2\}}\max\left\{ \frac{\epsilon_0}{\epsilon}, \frac{\eta_0\hat{L}S\sigma^2}{(c+2\mu)M\epsilon} \right\} \right) \\
&\leq \widetilde{O}\left( \max\left\{ \frac{(L+\ell)\epsilon_0}{\eta_0\mu\min\{\ell, \mu_2\}\epsilon}, \frac{(L+\ell)^2\sigma^2}{\mu^2\min\{\ell, \mu_2\}M\epsilon} \right\} \right) \\
&\leq \widetilde{O}\left( \max\left\{ \frac{1}{\mu_1\mu_2^2\epsilon}, \frac{1}{\mu_1^2\mu_2^3M\epsilon} \right\} \right),
\end{aligned}
\tag{54}
$$

where $(a)$ uses the setting of $S$ and $\exp(x) - 1 \geq x$, and $\widetilde{O}$ suppresses logarithmic factors.

$\eta_s = \eta_0\exp(-(s-1)\frac{2\mu}{c+2\mu})$, $T_s = \frac{212}{\eta_0\mu_2}\exp\left( (s-1)\frac{2\mu}{c+2\mu} \right)$.

Next, we will analyze the communication cost.

To assure $\left( \frac{3\ell^2}{2\mu_2} + \frac{3\ell}{2} \right)36\eta^2 I_s^2 K^2 G^2 \leq \frac{\eta_s\sigma^2}{M}$ which we used in above proof, we need to take $I_s = O(\frac{1}{K\sqrt{\eta_s M}})$.

If $\frac{1}{K\sqrt{\eta_0 M}} \leq O(1)$, for $s \leq S_2 := O(\frac{c+2\mu}{2\mu}\log(M\eta_0))$, we take then $I_s = 1$ and correspondingly $E_s = T_s/KI_s = T_s/K$. For $s > S_2$, $I_s = O(\frac{\exp((s-1)\frac{\mu}{c+2\mu})}{K(\eta_0 M)^{1/2}})$, and correspondingly $E_s = T_s/KI_s = O(KM^{1/2}\exp((s-1)\frac{\mu}{c+2\mu}))$.

We have

$$\sum_{s=1}^{S_2} T_s = \sum_{s=1}^{S_2} O\left(\frac{212}{\eta_0}\exp\left((s-1)\frac{2\mu}{c+2\mu}\right)\right) = \widetilde{O}\left(\frac{M}{\mu}\right). \tag{55}$$

Thus, the communication complexity can be bounded by

$$\sum_{s=1}^{S_2} T_s + \sum_{s=S_2+1}^{S} \frac{T_s}{I_s} = \widetilde{O}\left(\frac{K}{\mu} + \sqrt{M}K\exp\left(\frac{(s-1)\frac{2\mu}{c+2\mu}}{2}\right)\right) \tag{56}$$

$$\leq \widetilde{O}(\frac{M}{\mu} + \sqrt{M}K\frac{\exp\left(\frac{S}{2}\frac{2\mu}{c+2\mu}\right)-1}{\exp\frac{\mu}{c+2\mu}-1}) \leq O\left(\frac{M}{\mu} + \frac{K}{\mu^{3/2}\epsilon^{1/2}}\right).$$

$$\qquad\qquad\qquad\qquad\qquad\qquad\qquad\qquad\qquad\qquad\qquad\qquad\qquad\qquad\qquad\qquad\qquad\qquad\square$$

## C   Analysis of CyCp-FedX

In this section, we show the analysis of Theorem 5.2.

*Proof.* We denote

$$\begin{aligned}
G_1(\mathbf{w},\mathbf{z},\mathbf{w}',\mathbf{z}') &= \nabla_1\psi(h(\mathbf{w};\mathbf{z}),h(\mathbf{w};\mathbf{z}'))^\top \nabla h(\mathbf{w};\mathbf{z}) \\
G_2(\mathbf{w};\mathbf{z},\mathbf{w}';\mathbf{z}') &= \nabla_2\psi(h(\mathbf{w};\mathbf{z}),h(\mathbf{w};\mathbf{z}'))^\top \nabla h(\mathbf{w};\mathbf{z}') \\
G_{m,t,1}^{e,k} &= \nabla_1\psi(h(\mathbf{w}_{m,t}^{e,k},\mathbf{z}_{m,t,1}^{e,k}),h_{2,\xi})\nabla h(\mathbf{w}_{m,t}^{e,k},\mathbf{z}_{m,t,1}^{e,k}) \\
G_{m,t,2}^{e,k} &= \nabla_2\psi(h_{1,\xi},h(\mathbf{w}_{m,t}^{e,k},\mathbf{z}_{m,t,2}^{e,k}))\nabla h(\mathbf{w}_{m,t}^{e,k},\mathbf{z}_{m,t,2}^{e,k})
\end{aligned} \tag{57}$$

With $\tilde{\eta}=\eta I$,

$$F(\mathbf{w}^{e+1,0}) - F(\mathbf{w}^{e,0}) \leq \nabla F(\mathbf{w}^{e,0})^\top(\mathbf{w}^{e+1,0}-\mathbf{w}^{e,0}) + \frac{L}{2}\|\mathbf{w}^{e+1,0}-\mathbf{w}^{e,0}\|^2$$

$$= -\tilde{\eta}\nabla F(\mathbf{w}^{e,0})^\top \frac{1}{MI}\sum_{k=1}^{K}\sum_{m\in\mathcal{G}^{e,k}}\sum_{t=1}^{I}(G_{m,t,1}^{e,k}+G_{m,t,2}^{e,k}) + \frac{L}{2}\|\mathbf{w}^{e+1,0}-\mathbf{w}^{e,0}\|^2$$

$$= -\tilde{\eta}(\nabla F(\mathbf{w}^{e,0})-\nabla F(\mathbf{w}^{e-1,0})+\nabla F(\mathbf{w}^{e-1,0}))^\top \frac{1}{MI}\sum_{k=1}^{K}\sum_{m\in\mathcal{G}^{e,k}}\sum_{t=1}^{I}(G_{m,t,1}^{e,k}+G_{m,t,2}^{e,k}) + \frac{L}{2}\|\mathbf{w}^{e+1,0}-\mathbf{w}^{e,0}\|^2$$

$$\leq \frac{1}{2L}\|\nabla F(\mathbf{w}^{e,0})-\nabla F(\mathbf{w}^{e-1,0})\|^2 + 2\tilde{\eta}^2 L\|\frac{1}{MI}\sum_{k=1}^{K}\sum_{m\in\mathcal{G}^{e,k}}\sum_{t=1}^{I}(G_{m,t,1}^{e,k}+G_{m,t,2}^{e,k})\|^2$$

$$- \tilde{\eta}\nabla F(\mathbf{w}^{e-1,0})^\top\left(\frac{1}{MI}\sum_{k=1}^{K}\sum_{m\in\mathcal{G}^{e,k}}\sum_{t=1}^{I}(G_{m,t,1}^{e,k}+G_{m,t,2}^{e,k})\right) + \frac{L}{2}\|\mathbf{w}^{e+1,0}-\mathbf{w}^{e,0}\|^2. \tag{58}$$

Denoting $k', k'', m', m'', t', t''$ as random variables corresponding to be the indexes that used passive parts at $k, m, t$,

$$
-\mathbb{E}\left[\tilde{\eta}\nabla F(\mathbf{w}^{e-1,0})^\top\left(\frac{1}{MI}\sum_{k=1}^K\sum_{m\in\mathcal{G}^{e,k}}\sum_{t=1}^I(G_{m,t,1}^{e,k}+G_{m,t,2}^{e,k})\right)\right]
$$

$$
= -\mathbb{E}\left[\tilde{\eta}\nabla F(\mathbf{w}^{e-1,0})^\top\frac{1}{MI}\sum_{k=1}^K\sum_{m\in\mathcal{G}^{e,k}}\sum_{t=1}^I\left(G_1(\mathbf{w}_{m,t}^{e,k},\mathbf{z}_{m,t,1}^{e,k},\mathbf{w}_{t'}^{e-1,m'},\mathbf{z}_{m',t',2}^{e-1,k'})+G_2(\mathbf{w}_{t''}^{e-1,c''},\mathbf{z}_{m'',t'',1}^{e-1,k''},\mathbf{w}_{m,t}^{e,k},\mathbf{z}_{m,t,2}^{e,k})\right.\right.
$$

$$
- G_1(\mathbf{w}^{e-1,0},\mathbf{z}_{m,t,1}^{e,k},\mathbf{w}^{e-1,0},\mathbf{z}_{m',t',2}^{e-1,k'})-G_2(\mathbf{w}^{e-1,0},\mathbf{z}_{m'',t'',1}^{e-1,k''},\mathbf{w}^{e-1,0},\mathbf{z}_{m,t,2}^{e,k})
$$

$$
\left.\left.+ G_1(\mathbf{w}^{e-1,0},\mathbf{z}_{m,t,1}^{e,k},\mathbf{w}^{e-1,0},\mathbf{z}_{m',t',2}^{e-1,k'})+G_2(\mathbf{w}^{e-1,0},\mathbf{z}_{m'',t'',1}^{e-1,k''},\mathbf{w}^{e-1,0},\mathbf{z}_{m,t,2}^{e,k})\right)\right]
$$

$$
\overset{(a)}{\leq} 4\tilde{\eta}KL^2\frac{1}{K}\sum_{k=1}^K\frac{1}{MI}\sum_{m\in\mathcal{G}^{e,k}}\sum_{t=1}^I\mathbb{E}(2\|\mathbf{w}_{m,t}^{e,k}-\mathbf{w}^{e-1,0}\|^2)+4\tilde{\eta}KL^2\frac{1}{K}\frac{1}{MI}\sum_{k'}\sum_{m'\in\mathcal{S}^{e-1,k'}}\sum_{t'}(\|\mathbf{w}_{m',t'}^{e-1,k'}-\mathbf{w}^{e-1,0}\|^2)
$$

$$
+\frac{\tilde{\eta}K}{4}\mathbb{E}\|\nabla F(\mathbf{w}^{e-1,0})\|^2-\mathbb{E}\left[\tilde{\eta}K\nabla F(\mathbf{w}^{e-1,0})^\top\left(\frac{1}{K}\sum_{k\in[K]}\nabla F_k(\mathbf{w}^{e-1,0})\right)\right]
$$

$$
\leq 16\tilde{\eta}KL^2\|\mathbf{w}^{e,0}-\mathbf{w}^{e-1,0}\|^2+16\tilde{\eta}KL^2\frac{1}{KMI}\sum_k\sum_{m\in\mathcal{G}^{e,k}}\sum_t\|\mathbf{w}_{m,t}^{e,k}-\mathbf{w}^{e,0}\|^2
$$

$$
+8\tilde{\eta}KL^2\frac{1}{KMI}\sum_{k'}\sum_{m'\in\mathcal{S}^{e-1,k'}}\sum_t\|\mathbf{w}_{m',t'}^{e-1,k'}-\mathbf{w}^{e-1,0}\|^2-\frac{\tilde{\eta}K}{2}\mathbb{E}\|\nabla F(\mathbf{w}^{e-1,0})\|^2,
$$

$$(59)$$

where the (a) holds because

$$
\mathbb{E}\left[G_1(\mathbf{w}^{e-1,0},\mathbf{z}_{m,t,1}^{e,k},\mathbf{w}^{e-1,0},\mathbf{z}_{m',t',2}^{e-1,k'})+G_2(\mathbf{w}^{e-1,0},\mathbf{z}_{m'',t'',1}^{e-1,k''},\mathbf{w}^{e-1,0},\mathbf{z}_{m,t,2}^{e,k})\right)\right]=\nabla F(\mathbf{w}^{e-1,0}).\qquad(60)
$$

By the updates of $\mathbf{w}$, we obtain

$$
\mathbb{E}\|\mathbf{w}^{e+1,0}-\mathbf{w}^{e,0}\|^2=\tilde{\eta}^2\left\|\frac{1}{MI}\sum_{k=1}^K\sum_{m\in\mathcal{G}^{e,k}}\sum_t(G_{m,t,1}^{e,k}+G_{m,t,2}^{e,k})\right\|^2
$$

$$
=\tilde{\eta}^2\mathbb{E}\left\|\frac{1}{MI}\sum_{k=1}^K\sum_{m\in\mathcal{G}^{e,k}}\sum_t(G_1(\mathbf{w}_{m,t}^{e,k},\mathbf{z}_{m,t,1}^{e,k},\mathbf{w}_{m',t'}^{e-1,k''},\mathbf{z}_{m',t',2}^{e-1,k'})+G_2(\mathbf{w}_{m'',t''}^{e-1,k''},\mathbf{z}_{m'',t'',1}^{e-1,k''},\mathbf{w}_{m,t}^{e,k},\mathbf{z}_{m,t,2}^{e,k}))\right\|^2
$$

$$
\leq 3\tilde{\eta}^2\mathbb{E}\left\|\frac{1}{MI}\sum_{k=1}^K\sum_{m\in\mathcal{G}^{e,k}}\sum_t(G_1(\mathbf{w}_{m,t}^{e,k},\mathbf{z}_{m,t,1}^{e,k},\mathbf{w}_{m',t'}^{e-1,k''},\mathbf{z}_{m',t',2}^{e-1,k'})+G_2(\mathbf{w}_{m'',t''}^{e-1,k''},\mathbf{z}_{m'',t'',1}^{e-1,k''},\mathbf{w}_{m,t}^{e,k},\mathbf{z}_{m,t,2}^{e,k}))\right.
$$

$$
\left.-\frac{1}{MI}\sum_{k=1}^K\sum_{m\in\mathcal{G}^{e,k}}\sum_t(G_1(\mathbf{w}^{e-1,0},\mathbf{z}_{m,t,1}^{e,k},\mathbf{w}^{e-1,0},\mathbf{z}_{m',t',2}^{e-1,k'})+G_2(\mathbf{w}^{e-1,0},\mathbf{z}_{m'',t'',1}^{e-1,k''},\mathbf{w}^{e-1,0},\mathbf{z}_{m,t,2}^{e,k}))\right\|^2
$$

$$
+3\tilde{\eta}^2\mathbb{E}\left\|\frac{1}{MI}\sum_{k=1}^K\sum_{m\in\mathcal{G}^{e,k}}\sum_t(G_1(\mathbf{w}^{e-1,0},\mathbf{z}_{m,t,1}^{e,k},\mathbf{w}^{e-1,0},\mathbf{z}_{t',2}^{e-1,k'})+G_2(\mathbf{w}^{e-1,0},\mathbf{z}_{m'',t'',1}^{e-1,k''},\mathbf{w}^{e-1,0},\mathbf{z}_{m,t,2}^{e,k}))-\nabla F(\mathbf{w}^{e-1,0})\right\|^2
$$

$$
+3\tilde{\eta}^2K^2\mathbb{E}\|\nabla F(\mathbf{w}^{e-1,0})\|^2,
$$

$$(61)$$

which leads to

$$\mathbb{E}\|\mathbf{w}^{e+1,0} - \mathbf{w}^{e,0}\|^2$$

$$\leq 6\tilde{\eta}^2 K^2 \frac{\tilde{L}^2}{KMI} \sum_{k=1}^{K} \sum_{m \in \mathcal{G}^{e,k}} \sum_{t} \mathbb{E}\|\mathbf{w}_{m,t}^{e,k} - \mathbf{w}^{e,0}\|^2 + 6\tilde{\eta}^2 K^2 \frac{\tilde{L}^2}{KMI} \sum_{k=1}^{K} \sum_{m \in \mathcal{G}^{e,k}} \sum_{t} \mathbb{E}\|\mathbf{w}_{m',t'}^{e-1,k'} - \mathbf{w}^{e-1,0}\|^2$$

$$+ 6\tilde{\eta}^2 K^2 \frac{\tilde{L}^2}{KMI} \sum_{k=1}^{K} \sum_{m \in \mathcal{G}^{e,k}} \sum_{t} \mathbb{E}\|\mathbf{w}^{e,0} - \mathbf{w}^{e-1,0}\|^2$$

$$+ 3\tilde{\eta}^2 K^2 \mathbb{E}\left\| \frac{1}{KMI} \sum_{k=1}^{K} \sum_{m \in \mathcal{G}^{e,k}} \sum_{t} (G_1(\mathbf{w}^{e-1,0}, \mathbf{z}_{m,t,1}^{e,k}, \mathbf{w}^{e-1,0}, \mathbf{z}_{m',t',2}^{e-1,k'}) + G_2(\mathbf{w}^{e-1,0}, \mathbf{z}_{m'',t'',1}^{e-1,k''}, \mathbf{w}^{e,0}, \mathbf{z}_{m,t,2}^{e,k})) - \nabla F_k(\mathbf{w}^{e-1,0}) \right\|^2$$

$$+ 3\tilde{\eta}^2 K^2 \mathbb{E}\|\nabla F(\mathbf{w}^{e-1,0})\|^2.$$

$$(62)$$

Therefore,

$$\frac{1}{E} \sum_{e=1}^{E} \mathbb{E}\|\mathbf{w}^{e+1,0} - \mathbf{w}^{e,0}\|^2$$

$$\leq \frac{1}{E} \sum_{e=1}^{E} \left[ 6\tilde{\eta}^2 K^2 \frac{\tilde{L}^2}{KMI} \sum_{k=1}^{K} \sum_{m \in \mathcal{G}^{e,k}} \sum_{t} \mathbb{E}\|\mathbf{w}_{m,t}^{e,k} - \mathbf{w}^{e,0}\|^2 + 6\tilde{\eta}^2 K^2 \frac{\tilde{L}^2}{KMI} \sum_{k=1}^{K} \sum_{m \in \mathcal{G}^{e,k}} \sum_{t} \mathbb{E}\|\mathbf{w}_{m,t}^{e-1,k} - \mathbf{w}^{e-1,0}\|^2 \right.$$

$$\left. + 6\tilde{\eta}^2 K^2 \frac{\tilde{L}^2}{KMI} \sum_{k=1}^{K} \sum_{m \in \mathcal{G}^{e,k}} \sum_{t} \mathbb{E}\|\mathbf{w}_{m',t'}^{e-1,k'} - \mathbf{w}^{e-1,0}\|^2 + 6\tilde{\eta}^2 K^2 \frac{\sigma^2}{KMI} + 3\tilde{\eta}^2 K^2 \mathbb{E}\|\nabla F(\mathbf{w}^{e-1,0})\|^2 \right].$$

$$(63)$$

$$\|\mathbf{w}_{m,t}^{e,k} - \mathbf{w}^{e,0}\|^2 = \tilde{\eta}^2 \left\| \frac{1}{I} \sum_{\tau=1}^{k} \sum_{\nu=1}^{t_\tau} (G_{m,\nu,1}^{e,\tau} + G_{m,\nu,2}^{e,\tau}) \right\|^2$$

$$= \tilde{\eta}^2 \mathbb{E} \left\| \frac{1}{I} \sum_{\tau=1}^{k} \sum_{\nu=1}^{t_\tau} (G_1(\mathbf{w}_{m,t}^{e,\tau}, \mathbf{z}_{m,t,1}^{e,\tau}, \mathbf{w}_{m',t'}^{e-1,\tau'}, \mathbf{z}_{m',t',2}^{e-1,\tau'}) + G_2(\mathbf{w}_{m'',t''}^{e-1,\tau''}, \mathbf{z}_{m'',t'',1}^{e-1,\tau''}, \mathbf{w}_{m,t}^{e,\tau}, \mathbf{z}_{m,t,2}^{e,\tau})) \right\|^2$$

$$\leq 4\tilde{\eta}^2 \mathbb{E} \left\| \frac{1}{I} \sum_{\tau=1}^{k} \sum_{\nu=1}^{t_\tau} (G_1(\mathbf{w}_{m,t}^{e,\tau}, \mathbf{z}_{m,t,1}^{e,\tau}, \mathbf{w}_{m',t'}^{e-1,\tau'}, \mathbf{z}_{m',t',2}^{e-1,\tau'}) + G_2(\mathbf{w}_{m'',t''}^{e-1,\tau''}, \mathbf{z}_{m'',t'',1}^{e-1,\tau''}, \mathbf{w}_{m,t}^{e,\tau}, \mathbf{z}_{m,t,2}^{e,\tau})) \right.$$

$$\left. - \frac{1}{I} \sum_{\tau=1}^{k} \sum_{\nu=1}^{t_\tau} (G_1(\mathbf{w}^{e-1,0}, \mathbf{z}_{m,t,1}^{e,\tau}, \mathbf{w}^{e-1,0}, \mathbf{z}_{m',t',2}^{e-1,\tau'}) + G_2(\mathbf{w}^{e-1,0}, \mathbf{z}_{m'',t'',1}^{e-1,\tau''}, \mathbf{w}^{e-1,0}, \mathbf{z}_{m,t,2}^{e,\tau})) \right\|^2$$

$$+ 4\tilde{\eta}^2 \mathbb{E} \left\| \frac{1}{I} \left[ \sum_{\tau=1}^{k} \sum_{\nu=1}^{t_\tau} (G_1(\mathbf{w}^{e-1,0}, \mathbf{z}_{m,t,1}^{e,\tau}, \mathbf{w}^{e-1,0}, \mathbf{z}_{t',2}^{e-1,\tau'}) + G_2(\mathbf{w}^{e-1,0}, \mathbf{z}_{m'',t'',1}^{e-1,\tau''}, \mathbf{w}^{e-1,0}, \mathbf{z}_{m,t,2}^{e,\tau})) - \nabla F_\tau(\mathbf{w}^{e-1,0}) \right] \right\|^2$$

$$+ 4\tilde{\eta}^2 \mathbb{E}\| \frac{1}{I} \sum_{\tau=1}^{k} \sum_{\nu=1}^{t_\tau} (\nabla F_\tau(\mathbf{w}^{e-1,0}) - \nabla F(\mathbf{w}^{e-1,0}))\|^2 + 4\tilde{\eta}^2 \mathbb{E}\| \frac{1}{I} \sum_{\tau=1}^{k} \sum_{\nu=1}^{t_\tau} \nabla F(\mathbf{w}^{e-1,0})\|^2$$

$$\leq 8\tilde{\eta}^2 \tilde{L}^2 \frac{K}{I} \sum_{\tau=1}^{k} \sum_{\nu=1}^{t_\tau} \|\mathbf{w}_{m,t}^{e,\tau} - \mathbf{w}^{e,0}\|^2 + 8\tilde{\eta}^2 \tilde{L}^2 \frac{K}{I} \sum_{\tau=1}^{k} \sum_{\nu=1}^{t_\tau} \|\mathbf{w}^{e,0} - \mathbf{w}^{e-1,0}\|^2$$

$$+ 4\tilde{\eta}^2 k^2 \frac{\sigma^2}{kI} + 4\tilde{\eta}^2 k^2 D^2 + 4\tilde{\eta}^2 k^2 \mathbb{E}\|\nabla F(\mathbf{w}^{e-1,0})\|^2.$$

$$(64)$$

$$\frac{1}{EKMI}\sum_e\sum_k\sum_{m\in\mathcal{G}^{e,k}}\sum_t\|\mathbf{w}_{m,t}^{e,k}-\mathbf{w}^{e,0}\|^2$$

$$\leq 8\tilde{\eta}^2\tilde{L}^2\frac{K}{I}\frac{1}{EKMI}\sum_{e=1}^{E}\sum_{\tau=1}^{K}\sum_{m\in\mathcal{S}^{e,\tau}}\sum_t KI\|\mathbf{w}_{m,t}^{e,\tau}-\mathbf{w}^{e,0}\|^2+8\tilde{\eta}^2\tilde{L}^2\frac{K}{I}\frac{1}{EKMI}\sum_{e=1}^{E}\sum_{\tau=1}^{K}\sum_{m\in\mathcal{S}^{e,\tau}}\sum_t KI\|\mathbf{w}^{e,0}-\mathbf{w}^{e-1,0}\|^2$$

$$+4\tilde{\eta}^2K\frac{\sigma^2}{I}+4\tilde{\eta}^2K^2D^2+4\tilde{\eta}^2K^2\frac{1}{E}\sum_{e=1}^{E}\mathbb{E}\|\nabla F(\mathbf{w}^{e-1,0})\|^2. \tag{65}$$

$$\frac{1}{EKMI}\sum_e\sum_k\sum_{m\in\mathcal{G}^{e,k}}\sum_t\|\mathbf{w}_{m,t}^{e,k}-\mathbf{w}^{e,0}\|^2$$

$$\leq 16\tilde{\eta}^2\tilde{L}^2K^2\frac{1}{E}\sum_{e=1}^{E}\|\mathbf{w}^{e,0}-\mathbf{w}^{e-1,0}\|^2+8\tilde{\eta}^2K\frac{\sigma^2}{I}+8\tilde{\eta}^2K^2\alpha^2+8\tilde{\eta}^2K^2\mathbb{E}\|\nabla F(\mathbf{w}^{e-1,0})\|^2. \tag{66}$$

Using $\tilde{\eta}\tilde{L}K\leq O(1)$,

$$\frac{1}{EKMI}\sum_e\sum_k\sum_{m\in\mathcal{G}^{e,k}}\sum_t\|\mathbf{w}_{m,t}^{e,k}-\mathbf{w}^{e,0}\|^2\leq 16\tilde{\eta}^2K\frac{\sigma^2}{I}+16\tilde{\eta}^2K^2\alpha^2+16\tilde{\eta}^2K^2\|\nabla F(\mathbf{w}^{e-1,0})\|^2 \tag{67}$$

and

$$\frac{1}{E}\sum_{e=1}^{E}\mathbb{E}\|\mathbf{w}^{e+1}-\mathbf{w}^e\|^2\leq 16\tilde{\eta}^2K\frac{\sigma^2}{MI}+16\tilde{\eta}^2K^2\|\nabla F(\mathbf{w}^{e-1,0})\|^2. \tag{68}$$

Plugging these bounds into (58) and (59),

$$F(\mathbf{w}^{e+1,0})-F(\mathbf{w}^{e,0})\leq\tilde{L}^2\|\mathbf{w}^{e,0}-\mathbf{w}^{e-1,0}\|^2+\tilde{\eta}^2L^2K\|\mathbf{w}^{e,0}-\mathbf{w}^{e-1,0}\|^2$$

$$+16\tilde{\eta}^2\tilde{L}^2\frac{1}{MI}\sum_{m\in\mathcal{G}^{e,k}}\sum_t\|\mathbf{w}_{m,t}^{e,k}-\mathbf{w}^{e,0}\|^2-\frac{\tilde{\eta}K}{2}\mathbb{E}\|\nabla F(\mathbf{w}^{e-1,0})\|^2. \tag{69}$$

$$\frac{1}{E}\sum_{e=1}^{E}\mathbb{E}\|\nabla F(\mathbf{w}^{e-1})\|^2\leq O\left(\frac{F(\mathbf{w}^{e,0})-F(\mathbf{w}_*)}{\tilde{\eta}KE}+\tilde{\eta}^2K\frac{\sigma^2}{I}+\tilde{\eta}^2K^2D^2+24\tilde{\eta}K\frac{\sigma^2}{KMI}\right). \tag{70}$$

Since $\tilde{\eta}=\eta I$. Set $\eta=O(\epsilon^2M)$, $I=O(\frac{1}{MK\epsilon})$, $E=\frac{1}{\epsilon^3}$), iteration complexity is $EKI=O(\frac{1}{M\epsilon^4})$, and communication cost is $EK=O(\frac{K}{\epsilon^3})$. □

## D  Analysis of CyCp-FedX with PL condition

Below we show the proof of Theorem 5.3.

*Proof.* Using the property of PL condition, we have

$$F(\mathbf{w}^{s+1,0})-F(\mathbf{w}_*)\leq\frac{1}{2\mu}\|\nabla F(\mathbf{w}^{s+1,0})\|^2 \tag{71}$$

To ensure $F(\mathbf{w}^{s+1,0}) - F(\mathbf{w}_*) \leq \epsilon_s$, $\|\nabla F(\mathbf{w}^{s+1,0})\|^2 \leq \mu\epsilon_s$. Let $\epsilon_0 = \max(F(\mathbf{w}^{0,0}) - F(\mathbf{w}_*), \|\nabla F(\mathbf{w}^{0,0})\|^2/\mu)$. We need

$$\frac{F(\mathbf{w}^{s,0}) - F(\mathbf{w}_*)}{\tilde{\eta}KE} \leq \frac{\mu\epsilon_s}{3}$$
$$\tilde{\eta}^2 K^2 \alpha^2 \leq \frac{\mu\epsilon_s}{3} \tag{72}$$
$$\tilde{\eta}K\frac{\sigma^2}{KMI} \leq \frac{\mu\epsilon_s}{3},$$

where

$$\tag{73}$$

Let $\eta_s = O(\frac{\mu\epsilon_s M}{\sigma^2})$, $I_s = O(\frac{\sigma^2}{MK\sqrt{\mu\epsilon_s}})$, $E_s = (\frac{1}{\mu^{3/2}\epsilon_s^{1/2}})$. The number of iterations in each stage is $I_s E_s = \frac{\sigma^2}{MK\mu^2\epsilon_s}$. To ensure $\epsilon_S \leq \epsilon$, total number of stage is $\log(\mu\epsilon)$. Thus, the total iteration complexity is

$$\sum_{s=1}^{S} I_s E_s = O(\sum_{s=1}^{S} \frac{1}{MK\mu^2\epsilon_s}) = O(\frac{1}{MK\mu^2\epsilon}). \tag{74}$$

Total communication complexity is

$$\sum_{s=1}^{S} E_s K = O(\sum_{s=1}^{S} \frac{1}{\mu^{3/2}\epsilon_s^{1/2}}K) = O(\frac{K}{\mu^{3/2}\epsilon^{1/2}}). \tag{75}$$

$\square$

## E  Data Statistics

| Dataset | Split | Positive Samples | Negative Samples | Positive % |
|---------|-------|------------------|------------------|------------|
| CIFAR-10 | Training | 224 | 40,505 | 0.55% |
| | Validation | 505 | 4,495 | 10.10% |
| | Test | 1,000 | 9,000 | 10.00% |
| CIFAR-100 | Training | 227 | 44,546 | 0.51% |
| | Validation | 46 | 4,954 | 0.92% |
| | Test | 100 | 9,900 | 10.00% |
| ChestMNIST | Training | 1,994 | 74,480 | 2.61% |
| | Validation | 625 | 10,594 | 5.57% |
| | Test | 1,133 | 21,300 | 5.05% |
| Insurance | Training | 2,863 | 4,028,889 | 0.07% |
| | Validation | 265 | 1,059,078 | 0.03% |
| | Test | 132 | 1,085,053 | 0.01% |

Table 7: Data statistics (without flipping).

## F  More Experimental Results

In this section, we show ablation studies to test the algorithm performance under different settings.

In Table 8, we show experiments of varying number of clients. The advantages of our methods are preserved.

Table 9 reports the results over three independent rounds of random data removal and, for each round, three runs with different random seeds. Our methods consistently outperform the baselines across all repetitions.

Table 8: Ablation Number of Clients ($dir$=0.1, $flip$=0)

| | Cifar-10 | | Cifar-100 | |
| --- | --- | --- | --- | --- |
| | 50 client | 200 clients | 50 client | 200 clients |
| CyCp-FedAVG | $0.7227 \pm 0.0015$ | $0.7846 \pm 0.0011$ | $0.8817 \pm 0.0020$ | $0.8920 \pm 0.0018$ |
| A-FedAVG | $0.7418 \pm 0.0026$ | $0.7923 \pm 0.0016$ | $0.8756 \pm 0.0025$ | $0.8973 \pm 0.0012$ |
| A-SCAFFOLD | $0.7450 \pm 0.0024$ | $0.7940 \pm 0.0013$ | $0.8866 \pm 0.0017$ | $0.9012 \pm 0.0015$ |
| RS-Minimax | $0.7572 \pm 0.0018$ | $0.8012 \pm 0.0006$ | $0.8994 \pm 0.0013$ | $0.9183 \pm 0.0013$ |
| RS-Pairwise | $0.7721 \pm 0.0014$ | $0.8225 \pm 0.0010$ | $0.9188 \pm 0.0012$ | $\mathbf{0.9422 \pm 0.0021}$ |
| CyCp-Minimax | $\mathbf{0.8209 \pm 0.0012}$ | $\mathbf{0.8478 \pm 0.0016}$ | $\mathbf{0.9275 \pm 0.0022}$ | $0.9365 \pm 0.0016$ |
| CyCp-Pairwise | $\mathbf{0.8367 \pm 0.0013}$ | $\mathbf{0.8448 \pm 0.0017}$ | $\mathbf{0.9647 \pm 0.0015}$ | $\mathbf{0.9653 \pm 0.0019}$ |

Table 9: Repeated Experiments on Randomly Removing Data. ($dir$=0.5 for Cifar-10/100)

| | Cifar-10 | Cifar-100 |
| --- | --- | --- |
| CyCp-FedAVG | $0.8083 \pm 0.0175$ | $0.8815 \pm 0.0073$ |
| A-FedAVG | $0.8123 \pm 0.0037$ | $0.9006 \pm 0.0052$ |
| A-SCAFFOLD | $0.8208 \pm 0.0041$ | $0.9016 \pm 0.0039$ |
| RS-Minimax | $0.8225 \pm 0.0028$ | $0.9342 \pm 0.0044$ |
| RS-Pairwise | $0.8303 \pm 0.0019$ | $\mathbf{0.9541 \pm 0.0037}$ |
| CyCp-Minimax | $\mathbf{0.8460 \pm 0.0060}$ | $0.9422 \pm 0.0029$ |
| CyCp-Pairwise | $\mathbf{0.8503 \pm 0.0048}$ | $\mathbf{0.9730 \pm 0.0055}$ |

In Figure 2, we divide the data into 10 client groups with 10 clients per group. By varying $M$, which is the number of simultaneously participating clients, we observe that larger values of $M$ lead to faster convergence, confirming the expected speed-up effect.

In Figure 3, we randomly partition the clients ($N = 100$) into different numbers of groups. In each round, a single client is sampled to participate. We observe that larger values of $K$ generally lead to faster convergence, while the case $K = 1$ essentially reduces to the random-sampling baseline. In real-world deployments, naturally formed groups (e.g., by geography, device type, or availability pattern) may exhibit internally similar data distributions. Splitting such groups into smaller ones may cause the model to overfit to a narrow distribution before sufficiently exploring others. Therefore, the optimal choice of $K$ depends on the underlying real-world heterogeneity and grouping structure. A more systematic investigation of how $K$ interacts with real data distributions is an important direction for future work.

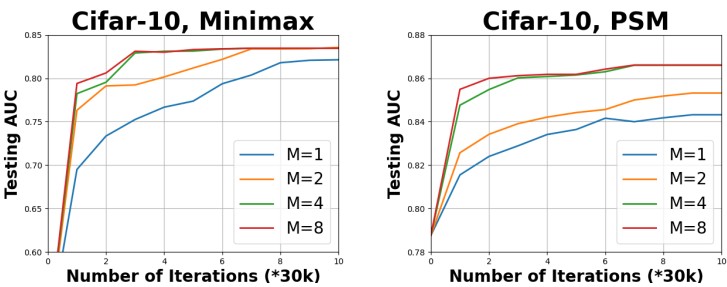

Figure 2: Ablation Study: Effect of the Number of Simultaneously Participating Clients $M$ ($dir = 0.5, flip = 0$)

# G   Applicability to a Broad Class of AUC Maximization Formulations

Table 10 lists multiple AUC-consistent losses and their compatibility with our algorithms.

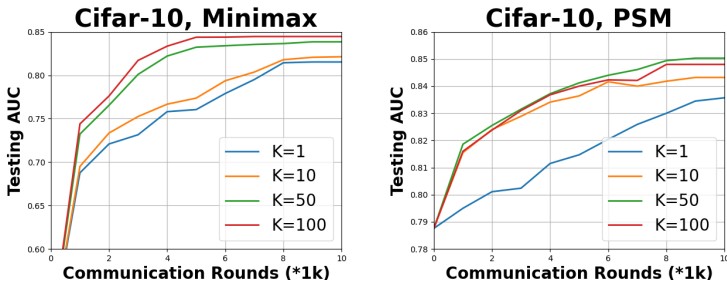

Figure 3: Ablation Study: Effect of the Number of Client Groups $K$ ($dir = 0.5, flip = 0$)

Table 10: Applicability to Different Formulations of AUC Maximization ($m$ denotes a margin constant as a hyper-parameter, and $\tau$ denotes a scaling hyper-parameter).

| Loss | Formulation | Applicable? |
|---|---|---|
| Minimax Loss (Ying et al., 2016) | (3) | Yes (Algorithm 1,2) |
| Pairwise Square (Gao et al., 2013) | (1) with $\psi(a,b) = (m - (a-b))^2$ | Yes (Algorithm 3,4) |
| Pairwise Squared Hinge (Zhao et al., 2011b) | (1) with $\psi(a,b) = (m - (a-b))_+^2$ | Yes (Algorithm 3,4) |
| Pairwise Logistic (Gao & Zhou, 2015) | (1) with $\psi(a,b) = \log(1 + \exp(-s(a-b)))$ | Yes (Algorithm 3,4) |
| Pairwise Sigmoid (Calders & Jaroszewicz, 2007) | (1) with $\psi(a,b) = (1 + \exp(s(a-b)))^{-1}$ | Yes (Algorithm 3,4) |
| Pairwise Barrier Hinge (Charoenphakdee et al., 2019b) | (1) with $\psi(a,b) = \max(m - \tau(m+t),$ $\max(\tau(t-m),\ m-t))$, where $t = a - b$ | Yes/No[1](Algorithm 3,4) |
| q-norm hinge loss | (1) with $\psi(a,b) = (m - (a-b))^q (q > 1)$ | Yes (Algorithm 3,4) |

[1] For the Pairwise Barrier Hinge loss, our algorithms remain directly applicable. However, because this surrogate does not satisfy the smoothness assumption, it does not enjoy the linear speed-up guarantee in theory, which is a common limitation of nonsmooth FL objectives (Yuan et al., 2021a).

