# OpenReview forum: "Communication-Efficient Federated AUC Maximization with Cyclic Client Participation"
_TMLR — Accepted by TMLR_

### Review · Reviewer_M6NE · 2025-11-06

**Summary Of Contributions:**

This paper studies federated AUC maximization under cyclic client participation, a realistic setting where clients join training in a fixed, repeating schedule rather than through random sampling or full participation. The authors make the following contributions:
- Minimax Formulation: For AUC maximization with squared surrogate loss, the authors develop a stagewise algorithm with novel auxiliary sequences to handle biased gradient estimates under cyclic scheduling. Under the PL condition, they achieve $\tilde{\mathcal{O}}(K/(\mu^{3/2} \epsilon^{1/2}))$ communication complexity and $\tilde{\mathcal{O}}(\hat{L}/(\mu^2 M \epsilon))$ iteration complexity, matching rates from full-participation settings.
- Pairwise Formulation: For general pairwise AUC losses, they propose an active-passive gradient decomposition strategy that shares prediction scores across clients. Without PL: $\mathcal{O}(K/\epsilon^3)$ communication and $\mathcal{O}(1/M \epsilon^4)$ iteration complexity. With PL: $\tilde{\mathcal{O}}(K/(\mu^{3/2} \epsilon^{1/2}))$ communication and $\mathcal{O}(1/(\mu^2 M \epsilon))$ iteration complexity.
- Technical Innovation: Introduction of virtual sequences,  that decouple dependencies between client groups, enabling convergence analysis despite deterministic cyclic ordering, a key departure from random sampling analyses.

# Strengths:

- The paper addresses a practically important yet theoretically underexplored setting. Class imbalance is common in FL (e.g., medical, finance), making AUC maximization a more suitable objective than ERM. Addressing this under the realistic (but complex) cyclic participation setting is a valuable contribution.
- The paper provides a comprehensive and rigorous theoretical analysis for both the minimax and general pairwise formulations. The development of auxiliary sequences and the analysis of the two-cycle delay are non-trivial and effectively address the challenges of the deterministic schedule.
- A key finding is that the proposed cyclic-aware algorithms achieve convergence rates (both iteration and communication complexity) that match the best-known rates for the simpler random-participation setting. This strongly suggests that cyclic participation, when handled correctly, does not inherently degrade asymptotic efficiency.
- The experiments are conducted on a diverse set of tasks, including vision, medical, and tabular fraud data. The inclusion of a large-scale, naturally heterogeneous fraud dataset (50 clients) is a good addition. The ablation study on local steps (Figure 1) also effectively supports the claim of communication efficiency.

# Weaknesses:
- A major weakness is the ambiguity in the definition of "cyclic client participation." The introduction and abstract describe it as clients joining in a "fixed, repeating schedule". However, Algorithms 1 and 3 describe a process of cyclically iterating through $K$ groups (for $k \in [K]$) and then randomly sampling $M$ clients from within that group ("Sample M clients from k-th client set uniformly at random w/o replacement"). This appears to be a hybrid "cyclic-group-random-client" model, not a standard "cyclic-client" model (like in Cho et al. 2023 ). This discrepancy between the paper's premise and its algorithms is confusing and needs to be clarified.
- The experimental comparisons are not sufficient to validate the paper's core contribution. The proposed methods (CyCp-Minimax, CyCp-Pairwise) are compared almost exclusively against ERM baselines (CyCp-FedAVG, A-FedAVG, A-SCAFFOLD). It is already well-established that AUC maximization methods outperform ERM methods on imbalanced data. The current experiments (e.g., Tables 1 and 2) primarily re-demonstrate this known fact, rather than demonstrating the necessity of the proposed cyclic-aware AUC algorithms.
- To properly situate the contribution, the experiments must include comparisons against SOTA Federated AUC Algorithms (Random Sampling): The performance of methods like Guo et al. (2020), Yuan et al. (2021a), Guo et al. (2023a),  using standard random client sampling. This would provide a direct comparison of the performance (e.g., final AUC, convergence speed) of the cyclic vs. random participation schemes.
- The abstract states the communication complexity for the minimax problem is $\tilde{O}(1/\epsilon)$. However, the introduction and the formal result in Theorem 4.5 both state it is $\tilde{O}(1/\epsilon^{1/2})$ (under the PL condition). The $\tilde{O}(1/\epsilon)$ rate from the abstract appears to be the iteration complexity. This should be corrected for clarity.
- The transition from biased gradient estimates to unbiased analysis via auxiliary sequences (Lemma 4.3) requires clearer exposition of which random variables are being conditioned on.
- Algorithm 1 references client set $G^{e,k}$ in line 4 but uses $S^{e,k}$ in subsequent lines without clear definition.
- The transition from single-stage (Algorithm 1) to multi-stage (Algorithm 2) framework lacks an intuitive explanation.
- The choice of $\gamma = 2l$ appears arbitrary and lacks justification.
- The paper doesn't discuss computational overhead of maintaining auxiliary sequences or historical prediction scores ($H^e_1$, $H^e_2$).
- In Algorithm 3, line 20 mentions "Update buffer $B_{i,1}$, $B_{i,2}$ using $R^{e,k-1}{i,1}$, $R^{e,k-1}{i,2}$ with shuffling". What is the shuffling procedure, and why is it important? Specify buffer size management for $H^e_1$, $H^e_2$.
- How does the method perform when the number of groups $K$ varies? Is there a sweet spot for $K$ given $N$ clients?

**Audience:**

Yes

**Audience Explanation:**

This paper is a strong fit for the TMLR audience. It addresses a core machine learning problem (AUC maximization), applies it to a major research area (federated learning), and provides a rigorous, in-depth theoretical optimization analysis.

**Broader Impact Concerns:**

Not applicable. This work is theoretical and algorithmic, focused on improving optimization methods. It does not appear to have immediate, direct broader impact concerns.

**Claims And Evidence:**

Yes

**Claims Explanation:**

The theoretical claims are extensively supported by detailed proofs in the appendix. The empirical claims (i.e., that the proposed methods outperform ERM baselines) are supported by the provided tables. However, as noted in the weaknesses, the evidence is insufficient to support the more crucial implicit claim: that these new algorithms are superior to existing AUC algorithms in the cyclic setting.

**Requested Changes:**

See the above weakness section.

---

> ### Author Response · Authors · 2025-11-25
> **Response to Reviewer M6NE Part I**
>
> We are grateful for the reviewer’s rigorous assessment and constructive viewpoints. The revised manuscript has been uploaded, and we address each point below.
>
> ***Q1: A major weakness is the ambiguity in the definition of "cyclic client participation."***
>
> ***A:*** Thank you for raising this point, as it has helped us clarify the presentation and better motivate our work. Our setup follows Cho et al. (2023, Section 3, Problem Formulation: \url{https://arxiv.org/pdf/2302.03109}). We divide the $N$ clients into $K$ disjoint groups, each containing $N/K$ clients, denoted by $\mathcal{G}^k$ for $k \in [K]$. During training, the server cycles through these groups in a fixed, pre-determined order $(\mathcal{G}^1, \dots, \mathcal{G}^K)$, implementing a cyclic participation pattern. In each communication round, $M$ clients are selected uniformly at random without replacement from the active group.
>
> This framework is especially relevant for real-world federated learning applications. For instance, in cross-device FL, mobile devices can be grouped by regions or time zones, while random sampling within each group provides flexibility especially when the number of clients in a group is large. For simplicity, the framework can also be interpreted as a fully deterministic schedule in the special case of $K=N$ or $M=N/K$, where all clients in the active group participate in every round; in this case, the theoretical guarantees still hold.
>
> These discussions have been incorporated into the preliminary section and are highlighted in orange.
>
> ***Q2: The experiments must include comparisons against SOTA Federated AUC Algorithms (Random Sampling)***
>
> ***A:*** We have added comparisons to both the Minimax method (Guo et al., 2020) and the pairwise loss method (Guo et al., 2023a) under random client sampling in all experimental tables. These additional baselines appear as RS-Minimax and RS-PSM, respectively. Our proposed methods, CyCP-Minimax and CyCP-PSM, achieve superior performance to these random-sampling baselines in most cases since random sampling does not ensure comprehensive population coverage, which is essential for mitigating non-IID bias.
>
> We also emphasize that random sampling implicitly assumes that all sampled clients are available to participate, an assumption that often breaks down in real-world federated environments. In contrast, cyclic client participation overcomes these limitations by utilizing predictable participation.
>
> We have incorporated this discussion into the experiment section (marked in orange). In addition, we expanded the explanation of practical cyclic participation scenarios in the Introduction (beginning of page 2) and in Section 2.2, both marked in red.
>
> ***Q3: The abstract states the communication complexity for the minimax problem is $\widetilde{O}(1/\epsilon)$. However, the introduction and the formal result in Theorem 4.5 both state it is $\widetilde{O}(1/\epsilon^{1/2})$ (under the PL condition).***
>
> ***A:*** The communication cost under the PL condition should indeed be $\widetilde{O}(1/\epsilon^{1/2})$. We have corrected this in the abstract of the revision. Thank you for noting the discrepancy.
>
> ***Q4: The transition from biased gradient estimates to unbiased analysis via auxiliary sequences (Lemma 4.3) requires clearer exposition of which random variables are being conditioned on.***
>
> ***A:*** Thank you for the comment. We have clarified this in Lemma 4.3 by introducing the subscript {$\mathbf{E}\_{e,0}$},and specifying that "where $\mathbf{E}\_{e,0}$ denotes expectation with respect to all randomness realized prior to epoch $e$". This clarification is temporarily highlighted in orange.
>
> ***Q5: Algorithm 1 references client set $\mathcal{G}^{e,k}$ in line 4 but uses $\mathcal{S}^{e,k}$ in subsequent lines without clear definition.***
>
> ***A:*** Thank you for catching this oversight. We have revised the notation so that $\mathcal{G}$ is used consistently to denote the client set.
>
> ***Q6: The transition from single-stage (Algorithm 1) to multi-stage (Algorithm 2) framework lacks an intuitive explanation.***
>
> ***A:*** Thank you for the suggestion to improve clarity. We have added the following remark under Lemma 4.4 to provide intuition for the transition from a single-stage to a multi-stage algorithm: ``Lemma above shows that the bound for the output of a stage (Algorithm 1) depends on the quality of its inputs, $\mathbf{v}_0$ and $\alpha_0$, which are the outputs of the previous stage. By employing a stagewise algorithm (Algorithm 2) and setting parameters appropriately, we can ensure that the duality gap decreases exponentially across stages.''
> This remark is temporarily highlighted in orange. In addition, we have included overviews of the algorithms in Sections 4 and 5, temporarily highlighted in green.

---

> ### Author Response · Authors · 2025-11-25
> **Response to Reviewer M6NE Part II**
>
> ***Q7: The choice of $\gamma=2\ell$ appears arbitrary and lacks justification.***
>
> ***A: ***
> Since $f(\mathbf{v}, \alpha; \mathbf{z})$ is $\ell$-smooth in $\mathbf{v}$, it is also $\ell$-weakly convex in $\mathbf{v}$.
> Accordingly, in the subsequent analysis we choose $\gamma = 2\ell$, which guarantees that the subproblem~(5) in each stage becomes $\ell$-strongly convex in $\mathbf{v}$. We have added this explanation in the remark under Assumption 4.1 to justify setting $\gamma=2\ell$, temporarily highlighted in orange.
>
> ***Q8: The paper doesn't discuss computational overhead of maintaining auxiliary sequences or historical prediction scores $(\mathcal{H}^e\_1, \mathcal{H}^e\_2)$.***
>
> ***A:*** First, the auxiliary sequences $\hat{\mathbf{v}}^{e,0}$, $\tilde{\mathbf{v}}^{e,0}$, $\hat{\alpha}^{e,0}$, and $\tilde{\alpha}^{e,0}$ are purely conceptual and require no actual computation. We have added this clarification prior to Lemma 4.3 (currently marked in orange). Second, the prediction scores $(\mathcal{H}^e\_1, \mathcal{H}^e\_2)$ incur no additional computation, as they simply reuse the scores already produced during local updates. This discussion has been added in the remark under Theorem 5.2 (also marked in orange).
>
> ***Q9: In Algorithm 3, line 20 mentions "Update buffer $\mathcal{B}\_{m,1},\mathcal{B}\_{m,2}$ using $\mathcal{R}
> ^{e,k-1}\_{i,1}, \mathcal{R}^{e,k-1}\_{m,2}$, with shuffling". What is the shuffling procedure, and why is it important? Specify buffer size management for $(\mathcal{H}^e\_1, \mathcal{H}^e\_2)$.***
>
> ***A:*** The buffer update procedures in Algorithm 3 have been revised for improved clarity. Shuffling simply rearranges the received prediction scores and stores them in the local buffer. Shuffling ensures that, over time, each client has the opportunity to interact with every other client.  Since $(\mathcal{H}^e\_1, \mathcal{H}^e\_2)$ retains only the scores from the previous communication round, the required memory is $O(IMK)$, where $I$ is the communication interval, $K$ is the number of client groups, and $M$ is the number of simultaneously participating clients per group. This storage requirement is negligible compared with the number of parameters in a modern neural network and can be adjusted in practice by tuning $M$ or $I$. Each client’s local buffers, $\mathcal{B}\_{m,1}$ and $\mathcal{B}\_{m,2}$, are of size $O(I)$, which stores sufficient historical predictions used to construct the loss function at every local update iteration.
>
> We have added this discussion to the remark following Theorem 5.2, currently highlighted in orange.
>
> ***Q10:How does the method perform when the number of groups
> $K$ varies? Is there a sweet spot for $K$ given $N$ clients?***
>
> ***A:*** We thank the reviewer for bringing this to our attention. We have added ablation experiments in the revision. In Figure 3 of Appendix F, we randomly partition the clients $(N=100)$ into different numbers of groups. We observe that larger values of $K$ generally lead to faster convergence, while the case $K=1$ essentially reduces to the random-sampling baseline. This trend is consistent with the findings of Cho et al. (2023).
>
> However, we emphasize that in real-world deployments, naturally formed groups (e.g., by geography, device type, or availability pattern) may exhibit internally similar data distributions. Splitting such groups into smaller ones may cause the model to overfit to a narrow distribution before sufficiently exploring others. Therefore, the optimal choice of $K$ depends on the underlying real-world heterogeneity and grouping structure. A more systematic investigation of how $K$ interacts with real data distributions is an important direction for future work. This discussion is also added to Appendix F.

---

> > ### Comment · Reviewer_M6NE · 2025-11-27
> > **Reply Reviewer M6NE**
> >
> > The authors have satisfactorily addressed most of my comments with appropriate revisions to the manuscript. The paper is now clearer in its problem formulation, includes necessary baseline comparisons, and provides adequate justification for algorithmic choices. However, I have some follow-up questions:
> > - While I appreciate the addition of RS-Minimax and RS-Pairwise baselines, I have concerns about whether the comparison is entirely fair. In cyclic participation, all clients are guaranteed to participate within each meta-epoch. With random sampling, some clients may be sampled multiple times while others are never sampled within the same period. Could you clarify whether the total number of client participations per epoch is matched between cyclic and random sampling?
> > - The communication complexity is $\mathcal{O}(K/(\mu^{3/2} \epsilon^{1/2}))$, which scales linearly with $K$. This raises practical concerns. In the ablation (Figure 3), $K = 100$ shows good convergence, but how does the total communication cost compare across different $K$ values? Faster convergence in iterations doesn't necessarily mean lower total communication. Could you provide a plot showing communication cost vs. final AUC for different $K$ values?

---

> > > ### Author Response · Authors · 2025-11-27
> > > **Response to follow-ups of Reviewer M6NE**
> > >
> > > We are glad to hear that most of your concerns have been resolved. Below we address your two follow-up questions.
> > >
> > > ***Q1: Concerns about whether the comparison is entirely fair. Could you clarify whether the total number of client participations per epoch is matched between cyclic and random sampling?***
> > >
> > > ***A:*** Thank you for raising this point. While we match the total number of client selections across methods, random sampling inherently allows some clients to be chosen multiple times while others may not be selected within the same period. This variability is an intrinsic limitation of random sampling and is one of the motivations for adopting a cyclic participation scheme (Cho et al., 2023; Zhu et al., 2023). We have added this clarification to the Introduction (top of Page~2), where the new text is highlighted in red.
> > >
> > > ***Q2: The communication complexity scales linearly with $K$. This raises practical concerns. How does the total communication cost compare across different values?***
> > >
> > > ***A:*** We appreciate the question. The linear scaling with $K$ is consistent with prior theoretical results (Cho et al., 2023). Even under random sampling (Karimireddy et al., 2020), the total communication cost increases with the number of clients when the number of participating clients per round is fixed. To make this comparison clearer, we have updated Figure~3 in the revision to use communication rounds as the horizontal axis. Under random client-group participation with the same number of simultaneously participating clients $M$, our empirical results indicate that larger values of $K$ generally lead to faster convergence. However, as noted in our response to an earlier question, we do recognize that the client group participation is an important question in practice for future study.

---

### Review · Reviewer_1g59 · 2025-11-13

**Summary Of Contributions:**

This paper studies federated AUC maximization under a practical client-participation scenario. Federated learning operates in privacy-sensitive settings with decentralized and often non-IID data, making classical ERM less suitable and motivating the use of AUC maximization for class-imbalanced tasks. While much prior work assumes full client availability, this paper instead focuses on a more realistic participation pattern: cyclic schedules.

The authors consider two surrogate loss families. Under the squared surrogate loss, which leads to a minimax formulation, they analyze the cyclic participation setting where independent client sampling does not hold. The proposed stage-wise algorithm achieves the same communication rate as full participation under a PL condition, suggesting that cyclic schedules do not degrade communication efficiency.

The second setting addresses more general pairwise surrogate losses. Here, the paper proposes an active–passive algorithm with gradient sharing, which attains competitive communication complexity and can accommodate delays of up to two full cycles.

The empirical evaluation spans multiple datasets representing common FL application domains, with class imbalance and heterogeneity appropriately simulated. The experiments vary heterogeneity levels and noise conditions, and include ablations on communication efficiency, collectively illustrating the scalability and robustness of the proposed approach.

**Strengths**

The theoretical development is sound and well aligned with the algorithmic design, the methods are clearly presented, and the empirical evaluation spans multiple application domains. Importantly, the paper focuses on a practical and underexplored participation scenario, which is conceptually valuable and may inspire further work on realistic FL settings.

**Weaknesses**

The motivation and introduction could be clearer, especially in explaining the practical meaning of the setups, and the experiment section would benefit from additional ablations to fully support some of the theoretical claims. In addition, the gradient-sharing mechanism raises potential privacy concerns that would benefit from brief discussion.

**Audience:**

Yes

**Audience Explanation:**

This work encourages the community to move beyond idealized FL assumptions and consider more practical deployment factors, including realistic participation patterns and class imbalance in non-IID settings. By highlighting the underexplored cyclic participation scenario, it contributes to research on federated optimization, fairness, and communication efficiency. Combined with empirical evaluation across diverse application domains, the findings may stimulate broader discussion on practical FL settings and non-IID challenges in real-world deployments.

**Broader Impact Concerns:**

One potential concern is gradient sharing in the active–passive method. Sharing gradients may leak sensitive information unless additional protections (e.g., differential privacy) are considered. And this is also a broad topic in FL. The paper would benefit from at least a short discussion of such risks and mitigation strategies. Beyond this, the work does not appear to pose major ethical or societal risks.

**Claims And Evidence:**

Yes

**Claims Explanation:**

The theoretical analyses under both loss settings support the proposed algorithms, and the empirical evaluation covers diverse domains, suggesting reasonable coverage of practical FL scenarios. Overall, the main claims appear plausible, though improving clarity would strengthen the supporting evidence.

**Requested Changes:**

**Clarity and Motivation:**

1. The authors may consider moving some formulas out of the introduction or placing them later in the paper. They interrupt the narrative and may hinder readers from understanding the high-level motivation (your storyline).
2. Please provide a practical explanation or real-life example illustrating how these sets arise (beyond formulas). This would help readers grasp the relevance of the AUC setup.
3. A concrete example from real deployments would strengthen the motivation and clarify why and how valuable the scenario cyclic schedule is in practice.
4. The paper focuses on two surrogate loss families. Could the authors explain why these two are particularly appropriate, and whether other losses might also satisfy similar assumptions or can be generalized from here?
5. Section 2.2 would benefit from a revised explanation from an application perspective. Clarifying the practical meaning of these schemes would make the presentation more aligned with the paper’s motivation.

**Experimental Details:**

1. Please clarify how data were removed to form the binary subsets and whether this procedure reliably simulates real class imbalance. How many times were these simulations repeated? Since Appendix E suggests that some datasets may contain very few positive samples, repeated runs may be important for obtaining stable results.
2. Additional ablations examining how performance scales with an increasing number of clients would further strengthen the empirical section. It would also be good to confirm whether it is necessary to include experiments that directly support or validate the claims in Theorems 5.2 and 5.3.
3. Since participation patterns may depend on the number of clients, could the authors provide experiments showing whether the proposed advantages persist as the client count changes?


**Presentation & Formatting:**

1. Table caption font size: decrease font size for readability in captions.
2. Figure 1 image size, redesign: 2×2 layout or slightly larger font may make the figure clearer.
3. Table formatting: The authors may consider using \small fonts in tables for numbers.
4. Table 2 caption has a missing parenthesis, and captions should include the evaluation metric for clarity.
5. Section 5: “enough” is misspelled, and please check the manuscript for other minor typos.


**Discussion and Limitations:**

1. Broader applicability / general impact: The paper mentions potential broader impact in section 5 but does not concretely explain how the approach can extend to other application domains. A short discussion or example would help.
2. Limitations and future work: A brief paragraph reflecting on limitations (e.g., assumptions on the PL condition, privacy implications of gradient sharing) would improve completeness.

---

> ### Author Response · Authors · 2025-11-25
> **Feedback to Reviewer 1g59 Part I**
>
> We sincerely thank the reviewer for the careful review and helpful perspectives. We have uploaded a revised version, and our replies are presented below.
>
> ***Q1: The motivation and introduction could be clearer, especially in explaining the practical meaning of the setups.***
>
> ***A:*** Thank you for the suggestion. We have added the following explanation to the Introduction at the beginning of page 2, currently highlighted in red:
>
> "Cyclic Client Participation (CyCP) is a structured approach designed to transition FL from idealized models to practical, real-world deployments where client availability is inherently intermittent due to factors such as battery limitations and unstable network connectivity (Huba et al., 2022; Paulik et al., 2021). This structured scheduling addresses diverse availability challenges, ranging from the natural periodicity of cross-device FL, where clients operate in different time zones or charge their devices at preferred times (Cho et al., 2023; Yang et al., 2018), to the rigorous, planned constraints of cross-silo FL, such as hospitals facing scheduling challenges due to large local data processing, internal network security, or planned IT maintenance windows.
> By enforcing a guarantee that all clients or client groups participate within a defined meta epoch, CyCP
> offers significant benefits over purely random sampling: it ensures comprehensive data coverage across the
> entire population—an important property for mitigating non-IID data bias (Cho et al., 2023; Zhu et al.,
> 2023)—and, critically, this controlled, predictable participation frequency enhances privacy preservation by
> strictly limiting how often each client contributes with the global model (Kairouz et al., 2021a)."
>
>
> ***Q2:  A concrete example from real deployments would strengthen the motivation and clarify why and how valuable the scenario cyclic schedule is in practice. Section 2.2 would benefit from a revised explanation from an application perspective***
>
> ***A:*** In addition to our response to your first comment, we have updated Section 2.2 to include a real-world scenario, temporarily highlighted in red: "This structured scheduling addresses diverse availability challenges. In cross-device
> FL, clients may operate in different time zones or charge their devices at preferred times (Cho et al., 2023;
> Yang et al., 2018). In cross-silo FL, clients such as hospitals face planned constraints, including managing
> large local datasets, internal network security, or scheduled IT maintenance windows. "
>
> ***Q3: The experiment section would benefit from additional ablations to fully support some of the theoretical claims.***
>
> ***A:*** Thank you for the suggestions. We have added the following experiments for completeness. First, in all tables of experimental results, we now include two new baselines on minimax and pairwise algorithms with random sampling. Our methods remain advantageous. Second, we added Figure 2 to Appendix F, where we show ablation study by varying $M$, which is the number of simultaneously participating clients, we observe that larger values of $M$ lead to faster convergence, confirming the expected speed-up effect. Third, we also added Table 8 in Appendix F, which includes experiments of varying number of clients. The advantages of our methods are preserved under different settings.  Lastly, We have also added Figure 3 to Appendix F with varying number of client groups.
>
> ***Q4: The gradient-sharing mechanism raises potential privacy concerns that would benefit from brief discussion.***
>
> ***A:*** We would like to clarify that our framework does not share gradients. Instead, clients share model parameters and prediction scores, similar to standard practice in many federated optimization methods. We have added a concise discussion of privacy considerations as a bullet point at the end of Section 5. The added text, temporarily highlighted in red, reads: "Privacy Considerations. The two algorithmic frameworks proposed in this work require clients to share model parameters—and, for pairwise objectives, to additionally share prediction scores—to enable collaborative optimization, consistent with prior literature (Guo et al., 2023a; McMahan et al., 2017). However, such information exchange can introduce privacy risks, as model updates and related signals may potentially leak information about individual data points (Zhu et al., 2019). These risks can be mitigated through several techniques, including: 1) adding noise to ensure differential privacy (Abadi et al., 2016; McMahan et al., 2018; Truex et al., 2020; Wei et al., 2020); 2) quantization (Kang et al., 2024; Youn et al., 2023; Xu et al., 2025); 3) dropout (Jain et al., 2015); and 4) homomorphic encryption during aggregation (Jin et al., 2023; Fang \& Qian, 2021)."

---

> ### Author Response · Authors · 2025-11-25
> **Feedback to Reviewer 1g59 Part II**
>
> ***Q5: Consider relocating formulas from the introduction to later sections, as they disrupt the narrative and make it harder for readers to follow the high-level motivation.***
>
> ***A:*** Thank you for the suggestion. We have removed the two specific formulations of AUC maximization problems from the introduction. The revised text is temporarily highlighted in red. These formulations are presented in details at the beginning of Sections 4 and 5.
>
> ***Q6: The paper focuses on two surrogate loss families. Could the authors explain why these two are particularly appropriate, and whether other losses might also satisfy similar assumptions or can be generalized from here?***
>
> ***A:*** The minimax formulation avoids the explicit construction of positive–negative pairs, making it naturally suited for online learning, where data arrive sequentially, and for federated settings, where data are distributed across many devices. This simplifies implementation and avoids the need for cross-client pair management.
>
> However, the minimax formulation does not cover all AUC surrogate losses (e.g., Zhao et al., 2011a; Kotłowski et al., 2011; Gao \& Zhou, 2015; Calders \& Jaroszewicz, 2007; Charoenphakdee et al., 2019a). In particular, symmetric pairwise losses have been shown to be more robust to label noise than the squared surrogate loss (Charoenphakdee et al., 2019b; Zhu et al., 2022).
>
> This explanation has been added on the second page of the introduction, temporarily marked in italics and purple.
>
> ***Q7: Please clarify how data were removed to form the binary subsets and whether this procedure reliably simulates real class imbalance. How many times were these simulations repeated? Repeated runs may be important for obtaining stable results.***
>
> ***A:*** For CIFAR-10 and CIFAR-100, we create binary subsets by randomly removing samples from the positive class, following the standard practice in prior work (Guo et al., 2020; Yuan et al., 2021a; Guo et al., 2023a). In the revision, we additionally include results averaged over three independent runs of this random removal procedure (see Table 9 in Appendix F). These results show that our methods consistently outperform the baselines under repeated simulations, demonstrating the stability.
>
> ***Q8: Regarding Presentation \& Formatting***
>
> ***A:*** Thank you for the constructive suggestions. In the revision, we have addressed these issues as recommended, including reducing the font sizes in table captions and entries, reorganizing Figure 1 into a 2$\times$2 layout, and correcting several typos.
>
> ***Q9: The paper mentions potential broader impact in section 5 but does not concretely explain how the approach can extend to other application domains. A short discussion or example would help.***
>
> ***A:*** We have added a concrete example to the General Impact bullet point in Section 5, illustrating how metric learning fits into our framework.
>
> ***Q10: A brief paragraph reflecting on limitations would improve completeness.***
>
> ***A:*** We have added a new section at the end of the paper to discuss limitations and future work, temporarily highlighted in red. Specifically, we include: "Our work has several limitations. First, the fast communication rate of $\widetilde{O}(1/\epsilon^{1/2})$ depends on the PL condition, which may limit the generalizability of the associated algorithms. Second, the convergence rates in both the PL and non-PL regimes have not yet reached known lower bounds, leaving room for improvement. Third, developing methods to formally ensure differential privacy remains an open question. Finally, extending our approach beyond AUC maximization to a broader class of non-ERM objectives constitutes an important direction for future research."

---

> > ### Comment · Reviewer_1g59 · 2025-11-28
> > **Follow-up Question**
> >
> > The authors have thoroughly addressed all of the concerns raised in my initial review. The revised manuscript now presents clearer motivation, more practical explanations of the setups, real-world deployment examples, additional ablation studies, and improved discussions on privacy considerations and surrogate loss choices. The clarifications on binary dataset construction, the inclusion of a limitations section, and the corrections to formatting and presentation further enhance the clarity and completeness of the work. I am satisfied with these revisions, and the issues I previously raised have been fully resolved.
> >
> > In addition, I have one follow-up question regarding the scope of the theoretical insights. The revision clarified why the squared minimax loss and symmetric pairwise losses are the focus of this work and how they operate in the FL setting. One question that remains is to what extent the theoretical conclusions may generalize to a broader family of AUC surrogate losses. Specifically, which arguments or properties appear readily transferable across different surrogates, and which aspects seem fundamentally loss-specific? I am not asking for additional proof; rather, I am seeking a high-level intuition on how likely these PL-type assumptions and convergence arguments are to extend to more general ranking-based objectives, and whether the current results contribute toward a more unified perspective on federated AUC maximization under cyclic participation

---

> > > ### Author Response · Authors · 2025-11-29
> > > **Response the Follow-up Question of Reviewer 1g59**
> > >
> > > It is reassuring to hear that the reviewer’s earlier concerns have been satisfactorily addressed. Below we respond to the follow-up question.
> > >
> > > While the minimax-based algorithm is inherently loss-specific, the pairwise algorithm is sufficiently general to accommodate a broad class of AUC-consistent objectives. In the revised version, we have added Appendix G, which summarizes multiple AUC-consistent loss functions and discusses their compatibility with our algorithms. All of these losses are supported, with the exception of non-smooth surrogates such as the Pairwise Barrier Hinge. Although our algorithms can still be applied to the Pairwise Barrier Hinge loss, its lack of smoothness prevents us from establishing the linear speed-up guarantee in theory—an expected limitation for nonsmooth FL objectives  (Yuan et al., 2021a).
> > >
> > > Regarding the PL condition, it has been established for minimax formulations (Guo et al., 2023b) and is widely used in the pairwise optimization literature (Yang et al., 2021c). Importantly, our Algorithm 3 for general pairwise losses does not rely on the PL condition, yet still achieves communication efficiency—meaning that the communication complexity is lower than the iteration complexity—as well as linear speed-up (see Theorem 5.2 and the accompanying remark).

---

### Review · Reviewer_1VvX · 2025-11-14

**Summary Of Contributions:**

The paper tackles the problem of federated AUC maximization under the realistic constraint of cyclic client participation. Here, clients join training in a fixed schedule rather than being always available or randomly sampled. The main contributions are
- Reformulating AUC maximization with squared surrogate loss as a nonconvex, strongly concave minimax problem. It proposes algorithm that accounts for deterministic client scheduling.
- Demonstrating empirical evaluations.

**Audience:**

Yes

**Audience Explanation:**

- TMLR’s audience includes researchers in federated learning, optimization, and machine learning theory.
- The paper addresses a practical and underexplored problem, which is cyclic client participation. It common in real-world deployments.

**Claims And Evidence:**

Yes

**Claims Explanation:**

The paper provides rigorous theoretical analysis with lemmas and theorems establishing convergence rates and communication complexity bounds. It carefully addresses the bias introduced by cyclic scheduling using virtual sequences and delayed gradient decomposition.

**Requested Changes:**

There are some notational inconsistencies such as,
- In equation 5 is $F(w, a, b, \alpha, z)$ was supposed to be $F(w, a, b, \alpha; z)$ which is also defiend later?
- Sometime $h()$ is used as $h(w;z)$ and sometime it is $h(w,z)$. These are confusing.
- Since the analysis uses multiple notations, all of them must be clearly defined in the preliminary, such as $v \in \mathbb{R}^{d+2}$ because $v = [w,a,b]$. In algorithm 1 what is $v^{e+1}\_{m}$ and how does this relate with $v^{e,k}_{m}$.
- Furthermore, add an overview of the algorithm to strengthen its readability.

---

> ### Author Response · Authors · 2025-11-25
> **Response to Reviewer 1VvX**
>
> We appreciate the reviewer's comments and constructive suggestions. A revised version has been uploaded, and our responses are provided below.
>
> ***Q1:In equation 5 is was $F(\mathbf{w}, a, b, \alpha, \mathbf{z})$ supposed to be $F(\mathbf{w}, a, b, \alpha; \mathbf{z})$ which is also defined later?***
>
> ***A:*** Yes. We have corrected this in the revision. Thank you for pointing it out.
>
> ***Q2: Sometime $h()$ is used as $h(\mathbf{w};\mathbf{z})$ and sometime it is $h(\mathbf{w},\mathbf{z})$. These are confusing.***
>
> ***A:*** We have revised the manuscript to consistently use the notation $h(\mathbf{w};\mathbf{z})$. Thank you for bringing this to our attention.
>
> ***Q3: Since the analysis uses multiple notations, all of them must be clearly defined in the preliminary***
>
> ***A:*** We have clarified notations in the preliminaries and added a notation summary in Appendix A. For instance, we now explicitly state that the notation $\mathbf{v}_m^{e,k}$ represents the final output of client $m$ in group $k$ at epoch $e$, while $\mathbf{v}^{e,k}$ represents the average of the participating clients. The previous notation $\mathbf{v}_m^{e+1}$ was a typo and has been corrected to $\mathbf{v}_m^{e,k}$.
>
> ***Q4: Add an overview of the algorithm to strengthen its readability.***
>
> ***A:*** We have added the following overview in Section 4 for minimax optimization, currently highlighted in green:
> "Each stage initializes the primal and dual variables using the outputs from the previous stage. Within a stage, the algorithm runs for multiple epochs, during which all client groups are visited multiple times. Each client group samples a subset of clients to participate, performs $I$ local update steps, and then passes the updated primal and dual variables to the next client group. The primal variables are updated via stochastic gradient descent, while the dual variable is updated via stochastic gradient ascent. After completing a stage, the step sizes and number of epochs are adjusted before proceeding to the next stage."
>
>
> We have also added the following overview in Section 5 for pairwise optimization, currently highlighted in green:
> "The One-Stage Federated Pairwise (Algorithm 3: OSFP) algorithm executes a single stage of federated optimization by iterating over multiple epochs. Within each epoch, it sequentially processes each client group. For every group, the server samples a subset of clients and broadcasts the current global model along with reference prediction sets. Each selected client then performs a LocalUpdate procedure, which maintains buffers of past predictions, samples new data points, computes pairwise losses and their gradients, and updates the local model using stochastic gradient steps. After completing local computations, clients return the updated models and prediction sets to the server, which aggregates them to update the global model. Once all epochs are completed, OSFP outputs the averaged global model. Furthermore, if the PL condition is satisfied, an outer loop (Algorithm 4) repeatedly calls OSFP over multiple stages, adjusting learning rates and epoch schedules to progressively refine the model."

---

> > ### Comment · Reviewer_1VvX · 2025-11-29
> >
> > Thank you for clarifying all the doubts. I am satisfied.

---

### Decision · Action_Editor_hKwa · 2025-12-22

**Recommendation:** Accept as is

**Audience:**

Yes

**Audience Explanation:**

Federated learning is an active area of research inside the ML community and this paper makes a contribution in this area.

**Claims And Evidence:**

Yes

**Claims Explanation:**

The paper addresses the optimization challenges of federated AUC maximization under cyclic client participation. This is claimed to be a more realistic setting, but introduces various challenges. The paper introduces a stagewise algorithm for squared surrogate losses and an active-passive strategy for general pairwise losses. While empirical evaluations across diverse domains verify the approach's efficiency, the reliance on gradient sharing for general pairwise losses introduces potential privacy concerns that remain are not addressed.